# Rapidly-changing subglacial hydrological pathways at a tidewater glacier revealed through simultaneous observations of water pressure, supraglacial lakes, meltwater plumes and surface velocities

Penelope How[1,2], Douglas I. Benn[3], Nicholas R.J. Hulton[1,2], Bryn Hubbard[4], Adrian Luckman[5,6], Heïdi Sevestre[3], Ward J.J. van Pelt[7], Katrin Lindbäck[8], Jack Kohler[8], and Wim Boot[9]

[1]Institute of Geography, School of GeoSciences, University of Edinburgh, Edinburgh, EH8 9XP, UK
[2]Department of Arctic Geology, University Centre in Svalbard (UNIS), Longyearbyen, PO Box 156, N-9171, Norway
[3]Department of Geography and Sustainable Development, University of St. Andrews, Fife, KY16 9AJ, UK
[4]Centre for Glaciology, Department of Geography and Earth Sciences, Aberystwyth University, Aberystwyth, SY23 3DB, UK
[5]Department of Geography, College of Science, Swansea University, Swansea, SA2 8PP, UK
[6]Department of Arctic Geophysics, University Centre in Svalbard (UNIS), Longyearbyen, PO Box, 156, N-9171, Norway
[7]Department of Earth Sciences, Uppsala University, Uppsala, PO Box 256, 751 05, Sweden
[8]Norwegian Polar Institute (NPI), Fram Centre, Tromsø, PO Box 6606, NO-9296, Norway
[9]Institute of Marine and Atmospheric Research (IMAU), Utrecht University, Utrecht, 3584 CC, The Netherlands

*Correspondence to:* Penelope How (p.how@ed.ac.uk)

**Abstract.** Subglacial hydrological processes at tidewater glaciers remain poorly understood due to the difficulty in obtaining direct measurements and lack of empirical verification for modelling approaches. Here, we investigate the subglacial hydrology of Kronebreen, a fast-flowing tidewater glacier in Svalbard during the 2014 melt season. We combine observations of borehole water-pressure, supraglacial lake drainage, surface velocities and plume activity with modelled runoff and water routing to de-velop a conceptual model that thoroughly encapsulates subglacial drainage at a tidewater glacier. Simultaneous measurements suggest that an early-season episode of subglacial flushing took place during our observation period, and a stable efficient drainage system effectively transported subglacial water through the north region of the glacier tongue. Drainage pathways through the central/south region of the glacier tongue were disrupted throughout the following melt season. Periodic plume activity at the terminus appears to be a signal for modulated subglacial pulsing i.e. an internally-driven storage and release of subglacial meltwater that operates independent of marine influences. This storage is a key control on ice flow in the 2014 melt season. Evidence from this work, and previous studies, strongly suggests that long-term changes in ice flow at Kronebreen are controlled by the location of efficient/inefficient drainage and the position of regions where water is stored and released.

# 1 Introduction

Subglacial hydrological processes at tidewater glaciers remain poorly understood due to the difficulty in obtaining direct measurements. Borehole data provide spatially-limited information and are often problematic in terms of relating discrete findings to glacier-wide processes. Modelling approaches can approximate the hydrological inputs and routing of subglacial meltwater beneath the glacier but commonly lack empirical verification. In recent years, studies have focused on indirect measurements to advance understanding of these processes, most prominently in terms of investigating supraglacial lake levels and the surface expressions of submarine meltwater plumes (e.g., Everett et al., 2016; Slater et al., 2017). However, simultaneous measurements of all these manifestations of the subglacial system are rare (e.g., Kamb et al., 1994; Sugiyama et al., 2011).

In this paper we adopt four complementary approaches to reconstruct the subglacial hydrology of Kronebreen, a fast-flowing tidewater glacier in Svalbard, through the summer melt season of 2014: (i) borehole data, to document subglacial water-pressure changes; (ii) time-lapse photogrammetry, to record supraglacial water storage and drainage, and meltwater plume activity at high temporal resolution; (iii) modelled surface melt, runoff and subglacial hydraulic potential to investigate meltwater generation and routing; and (iv) surface velocities from analysis of satellite image pairs to examine subglacial hydrology in relation to glacier dynamics.

# 2 Background

The presence of subglacial meltwater is understood to govern the basal water-pressure at the bed of a glacier (Meier et al., 1994; Bartholomew et al., 2010). Measurements of water-pressure via boreholes and moulins reflect complex changes in bed dynamics over time (e.g., Kavanaugh, 2009). Similarities and differences between borehole pairs have previously been used to diagnose and characterise local bed environments (e.g., Hubbard et al., 1995; Lefeuvre et al., 2015). Temporal variations, such as diurnal oscillations and rapid changes (i.e. changes between 0–12 hours), have been linked to changes in subglacial hydrology such as conduit growth, reorganisation of meltwater pathways, and pulsing related to episodic ice motion (e.g., Murray and Clarke, 1995; Kavanaugh and Clarke, 2001; Schoof et al., 2014). Consistently high basal water-pressures have also been observed over long periods of the melt season (e.g., Meier and Post, 1987; Jansson, 1995). It has been suggested that this is associated with meltwater storage in distributed regions of the subglacial zone. It has also been attributed to basal hydraulic systems that are not operating at atmospheric pressure, such as at lake-terminating and tidewater glaciers, where there is an inefficient evacuation of meltwater because the hydraulic gradient is small (e.g., Sugiyama et al., 2011).

Changes in basal water pressures have been linked to enhanced basal sliding and surface velocities at land-terminating Greenland outlets and valley glaciers. Velocities typically increase at the beginning of the melt season, which is associated with an influx of surface meltwater to the subglacial environment (Nienow et al., 1998; Howat et al., 2005). Ice velocities stabilise or fall later in the melt season in response to subglacial drainage re-organisation (i.e. changes in the network of channelised and distributed drainage pathways at the beginning and end of the melt season) and the establishment of efficient channels that reduce water-pressure at the bed (Iken and Truffer, 1997; Hewitt, 2013). Precipitation can disrupt drainage due to the high influx of water over a short period of time, in some cases causing speed-ups due to the timing of high-rainfall events

in relation to a melt season (e.g., Doyle et al., 2015). However, first-hand investigations of the role of subglacial hydrology at the terminus region of tidewater glaciers remain virtually absent (e.g., Kamb et al., 1994; Doyle et al., In review).

In recent years, modelling approaches have been adopted to simulate bed dynamics at tidewater glaciers. Ice velocity and basal water-pressure are typically calculated separately before linking them together to create a unifying model (e.g., Schoof, 2010; Pimental and Flowers, 2011). This work shows promise in representing the evolution of the subglacial hydro-dynamic environment. However, implementations of this approach are still imperfect as outputs do not always match real-world ice velocities (e.g., Werder et al., 2013). Difficulties lie in simulating water pressure in response to changing water transport and storage, and in simulating the connection between water pressure and basal sliding (Bueler and van Pelt, 2015).

As outlined above, meltwater typically enters the subglacial environment from the glacier surface via surface melt production. Melt can collect as supraglacial lakes on the glacier surface in topographically low areas where there are few or no drainage pathways. These lakes drain when they become connected to the bed by mechanical processes such as hydrofracturing (Van der Veen, 2007), providing an abrupt injection of meltwater to the subglacial environment. Water in supraglacial lakes can also be sourced from the subglacial zone when water-pressure at the bed exceeds ice overburden, effectively squeezing subglacial water up to the glacier surface. This water often contains entrained subglacial sediment, giving the lake a turbid appearance. Where such connections exist, the water level can be used as a measure of basal water-pressure (Danielson and Sharp, 2013).

The pattern of supraglacial lake drainage is linked to basal water pressure and ice velocity. Supraglacial lakes in the interior regions of South-West Greenland typically drain at progressively higher altitudes through the melt season (e.g., Sundal et al., 2009; Clason et al., 2015). Conversely, lakes have also been observed to drain in a down-glacier progression, albeit such instances are less common (e.g., Everett et al., 2016). However, many of these observations are based on temporally intermittent records (e.g. low repeat-pass satellite imagery). Improved observations (i.e. at a higher temporal resolution) of supraglacial lake drainage events are needed to better understand the differences between near-terminus and inland lake drainage patterns and gain greater insight into their influence on the subglacial environment in tidewater glacier settings.

The hydraulic routing and residency time of subglacial meltwater largely depends on properties of the bed (Hubbard and Nienow, 1997). In near-terminus settings, a rapid input of meltwater has been observed to increase the efficiency of a channelized system by enlarging channels to accommodate the extra meltwater (Andrews et al., 2014). If melt volumes increase due to warming climate, this can lead to a reduction in ice velocity over decadal timescales as pressure falls in an efficient channelised configuration (Tedstone et al., 2015). Below thick ice in the interior of an ice sheet, channels cannot grow as rapidly or sensitively to point inputs, and water evacuation is less efficient. Although it is challenging to directly observe, studies have suggested that water is stored at the bed for longer periods of time in these settings, causing localised areas of ice to uplift from the bed for up to 48 hours (e.g., Stevens et al., 2013).

Meltwater typically leaves the glacier via large subglacial channels that exit at the glacier terminus. At land-terminating glaciers, this meltwater flows through proglacial streams. In ocean-terminating settings, meltwater commonly exits as a fresh (and therefore buoyant) turbulent plume, the dynamics of which are driven by the density contrasts between the cold, fresh glacial water and warmer, saline seawater. A plume can reach neutral buoyancy at depth or rise to the ocean surface depending

on the discharge rate, fjord geometry and the density of the adjacent sea water column (Slater et al., 2015). Plumes promote submarine melting at the terminus as they increase the transfer of heat from the ocean to the submarine part of the ice front, drawing in warm water from the fjord (Straneo et al., 2010; Cowton et al., 2015). Submarine melting and changes in boundary conditions at the calving terminus have been linked to ice acceleration (e.g., Nick et al., 2009) and periods of enhanced glacier retreat (e.g., Luckman et al., 2015; Vallot et al., 2017).

The surface expression of plumes has previously been used as an indication of discharge rate and to infer the subglacial drainage network configuration in the near-terminus zone. Surfacing plumes have largely been observed from satellite (e.g., Darlington, 2015; Bartholomaus et al., 2016) and/or time-lapse imagery (e.g., Schild et al., 2016; Slater et al., 2017). However, there are few measurements of the size, number and locations of plume-related channels (Fried et al., 2015). It is also challenging to acquire a sufficiently high temporal frequency of images to capture typical rates of change in plume behaviour. It is therefore likely that plumes are changeable and much more dynamic than previously considered.

## 3 Study area

Kronebreen is a fast-flowing, tidewater glacier on the west coast of Spitsbergen, Svalbard (78.8°N, 12.7°E) (Figure 1). The glacier consists of a heavily crevassed tongue fed by two ice fields: Holtedahlfonna and Infantfonna. The total area of the glacier catchment is 295.5 km$^2$, with a maximum length of 49.25 km that spans an elevation range of 1345 m (Kargel et al., 2014). The glacier tongue exhibits consistently high surface velocities, making it the fastest non-surging glacier in Svalbard. Velocities near the terminus are typically 1.5–2 m d$^{-1}$ through the winter season and peak at 3–4 m d$^{-1}$ in the summer (Kääb et al., 2005; Eiken and Sund, 2012; Luckman et al., 2015). The seasonal speed-up propagates from the front of the glacier, which is argued to be largely driven by basal lubrication (Schellenberger et al., 2015; Vallot et al., Accepted). There is a clear contrast in surface velocity between the lower section of the tongue and its upper section, controlled by a high-point in the bed topography approximately 4 km from the terminus (Vallot et al., Accepted).

Kronebreen discharges into Kongsfjorden, an Arctic fjord affected by the West-Spitsbergen Current (WSC). The WSC drives warm, saline Atlantic water into the interior Arctic, allowing large influx of warm ocean water into Kongsfjorden. Calving activity persists throughout the year due to the presence of warm sub-surface ocean water, even in the winter season, although there are large seasonal variations (Luckman et al., 2015). The mean annual calving rate has increased in recent years to 0.20 ± 0.05 km$^3$ yr$^{-1}$ (1905–2007), coinciding with increasingly negative surface mass balance (Köhler et al., 2012; Nuth et al., 2012). Kronebreen is currently in a period of rapid retreat, having receded ∼1 km between 2011 and 2016. Strong correlation between bulk calving rates and fjord water temperature indicates that this retreat primarily reflects melting of the glacier front beneath the waterline (Köhler et al., 2012; Luckman et al., 2015; Vallot et al., 2017).

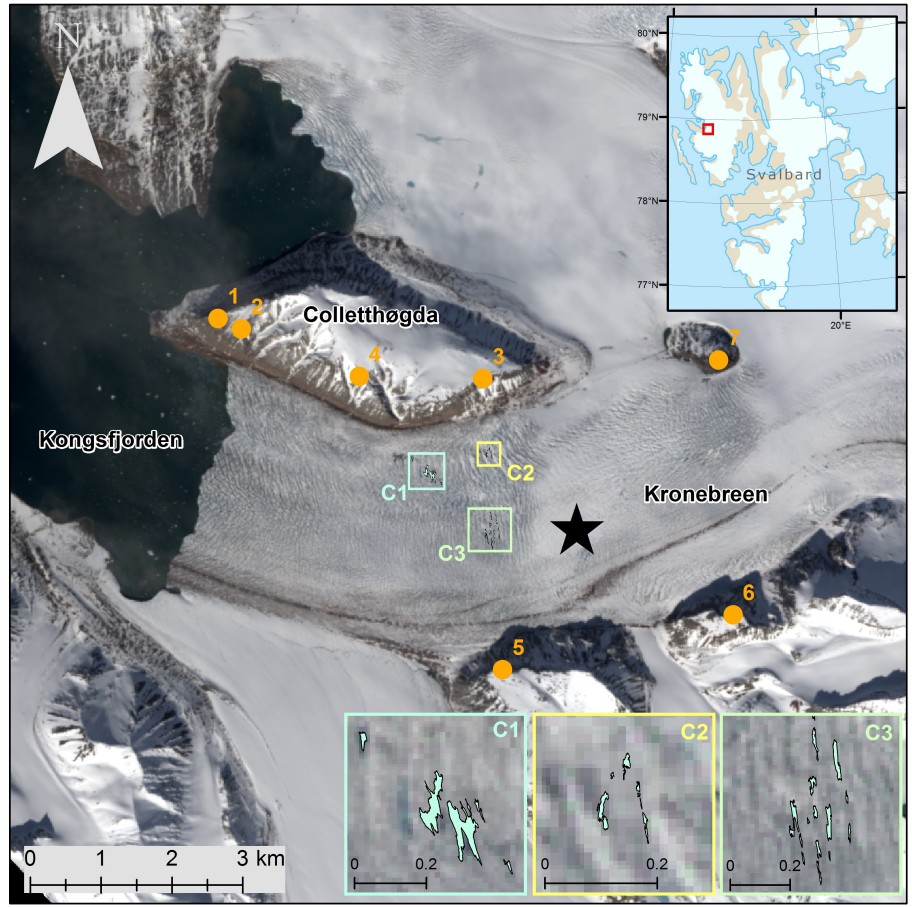

**Figure 1.** Kronebreen, a tidewater glacier situated in Kongsfjorden, Svalbard. The glacier consists of an 8 km-long tongue fed by two ice fields, Holtedahlfonna and Infantfonna. Numbered locations refer to the 2014 time-lapse camera sites, the star marks the location of the borehole drill site, and the coloured boxes refer to the three lake clusters visible from camera sites 3 and 5. Landsat imagery taken on 11 June 2014.

## 4 Methods

### 4.1 Time-lapse photogrammetry

A network of time-lapse cameras was installed on two ridges adjacent to Kronebreen (Colletthøgda and Garwoodtoppen) to gain full coverage over the glacier tongue (Figure 1). Each time-lapse system consisted of a Canon 600D camera body, an EF-S 18-55 mm f/3.5-5.6 IS II zoom lens and a Harbortronics Digisnap 2700 intervalometer, which was powered by a 12 V DC battery and a 10 W solar panel. Each system captured images every 30 minutes from 30th April till 30th September 2014. Of the five cameras that successfully acquired images throughout the season, one trained on the terminus obtained coverage of

surfacing meltwater plumes (Site 1, Figure 1) and two positioned farther up-glacier captured surface lake filling and drainage events (Sites 3 and 5, Figure 1).

Photogrammetric processing was undertaken using PyTrx, a Python-based suite of photogrammetric tools specifically designed for obtaining measurements from time-lapse imagery of glacial environments. PyTrx largely uses processing functions from the OpenCV computer vision toolbox (opencv.org) and georectification tools based on those available in ImGRAFT (imgraft.glaciology.net) (Messerli and Grinsted, 2014). Primarily, the suite was used to extract real-world areas and distances from sequential time-lapse imagery, with a particular focus on the extraction of high-frequency interval measurements. This is achieved by projecting features observed in the 2-D camera image onto their equivalent real-world positions based on camera position and pose, camera lens characteristics and a digital elevation model (DEM) of the observed scene. It is intended to make PyTrx publicly available at a later date.

Several additional datasets were collected to translate measurements from the image plane to three-dimensional space. Camera locations were measured using a Trimble GeoXR GPS rover linked to a SPS855 base station located at Ny Ålesund. Positions were differentially post-processed in a kinematic mode over a ∼15 km baseline using the Trimble Business Centre software to obtain an average horizontal positional accuracy of 1.15 m and an average vertical accuracy of 1.92 m. Ground Control Points (GCPs) were derived from known XYZ locations in the camera field of view. A DEM was obtained from airborne photogrammetric surveying in 2008 by the Norwegian Geodetic Survey, with a 10 m resolution. This DEM was smoothed using a linear interpolation approach to reduce discrepancies between the glacier surface in 2008 and in 2014. Data could thus be projected onto a homogenous surface (i.e. flattened and without abrupt changes/artefacts). In the case of georectifying meltwater plume extents, data were projected onto a horizontal DEM at sea level. Each camera (and focal length) was calibrated using the camera calibration functions in the Matlab Computer Vision Systems Toolbox to obtain lens distortion parameters and intrinsic camera matrices.

### 4.1.1 Supraglacial lake levels

Three groups of supraglacial lakes were monitored by our time-lapse systems during the 2014 melt season at Kronebreen. Two of these groups were visible from Site 5 on Garwoodtoppen, whilst the other was captured from Site 3 on Colletthøgda (Figure 1). Changes in lake surface area were used as a proxy for water storage on the glacier surface and its release into the englacial/subglacial environment. These lakes were automatically detected from images based on the high contrast in pixel intensity between the ice and water surface. This process involved multiple steps to reduce the change of misidentification: (i) the images were masked to reduce processing time; (ii) the images were enhanced to improve the detection of 'lake-like' objects; and (iii) these objects were manually verified in PyTrx to filter out falsely detected lakes such as shadows.

Each group of lakes was detected from images acquired every half-hour to: (i) isolate the effects of changes in illumination, which influence apparent lake surface area; (ii) match the temporal resolution in which other subglacial components are reconstructed in this study; and (iii) overcome the limited temporal resolution associated with previous satellite- and photography-based analysis in monitoring lake extent. The lakes were easiest to detect when the contrast between the ice surface and water was largest; hence it was difficult to detect the lakes at the beginning and end of the melt season when the lake surfaces

were snow-covered or frozen. In such instances, lake recognition was based on operator identification based on surface colour, homogeneity, and texture.

### 4.1.2 Visible meltwater plume extent

Activity from four surfacing plumes was captured from the time-lapse camera located at Site 1 (Figure 1), on the north side of
the terminus of Kronebreen. Surface areas were calculated for the three plumes on the north side of the terminus. It is assumed that plume surface area is a measure of meltwater discharge from the glacier. Although meltwater plumes can reach neutral buoyancy at depth, this is considered unlikely at Kronebreen due to its shallow depth ($\sim$80 m), weak stratification, and simple thermal, salinity and density structure (Cottier et al., 2005).

Plumes were consistently identifiable based on a combination of water colour, fjord water roughness, and the area from
which icebergs have been cleared by divergent flow. These characteristics are difficult to define automatically due to variation in illumination. Therefore the plume surface area was defined manually within the plane of each image and then georectified to obtain the surface area of each plume. Plume surface area was digitised from images every hour to capture the commonly rapid variability of surfacing plume extent. In some cases, plume extent was larger than the time-lapse image field of view. Such cases are noted in the subsequent results. For the plume on the south side of the terminus, it was hard to measure surface
area accurately due to its distance from the camera. Therefore we simply report its presence or absence.

### 4.2 Tidal level

Tidal measurements were obtained from a tidal gauge in Ny Ålesund, for which all data is hosted online by the Norwegian Mapping Authority (karverket.no). Measurements were made every two hours, and a 12-step moving average was calculated from this to evaluate longer-term trends. As the study area is within the same fjord as the tidal gauge, and located only 12 km
away, it is assumed that these measurements adequately represent tidal level at the glacier terminus.

### 4.3 Melt modelling

A distributed energy balance model (based on Klok and Oerlemans, 2002) coupled with a snow model (SOMARS, developed by Greuell and Konzelmann, 1994) was used to compute melt production and runoff for the 2014 melt season. The distributed energy balance model calculates meltwater production at the surface, which is then used as an input for the subsurface model.
The subsurface model simulates the subsurface evolution of temperature, density and water content; all of which are strongly affected by the storage and refreezing of meltwater (Van Pelt et al., 2012, 2016). Climate forcing at sea level is derived from the Ny Ålesund weather station (Norwegian Meteorological Institute; eklima.met.no). Lapse rates for precipitation (0.13% m$^{-1}$) and temperature (-0.0046 K m$^{-1}$) are used, which provide the best match between the modelled and observed winter and summer balance since 2003 (Van Pelt and Kohler, 2015). A 30-year model spin-up assured initialised subsurface conditions
at the start of the simulation in April 2014. Further details about the model, including model validation and calibration, are outlined in detail in Van Pelt and Kohler (2015).

The model calculates melt and runoff at an hourly resolution. Here, melt is defined as melt production at the surface whereas runoff is melt production and precipitation at the surface which subsequently enters the englacial system. Runoff is assumed to arrive at the glacier front without delay. Spatially-averaged melt and runoff was calculated for the glacier tongue (i.e. not including Holtedahlfonna and the upper part of the glacier catchment) based on elevation bands, with the glacier tongue defined as 0 to 500 m a.s.l. This was undertaken in order to isolate the hydrology of the glacier tongue from hydrological influence in the upper catchment area (i.e. Holtedahlfonna), and better observe direct hydrological effects in the region of interest.

## 4.4 Surface velocities

Glacier surface velocities were calculated from 11-day repeat, 2 m spatial resolution, TerraSAR-X Synthetic Aperture Radar (SAR) images. SAR images are advantageous over optical imagery because they are unaffected by weather conditions (e.g. cloud cover), polar nights, or differences in illumination.

Feature tracking was applied to image pairs using a $200 \times 200$ pixel correlation window ($400 \times 400$ m). These displacements were then orthorectified, resulting in a pixel size of 40 m. Uncertainties are estimated to be <0.4 m per day, which results from a co-registration error ($\pm$ 0.2 pixels) and smoothing of the velocity field over the tracking window (Luckman et al., 2015). Velocity maps were produced for sequential image pairs every 11 days, producing a record of velocity patterns through the 2014 melt season. Point values from these velocity maps were used to calculate spatially-averaged velocities for the glacier centreline, the location of the supraglacial lakes and the borehole site. In addition, absolute velocity differences were determined by tracking between subsequent images from 04 June 2014, from which velocity change maps were produced.

## 4.5 Borehole measurements

Two wireless pressure sensors were placed at the glacier bed in the upglacier section of the glacier tongue during September 2013 (78.8719°N, 12.7957°E, location shown in Figure 1). At this location, the bed elevation is -115 m a.s.l. and the ice surface elevation is 205 m a.s.l., giving an ice thickness of 320 m ($\pm$ 15 m), which is inferred from the borehole length and surface elevation. The sensors were installed with hot-water drilling and both were placed in the same borehole, one 0.2 m above the bed and the other ~2.5 m above the first. The sensors were developed at IMAU (Institute for Marine and Atmospheric Research, Utrecht University) and logged in-situ pressure, temperature and tilt every two hours, which was relayed through a transmitter at the glacier surface for remote access. More details about the specifications of these wireless sensors are presented in Smeets et al. (2012).

A Topcon Net-G3A GPS unit was installed at the position of the transmitter to track the approximate movement of the sensors. It was decided to use the surface velocities derived from TerraSAR-X images rather than the GPS because the GPS velocity record was incomplete and the higher temporal resolution of the GPS data did not add any further insights to this study. The GPS data appeared noisy due to difficulties in processing the positions.

Subglacial water pressure was derived from the difference between the sensor reading and atmospheric pressure, which was obtained from the Norwegian Meteorological Institute weather station at Ny Ålesund (data freely available at eklima.met.no). The sensor directly in contact with the glacier bed collected data between 16th September 2013 and 25 April 2014 before

it stopped recording. The upper sensor continued beyond this date, collecting data for 14 months in total (16/09/2013–03/12/2014). Both sensors exhibited abnormal temperature and tilt readings before they went offline, suggesting probe failure from gradual mechanical failure, inferred to be between the probe and the glacier bed over some days.

## 4.6  Hydraulic potential modelling

Routing of subglacial water was calculated based on the assumption that meltwater flow is governed by gradients in hydraulic potential (Shreve, 1972). Subglacial hydraulic potential ($\Phi$) was calculated according to the approach previously used by Rippin et al. (2003) and Willis et al. (2012):

$$\Phi = k\rho_i g(h - z) + \rho_w gz \tag{1}$$

Where $k$ is the cryostatic pressure factor, $\rho_i$ is the density of ice (917 kg m$^{-3}$), $g$ is acceleration due to gravity (9.81 m s$^{-2}$) ,
$h$ and $z$ are the elevations of the ice surface and bed, respectively (with the difference between them defining the ice thickness) and $\rho_w$ is the density of water (1000 kg m$^{-3}$). The cryostatic pressure factor is effectively the ratio of water-pressure to ice overburden pressure ($P_w/P_i$) and accounts for the possibility that water exists in low-pressure channels (Evatt et al., 2006). Variations in the value of $k$ reflect the degree to which subglacial drainage is pressurised, with $k = 1$ reflecting pressurised flow driven by the influence of gravity on both the overlying ice and the meltwater itself and $k = 0$ reflecting open channel flow
driven only by the influence of gravity. Hydraulic potential gradients change as a consequence of variations in $k$, leading to changes in the simulated subglacial drainage configuration. This allows us to explore the range of drainage paths that can be present.

Surface and bed topography digital elevation models were obtained from a series of radar (low-frequency common-offset radio-echo sounding) surveys which were conducted in 2009–2010 and 2014–2016. The spatial resolution of these two DEMs
is 50 × 50 m, with a vertical accuracy of ± 15 m. The bed DEM was generated by interpolating the measured ice thickness and subtracting it from the surface DEM using the technique referred to in Lindbäck et al. (2014).

## 5  Results

### 5.1  Supraglacial lake area

Three clusters of supraglacial lakes were detected in the time-lapse imagery (shown as C1, C2 and C3 in Figure 1). Changes
in lake surface area are shown in Figure 2A. Cluster 1 is located close to the glacier's north margin (78.8785°N, 12.7063°E). Cluster 2 is located farther upglacier (78.8814°N, 12.7420°E), also near to the north margin. Cluster 3 is adjacent to Cluster 2 (78.8715°N, 12.7493°E), but nearer to the glacier's central flow line. All three groups of lakes occupy crevasses. The lakes in Cluster 1 overspill and coalesce prior to drainage, and occasionally become brown in colour. The lakes in Clusters 2 and 3 are much smaller as they remain confined to crevasses through the melt season and do not coalesce. Their drainage is gradual
(with Cluster 2 draining from 05/07/2014 05:30 and Cluster 3 draining from 16/07/2014 12:30) and they do not drain entirely, with the remaining water gradually re-freezing over time. The colour of these lakes remains blue through the melt season.

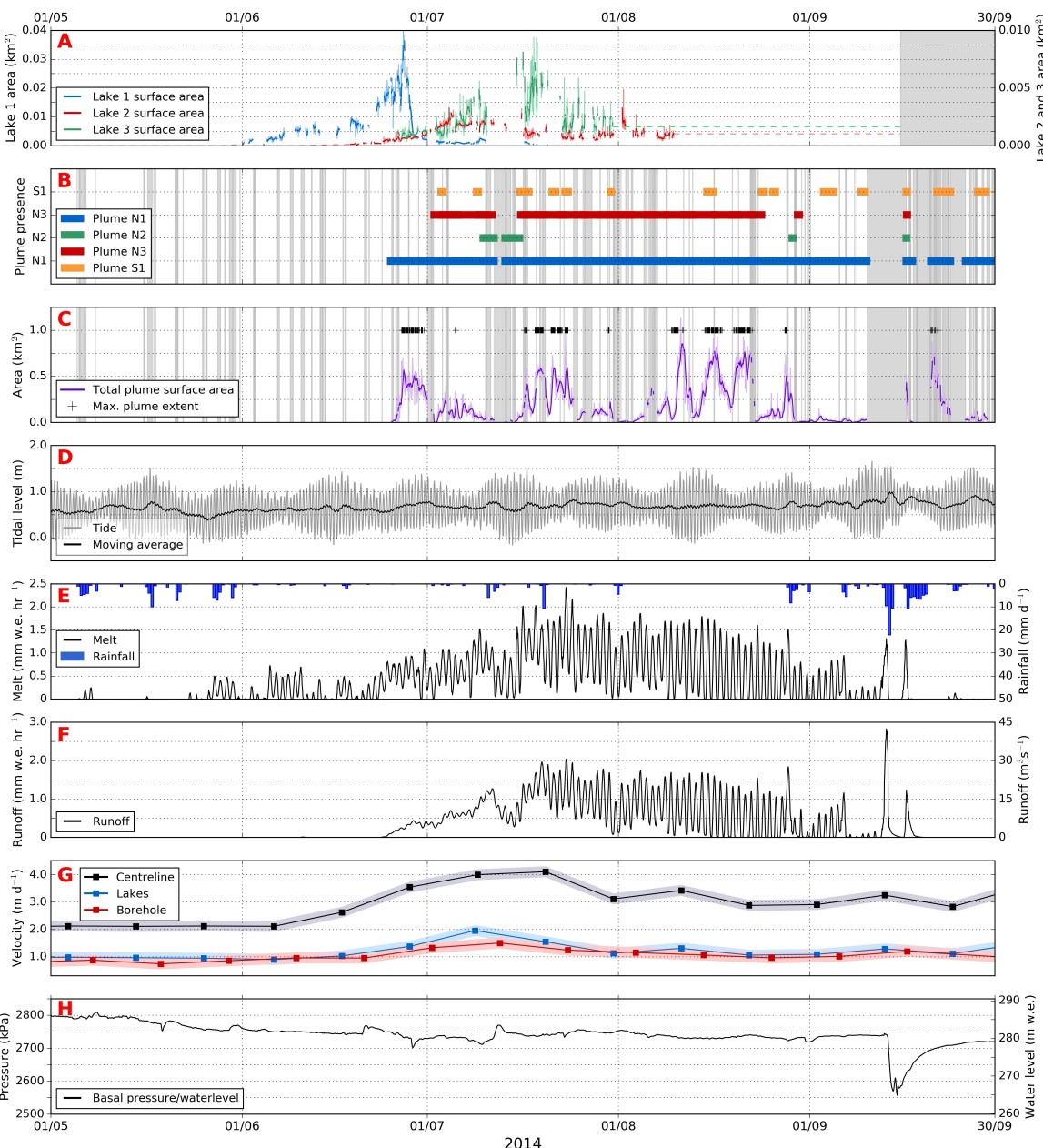

**Figure 2.** Hydrological results from Kronebreen. A) Surface area of the three visible lake clusters (moving averages included); B) Timeline of the appearance of the four plumes, three visible at the north side of the terminus (N1, N2, N3) and one visible from the south side (S1); C) Total surface area of Plume N1, N2 and N3 (moving averages included), plus episodes when the plume extent is out of the image frame (noted as 'max. plume extent'); D) Tidal level (moving averages included); E) Modelled melt (0–500 m elevation) and precipitation; F) Modelled runoff (0–500 m elevation); G) Glacier surface velocities, with spatial averages from the glacier centreline (<2 km from the terminus), the region of the supraglacial lakes, and the location of the borehole site. The faint area around each velocity line is the uncertainty range (<0.4 m/day); H) Water-pressure and corresponding water level from the borehole site.

While the lake clusters appear to act independently, the lakes within Cluster 1 fill and drain almost simultaneously, indicating that they are hydrologically linked. A timeline of changes in lake surface area at Cluster 1 is shown in Figure 3. Cluster 1 fills and drains first, beginning to fill from 01/06/2014 07:00 (Figure 3A–D) and initially draining on 27/06/2014 03:00 over 59 hours (Figure 3E–F), decreasing from a total surface area of 41,374 m$^2$ to 2,477 m$^2$ (see Lake 1 group surface area in Figure 2A). The lakes gradually drain after this, leaving them empty by 21/07/2014 14:00 (Figure 3G–J). The drainage of lakes within this group propagates up glacier, with a 13-hour lag between changes in the lower and upper lakes. This upglacier-propagating drainage is also evident at the upper marginal lakes (Cluster 2) and the upper central lakes (Cluster 3).

## 5.2 Meltwater plume extent

During the 2014 melt season, three surfacing plumes were visible on the north side of Kronebreen and one on the south side (Figure 4). The main, central plume in the north (N1) is the most persistent and largest. The two secondary north plumes (N2 and N3) surface intermittently either side of N1, with N2 to the south and N3 near to the north shoreline. The south plume, S1, surfaces for brief periods. These four plumes were monitored throughout the melt season (Figure 2B). Plume N1 first surfaces at 02:00 on 25 June, approximately 36 hours after the first runoff of the melt season begins, and 84 hours before Lake Cluster 1 fill enough that water is visible in the time-lapse imagery. Plume N3 activates a week later (02 July at 03:00) and is active throughout the monitoring period except for three periods of reduced runoff. Plume N2 is more intermittent, only surfacing for three short periods (10 July at 00:00 – 15 July at 23:00, 29 August at 04:00–22:00 and 16 September at 15:00–17:00), all of which coincide with periods of high runoff and substantial precipitation. Plume S1 is visible on thirteen separate occasions, and is quick to appear and disappear throughout the melt season.

Plume surface area is calculated as the combined surface area of the three plumes on the north side of the terminus when they are active (Figure 2C). Plume S1 could not be included in this total because the coverage of the time-lapse camera was inadequate for distinguishing a precise surface expression. Throughout the melt season, there are three distinct periods when total plume surface area is relatively large and variable (25 June – 08 July, 16–24 July and 08–29 August), and three when the surface area is smaller and relatively constant (08–16 July, 24 July – 08 August and 29 August – 10 September). Plume extent was difficult to distinguish during periods of high rainfall, especially during the highest rainfall event in mid-September when the vast majority of images were obscured.

## 5.3 Tidal level

Tidal level measurements from the nearby tidal gauge in Ny Ålesund show spring and neap tidal phases, with amplified and reduced tidal ranges visible throughout the monitoring period (as shown in Figure 2D). The maximum high tide was ∼1.6 m in mid-September, and the maximum low tide was approximately −0.1 m (i.e. below base level) in mid-July.

The 12-step moving average in Figure 2D shows that longer-term trends in tidal cycles are variable, with an average tidal level of 0.4–1.0 m throughout the season. Tidal cycles are fairly consistent between May and August, with an average tidal level of 0.5–0.8 m. Higher variability is visible in September, with the average tidal level fluctuating over short periods (particularly in mid-September), which coincides with a large precipitation event that is displayed in Figure 2E.

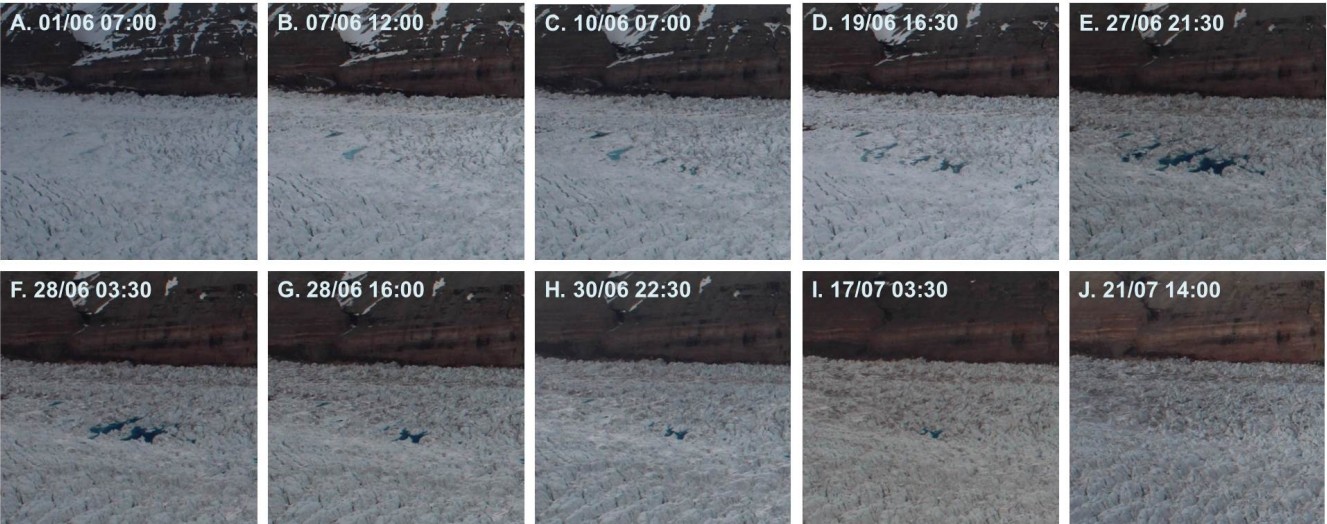

**Figure 3.** Selected time-lapse imagery showing the filling and drainage of the individual lakes in Lake Cluster 1. The glacier flows from right to left. A) Frozen water presides in crevasses; B) Frozen water thaws and lakes gradually fill; C) Upglacier crevasses begin to fill with water; D) All lakes continue to fill simultaneously; E) Lakes at maximum surface area; F) Downglacier lakes drain and no remaining water is visible from the given camera angle; G) Upglacier lakes partially drain and some remaining water is visible; H and I) Upglacier lakes continue to drain gradually; J) No remaining water is visible in any of the lakes by this point.

## 5.4 Melt and runoff

Spatially-averaged melt and runoff was calculated for the lower catchment of Kronebreen, from 0 to 500 m a.s.l. which covers all of the glacier tongue (Figure 2E and 2F). Surface melt production begins on the 26 May, approximately one month before the onset of runoff is predicted by the model. The highest melt production and the highest diurnal variation in melt production occur in mid-July, with 1.5–2.5 mm w.e. hr$^{-1}$ during the day and 0.25–0.9 mm w.e. hr$^{-1}$ during the night. This diurnal signal persists throughout the record until mid-September when two large precipitation events on the 13 and 16 September appear to dominate and overprint the diurnal pattern.

The model predicts water retention in snow until 08 June. Runoff initially has very low values (0–0.1 m$^3$ s$^{-1}$) and increases markedly from 23 June, coinciding with the drainage of Lake Cluster 1 and the activation of the meltwater plumes. From this point, melt and runoff regularly reaches 20–26 m$^3$ s$^{-1}$ in the day and between 0–3 m$^3$ s$^{-1}$ at night. Towards the end of August, melt and runoff are consistently negligible during the night. Thereafter, melt and runoff steadily decline through September and are very low from 07–13 September, although they spike abruptly on two occasions: a first event where runoff peaks at 44.6 m$^3$ s$^{-1}$ on 13 September and a second where runoff peaks at 19.5 m$^3$ s$^{-1}$ on 16 September. These instances coincide with two periods of high rainfall and large variations in tidal level. The first of these instances is the largest recorded precipitation event in that year.

## 5.5 Glacier surface velocity

From the TerraSAR-X velocity dataset, spatially averaged velocities were calculated to compare with components of the glacier's hydrology system. These were demarcated for three regions of interest (ROI's): (i) the near-terminus (0–2 km) centreline, (ii) the area of supraglacial lakes (3 km from the terminus) and (iii) the area of borehole study (5 km from the terminus) (Figure 2G). In addition, velocity maps were created to depict spatial patterns in velocities and absolute velocity differences since the beginning of the melt season (04 June 2014) to investigate spatial patterns in surface speed-up and slow-down events. Six maps are presented in Figure 5, presenting three velocity maps derived between image pairs and three absolute velocity difference maps.

Surface velocities over the lower portion of the glacier tongue are $\sim$1.2 m d$^{-1}$ throughout May, with higher velocities ($>$2.5 m d$^{-1}$) on the south side of the terminus and lower velocities ($<$1.5 m d$^{-1}$) at the glacier margins on account of lateral drag. The near-terminus velocity is the highest of the three ROI's, fluctuating between 2.0 and 4.0 m d$^{-1}$ over the course of the melt season. Velocities from the supraglacial lake areas and the borehole site range between 1.0 and 2.0 m d$^{-1}$.

A speed-up occurs at the beginning of the season from mid-June to the beginning of July. Velocity maps from this period are presented in Figure 5. The region of high velocities at the terminus gradually propagates upglacier through June as the rate of melt production increases. This area of high velocities ($>$2.5 m d$^{-1}$) is largely confined to the south region of the glacier tongue. Absolute velocity differences (Figure 5) show that the largest accelerations occur at the terminus, with a difference of $\sim$2 m d$^{-1}$ (since 04 June 2014). Acceleration generally decreases with distance from the terminus, with the exception of the north region of the terminus which generally experiences smaller accelerations ($\sim$ 1.5 m d $^{-1}$). This speed-up coincides with the drainage of the supraglacial lakes in Cluster 1 on 27 June 03:00, the activation of the meltwater plume, and the marked increase in runoff shown in Figure 2G.

Figure 5 shows that the area of high velocities ($>$2.4 m d$^{-1}$) is largest between 7 and 18 July, and that this encompasses most of the south and central regions of the near-terminus area. Velocities remain consistent until the end of August, with velocities in the terminus zone around 3.0 m d$^{-1}$ and velocities around the supraglacial lakes and the borehole site between 1.0–1.5 m d$^{-1}$ (Figure 2G). Velocities begin to subside from 25 July, with velocity patterns resuming to pre-melt season values by 16 August.

A second speed-up event is recorded in September, possibly caused by the two high rainfall events on the 13 and 16 September. While velocities remain constant at the lake and borehole ROI's through the rest of September, high velocities persist at the centreline ROI as shown in Figure 2G. These high velocities do not return to pre-melt season conditions for the remainder of our measurement period.

## 5.6 Borehole pressure

Upon reaching the glacier bed at a depth of -115 m a.s.l. when drilling, the water level in the borehole dropped abruptly, indicating an efficient means of drainage is present. Comparison of the water-pressures recorded by the two pressure sensors reveals very high correlation (R = 0.999) and a mean offset of 24.3 kPa, agreeing with the $\sim$2.5 m difference in installation depth.

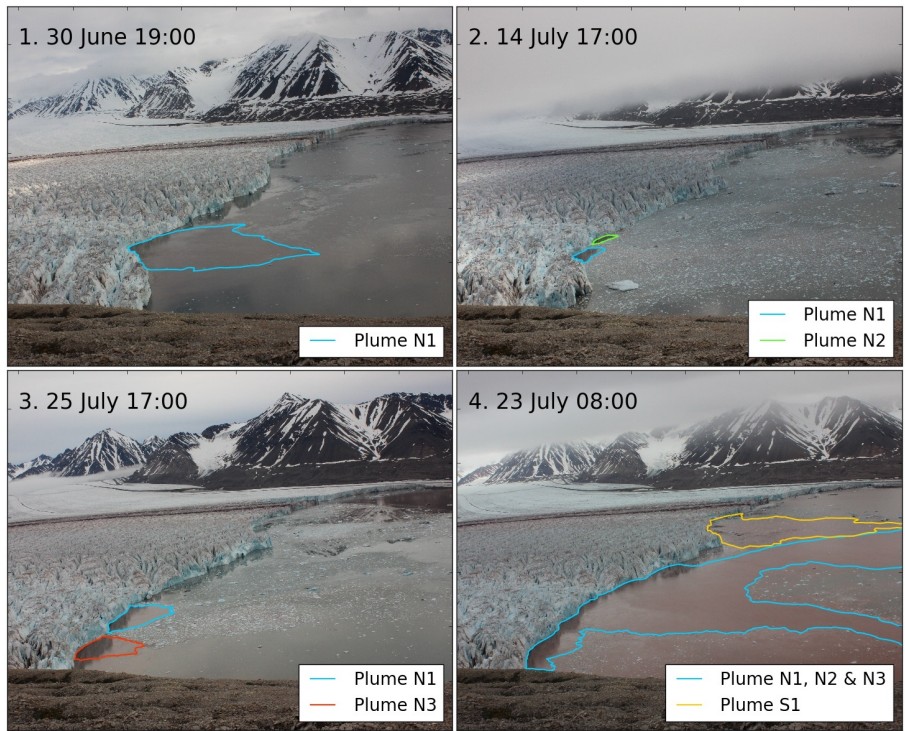

**Figure 4.** Meltwater plume scenarios from time-lapse imagery at Kronebreen. Top-left to bottom-right: 1) Surfacing meltwater plume from the main source on the north side of the glacier terminus, N1; 2) Sources from Plume N1 and Plume N2; 3) Sources from Plume N1 and Plume N3; 4) Plume N1 and Plume S1, the main source on the south side of the glacier terminus.

This close correspondence throughout the period over which both sensors were operating gives us confidence in assuming that subglacial water-pressure continues to be recorded by the upper sensor after failure of the lower sensor, providing a continuous 14-month record of subglacial water-pressure. Figure 2H shows a subset of the entire measurement period (May – September 2014).

5     The mean water-pressure from the beginning of May until 13 September was 2750 kPa. This equates to a water level of 280 m, which is close to the floatation level (291–293 m) based on a local ice thickness of 320 m and an ice density between 910–917 kg m$^{-3}$. The water level fluctuates over a relatively small range of 11 m in this part of the record. A marked fluctuation occurs on 13–14 September, involving a substantial drop of 17 m over a period of 24 hours, followed by a week-long recovery. This coincides with the largest precipitation event of the season (43.6 mm in a 24-hour period), which prompted high runoff

10 after a period of very little surface runoff.

    The record is also characterised by several minor, but rapid, pressure changes, most notably during the three events at the beginning of July: 1) An increase of 3 m occurred over a 14 hour period from 20 June at 10:00; 2) a 3 m drop occurred over a 12 hour period from 28 June at 04:00 followed by a subsequent recovery; and 3) an increase of 6 m occurred over a 64 hour

period from 09 July at 20:00. These three events coincide with, respectively, 1) initiation of notable runoff, 2) drainage of the largest set of supraglacial lakes (Figure 2A), and 3) activation of the main meltwater plume (P1) (Figure 2B).

## 5.7 Hydraulic potential

Several scenarios were considered in calculating the hydraulic potential at the bed of Kronebreen based on the $k$ value, which represents cryostatic pressure ratio (i.e. the extent to which meltwater routing is dictated by ice-pressure gradients). Subglacial hydraulic potential was calculated over several iterations, changing the value of $k$ each time. In total, we ran 11 simulations with the value of $k$ increasing incrementally by 0.1 (i.e. hydraulic potential was calculated each time with a $k$ value of 0.0, 0.1, 0.2, 0.3, 0.4, 0.5, 0.6, 0.7, 0.8, 0.9, and 1.0).

Results suggest that subglacial meltwater is routed along the north sector of the glacier when it is largely controlled by ice-pressure gradients ($k > 0.6$), and meltwater is channelled to the south region when bed topography is the greater control ($k < 0.6$). Flow routing changes between a cryostatic pressure ratio of 0.5 and 0.6, with anything less than, or greater than, this value having little effect on the overall drainage configuration. A scenario where hydraulic potential is dictated by ice-pressure gradients (i.e. a $k$ value between 0.6 and 1.0) is more realistic because the borehole record shows that water at the bed is persistently pressurised. The locations of the bed pressure sensor, the supraglacial lakes and the meltwater plumes on the north side of the terminus are hydraulically linked in this scenario (Figure 6). This being the case, it is probable these are connected throughout the melt season and that simultaneous changes are indicative of the hydraulic regime of the subglacial environment.

## 6 Interpretation

The data sets presented above – supraglacial lake area, plume visibility and extent, modelled melt and runoff, surface velocity, and borehole water-pressure – are signals of the subglacial drainage system. The relative timing of variations in these components can be used to construct a conceptual model to explain the storage and release of subglacial meltwater at Kronebreen. Additional insights into subglacial flow routing are obtained from the modelled hydraulic potential to support the ideas in this model.

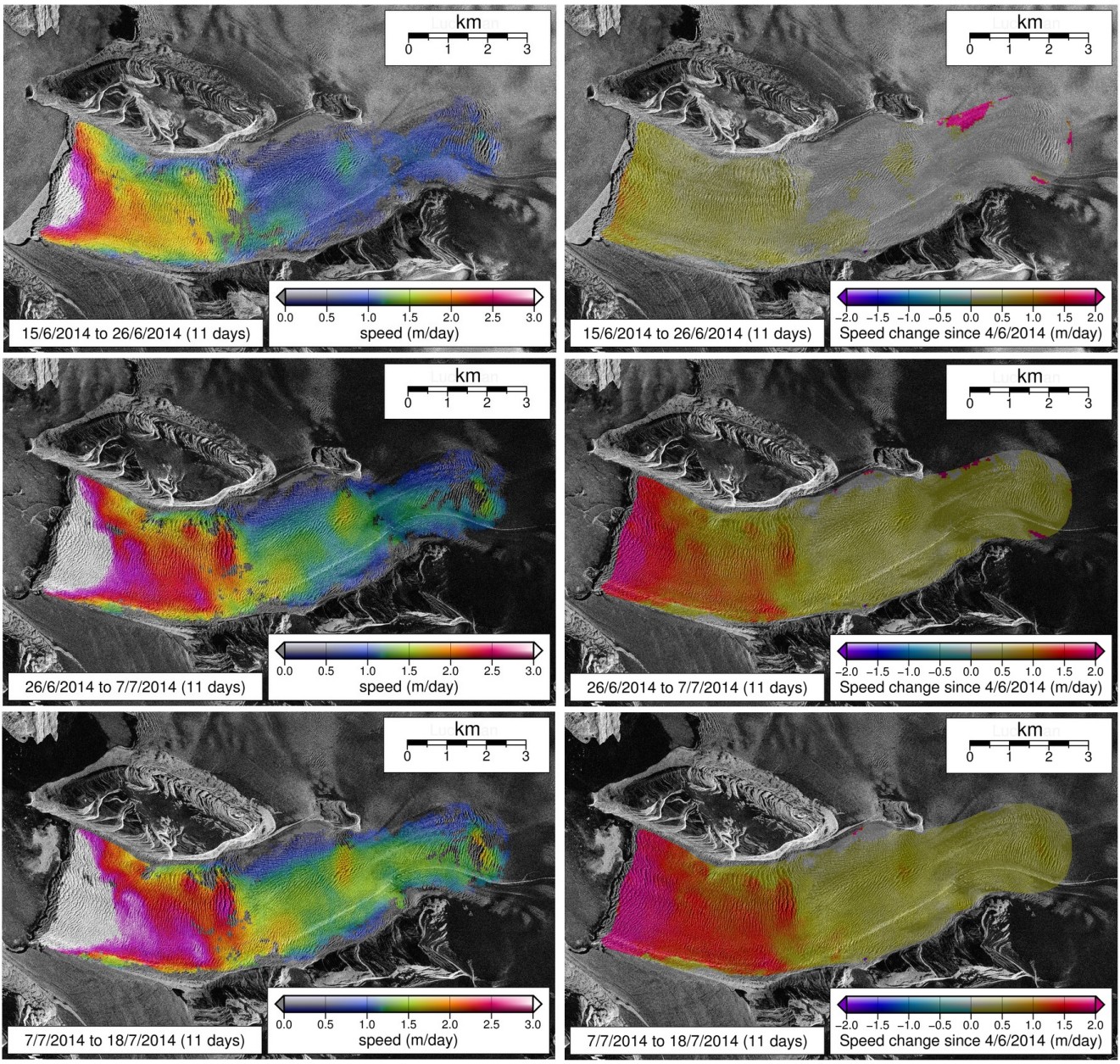

**Figure 5.** Sequential velocity maps from Kronebreen showing velocity from image-to-image (left) and absolute velocity difference since 04 June 2014 (right). These velocities are calculated from feature tracking through TerraSAR-X imagery, spanning 15/06/2014 to 18/07/2014. Maps show velocity differences between the north and south regions of the glacier tongue, with the highest velocities associated with the south region. There is also a distinct boundary in the velocity field approx. 3 km up the glacier tongue. This boundary is due to a high in the bed topography. The seasonal speed-up is generally consistent across the entire front, as shown in the maps depicting absolute velocity difference.

## 6.1 Beginning of the melt season (May – June)

A series of key events occurs at the beginning of the 2014 melt season (01 May - 30 June):

1. Melt production commences, increasing from ∼0.25 mm w.e. hr$^{-1}$ in the latter part of May, to 1 mm w.e. hr$^{-1}$ by the end of June (Figure 2E).

2. The supraglacial lakes in Cluster 1 fill from 01–27 June (Figure 2A).

3. Surface velocities increase uniformly across the lower part of the glacier tongue while the lakes in Cluster 1 fill, notably at the centreline from 2 to 3.5 m d$^{-1}$ (Figure 2G).

4. Runoff increases (to >0.1 m$^3$ s$^{-1}$) from 23 June (Figure 2F).

5. The dominant meltwater plume on the north side of the terminus (N1) surfaces in the fjord at 02:00 on 25 June (Figure 2B and 2C).

6. The supraglacial lakes in Cluster 1 drain from 03:00 on 27 June over a period of 59 hours, decreasing from a total surface area of 41,374 m$^2$ to 2,477 m$^2$ (Figure 2A). They drain in an upglacier-propagating fashion (Figure 3).

7. The water level in the borehole drops by 3 m over a 12-hour period from 28 June (04:00), followed by a subsequent recovery.

8. Surface velocities continue to increase into July, with continued uniform ice acceleration across the glacier tongue. The highest velocities are experienced in the central/south region of the glacier tongue (Figure 5). In addition, a second meltwater plume becomes active on the north side of the terminus (N3) and a plume intermittently surfaces on the south side (S1) at the beginning of July (Figure 2C).

The surface velocity of the glacier begins to gradually increase from 10 June, based on the velocities from the ROI's – the centreline, the region of the supraglacial lakes, and the borehole site (Figure 2G). The nature of this speed-up is similar to those observed by Howat et al. (2005) and modelled by Nick et al. (2009) at Helheim glacier, with acceleration occurring in an upglacier-propagating fashion. They attribute this to changing boundary conditions at the glacier terminus. Luckman et al. (2015) observed a marked increase in calving retreat at the front of Kronebreen at the beginning of the 2014 melt season, which precedes this early-season ice acceleration. It is likely that the observed change in conditions at the glacier terminus is linked to the changes in surface velocity. Specifically, the increase in calving rate could have reduced back-stress farther upglacier and enabled enhanced glacier flow (Nick et al., 2009).

Another likely influence is the presence of meltwater at the bed, which enhances basal lubrication and enables sliding. This has been highlighted as a key process at Kronebreen in previous years (Schellenberger et al., 2015), and could also be the case for the 2014 melt season. The coincident observations of the filling of the supraglacial lakes suggest that the subglacial system is gradually filling with meltwater, assuming that these lakes are connected to the bed and thus reflect hydraulic head.

However, the modelled runoff does not indicate this, predicting that meltwater only reaches the bed from 23 June (Figure 2F). This implies that water is either being generated at the bed, or that surface meltwater is bypassing storage in the snowpack and firn layer. Basal frictional melting could play a role in the generation of meltwater at the bed, but modelling of Kronebreen's basal properties suggest that surface runoff is more likely to be the key influencing factor (Vallot et al., Accepted). Surface

meltwater may have originated from higher elevations, but it is unlikely given that early-season melt production is understood to first originate from the lower elevations of this glacier catchment (Van Pelt and Kohler, 2015). Observations from the time-lapse images show that bare ice is visible from mid-June, after a small rainfall event on the 17 June. Also, the lower area of the glacier tongue is heavily crevassed, providing abundant meltwater pathways to the glacier bed. It is therefore likely that some surface meltwater is bypassing storage in the snowpack earlier than the model predicts, and the model under-represents

pathways from the surface to the bed.

    The continuous presence of a plume at the north side of the terminus (N1) indicates that a channel is established here from 25 June (Figure 2C). Two additional plumes (N2 and N3) surface in the fjord later in the season. The modelled hydraulic potential indicates that a channelised system may be present at the north side of the terminus (Figure 6). The location of the main outlet of this channelised drainage matches the location of the three plumes, further suggesting that these plumes are an

outflow of a channelised drainage system. Hydraulic potential is more likely to be governed by ice-pressure gradients than bed topography. In this scenario, channels in the north region of the glacier tongue drain a significant area of the glacier catchment, with channels connected to the upper ice field (Holtedahlfonna). It is therefore likely that the plumes on the north side of the catchment represent a large proportion of the glacier's subglacial outflow.

    The supraglacial lakes in Cluster 1 drain from 27 June (Figure 2A), which occurs after the activation of the main plume on

the north side of the terminus on 25 June. The downglacier lakes empty first by 28 June (03:00), the lakes in the middle of the cluster empty by 28 June (16:00), leaving the upglacier lakes partially drained by 30 June (22:30) and these eventually drain completely by 21 July (14:00) (Figure 3). The relative timing of these events indicates that these components are hydraulically linked. These events also coincide with a 3 m drop (over a 12-hour period) in the water level at the borehole site from 28 June (Figure 2H). It is uncertain whether the borehole is also hydraulically linked to these components. The water level in the

borehole dropped when it first made contact with the bed, indicating initial local efficient drainage. However, the water-pressure in the borehole remains close to ice overburden pressure throughout the melt season, suggesting that either a connecting channel is consistently full of meltwater, or the borehole is located in a region of the bed that has inefficient drainage. Therefore it is likely that a local subglacial channel is present in this region, but this is not necessarily connected to a broad subglacial drainage network.

Previous observations of lake drainage have been attributed to hydro-fracturing caused by changes in tensile stresses across the glacier (e.g., Everett et al., 2016). Longitudinal stretching may have initiated the activation of the plume and the drainage of the lakes at Kronebreen, and this may be controlled by changes at the terminus (e.g. an increase in calving activity) and/or the observed early-season speed-up. Hydro-fracturing has also been linked to changes in meltwater presence at the bed, which promote drainage via basal slip (e.g., Stevens et al., 2013). A similar scenario at Kronebreen could be an indication of widespread

drainage that occurs in an upglacier-propagating pattern (i.e. an early-season 'flushing event'). This idea is supported by the

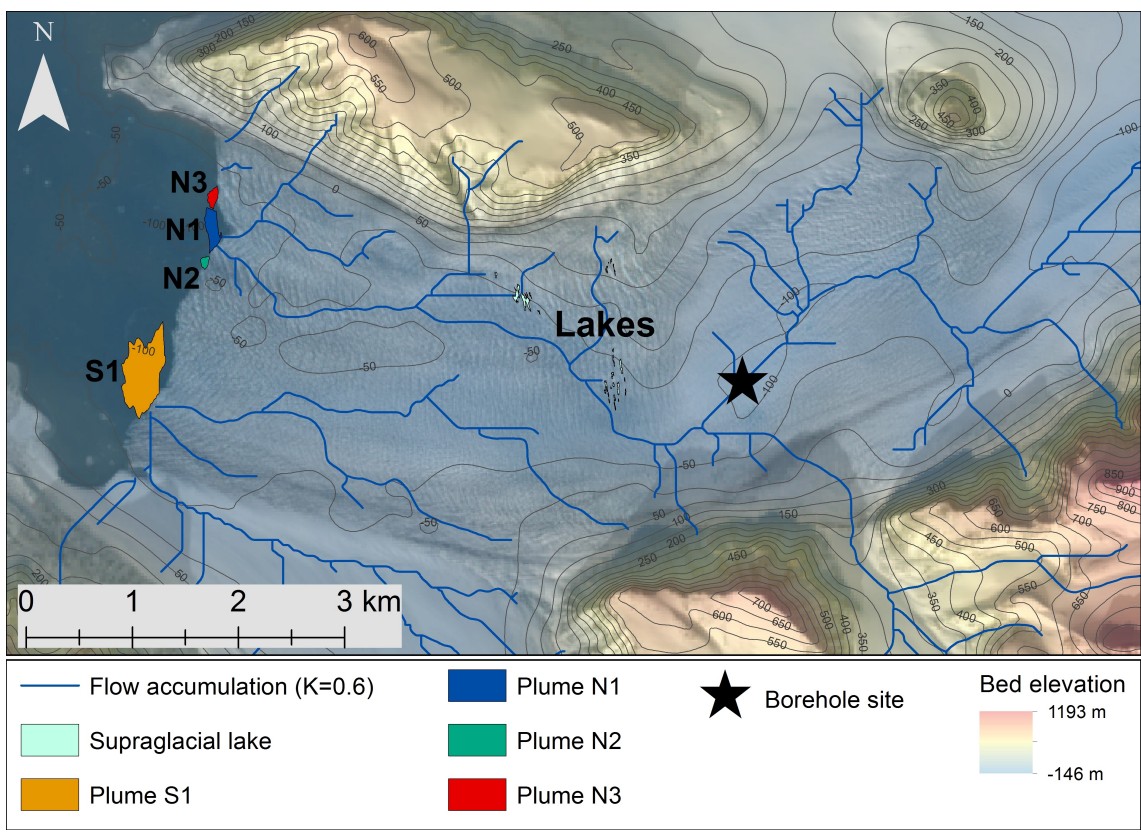

**Figure 6.** Potential subglacial water pathways at Kronebreen, as calculated from a scenario where hydraulic potential is governed by ice-pressure gradients (i.e. the cryostatic pressure ratio, $K$, is above 0.6). The surface expressions for Plume N1 and N2 are taken from 11 July 08:00, Plume N3 is taken from 20 July 2014 10:00, and Plume S1 is taken from 16 September 04:00. The expression of Plume S1 is, on average, smaller than the expression shown here. This expression was chosen because it is the most accurate shape of the surface expression that could be acquired during the monitoring period. The base map is a Landsat image (taken on 11 June 2014) overlaid with bed elevation and corresponding contours at 50 m intervals.

modelled hydraulic potential, which suggests that the north plume outlet, the supraglacial lakes and the borehole could be linked via a common channelised system (where hydraulic potential is governed by ice-pressure gradients) (Figure 6).

## 6.2 Middle of the melt season (July – August)

July and August are distinguished by distinct changes in surface velocities and plume activity. As noted above, surface velocities
5 increase gradually from the beginning of the melt season and this continues through to a peak in mid-July (Figure 2G). This peak coincides with the drainage of the supraglacial lakes in Cluster 2 (05/07/2014 05:30) and Cluster 3 (16/07/2014 12:30) (Figure 2A). The sequential velocity maps show that this speed-up propagates ~3 km upglacier by mid-July (Figure 5). Surface

velocities are highest to the central/south region of the glacier tongue ($>3$ m d$^{-1}$), while some of the north region only reaches velocities of $\sim$2.5 m d$^{-1}$.

Plume activity to the north of the terminus persists throughout August (Figure 2B). The main plume (N1) is always visible, the secondary plume (N3) is present for most of the month (01–20 August), and the third (N2) is briefly active on 29 August. The total surface area of these plumes fluctuates on a regular basis. This behaviour is repeated throughout August (08–28 August), with each fluctuation phase (i.e. a period of expansion followed by a reduction in surface area) lasting 4–5 days (Figure 2C). Changes in surface melt and runoff appear to have little influence on this pulsing. In addition, links between tidal level and plume surface area are very weak (Figure 2D). This implies additional controls on subglacial outflow, possibly cycles of internal, subglacial storage and release. Subglacial hydraulic pulsing has been observed previously at land-terminating glaciers where it was associated with episodic ice motion (Kavanaugh and Clarke, 2001).

Activity from the plume on the south side of the terminus (S1) is intermittent (Figure 2B): the plume surfaces for short phases ($<$62 hours) every 5 days on average. This release of water could either be internally driven or could indicate a dynamic drainage system, which can quickly transition between efficient and distributed configurations. This differs from the persistent plume activity in the north region, and possibly reflects differences in drainage efficiency across the terminus. Modelled hydraulic potentials indicate that it is likely for meltwater to be routed to the north region throughout the melt season (Figure 6). This being the case, meltwater is not efficiently evacuated from the central/south region. Meltwater will be slow-flowing and/or stored at the bed and enhance basal lubrication, providing an explanation for the spatial patterns in velocities reconstructed (Figure 5).

### 6.3   End of the melt season (September)

The end of the 2014 melt season is characterised by five main features:

1. Modelled melt and runoff decrease by the beginning of September and continue to do so till mid-September (Figure 2D and 2E). Additionally, plume extent is consistently small (Figure 2C) and activity is visible from only one of the outlets (N1) on the north side of the terminus (Figure 2B). Intermittent activity is also evident from the plume on the south side of the terminus (S1).

2. A large rainfall event occurs on 13 September, directly influencing runoff and likely also enhancing melt (Figure 2E and 2F). This rainfall event is the largest of the season (43.6 mm in a 24-hour period). This coincides with atypical fluctuations in tidal level (as shown by the moving average in Figure 2D) and a 17 m-drop in water level at the borehole site over a period of 24 hours (Figure 2H). The water-pressure at the bed recovers to previous values by 20 September (i.e. a 7-day return time).

3. A second large rainfall event occurs on 16 September, promoting a second spike in melt and runoff (Figure 2E and 2F). Recovery of the water-pressure in the borehole remains gradual and consistent during this period.

4. Although there is limited visibility of the plumes during these rainfall events, clear conditions from 16 September (15:00) show that all four plumes are active and were possibly active during the storm (Figure 2B). Plumes N2 and N3 stop surfacing by 19:00 on 16 September. The two main outlets on the north and south side of the terminus (N1 and S1) continue to surface for the rest of the month (Figure 2C).

5. High surface velocities continue through September, largely confined to the central/south region of the glacier tongue ($\sim$3 m d$^{-1}$ at the centreline, Figure 5).

It is likely that the presence of meltwater in the subglacial system beneath the north region of the glacier tongue has diminished by the beginning of September. Less water entered and left the system, as indicated by the decreased melt/runoff and the small plume extent on the north side of the terminus respectively (Figure 2C, 2E and 2F). Surface velocities remain high in the central/south region of the glacier tongue though, as shown by the velocity record from the centreline ($\sim$3 m d$^{-1}$, Figure 2G). Plume activity on the south side of the terminus is intermittent. This suggests that meltwater is not being effectively evacuated from the subglacial environment under the central/south region of the glacier tongue. It is likely that this meltwater is slow-moving and/or being stored, which would enhance basal lubrication and is a likely reason for high surface velocities in this region at this late stage in the melt season.

The substantial rainfall event on 13 September appears to re-activate melt and runoff which, in turn, is likely to cause a rapid influx of water to the glacier bed (Figure 2E and 2F). The atypical fluctuations in tidal level further suggest that this rainfall event is associated with a low-pressure weather front and an associated storm surge (Figure 2D). The coincident timing of the large drop in water-pressure at the borehole site indicates that rapid meltwater routing influences the upper area of the glacier tongue (Figure 2H). In addition, all four plume sources were active for at least part of the storm, suggesting that channels were present at the glacier terminus (Figure 2B). These observations support the idea that water was evacuated through a glacier-wide efficient drainage system during this period. This could be evacuated in a similar fashion to the 'flushing event' observed at the beginning of the melt season.

However, high surface velocities persist through the remaining part of September. These high velocities are largely confined to the central/south region of the glacier tongue, similar to the velocity field observed in June/July (Figure 5). This suggests that meltwater is being retained in the subglacial environment despite the presence of an efficient drainage system. It is likely that water is efficiently evacuated from the north region of the glacier tongue, but not from the south/central region. This hypothesis matches the hydraulic potential modelling, which indicates that the majority of subglacial meltwater is routed to the north of the glacier tongue, leaving the south/central region hydraulically isolated from the efficient drainage system (Figure 6).

## 7    Discussion

### 7.1    Early melt season meltwater storage

Surface velocities gradually rise at the beginning of the melt season, from mid-June onwards. As previously noted, it is likely that this early-season speed-up is linked to an increase in calving retreat at the terminus (Luckman et al., 2015). The presence

of meltwater at the bed is also a key component to this speed-up. Early-season melt production is routed to the bed earlier than the runoff model predicts, as it bypasses storage in the snowpack and is routed to the bed via abundant crevasses in the lower area of the glacier tongue (based on observations from the time-lapse images). This is likely to enhance basal lubrication and facilitates sliding and/or subglacial sediment deformation. This meltwater is being delivered to the bed and stored for a significant period of time before it is efficiently evacuated from the subglacial system. The activation of the main plume on the north side of the terminus (N1) suggests that either a sufficient volume of meltwater is being discharged to surface in the fjord, or an efficient system is established to evacuate meltwater on 25 June. We consider the second of these instances to be more likely as the plume was observed to be surfacing from a single source (based on observations from the time-lapse imagery), signifying that it was channel-fed. This being the case, meltwater is stored at the bed for ∼15 days before it is evacuated, based on the timing of the onset of the speed-up and the activation of Plume N1. It is likely that it was released either when sufficient pressure has accumulated to force a channel to open, or when subglacial water has sufficiently melted the cavity/conduit wall. Therefore the storage of water at the bed of the glacier could play a vital role in the seasonal speed-up at Kronebreen during the 2014 melt season.

## 7.2 Upglacier-propagating supraglacial lake drainage

The three groups of supraglacial lakes observed through the 2014 melt season exhibit different filling and draining patterns. The lakes in Cluster 1 overspill and coalesce, and drain rapidly. Water is no longer visible from the view of the time-lapse camera, which suggests that this drainage completely empties all stored water at the surface. The lakes in Clusters 2 and 3 are constrained within individual crevasses as small discontinuous ponds. Drainage of these lakes is rapid, but some water remains at the surface. Danielson and Sharp (2013) identified three types of lake drainage events, distinguished by the rate at which the drainage occurs and the volume of water that is drained: 1) Crevasse pond drainage – a region of unconnected lakes form within crevasses that drain asynchronously, suggesting that the crevasses empty from the base; 2) Slow lake drainage – supraglacial lakes that drain by overflowing, which commonly leaves a remnant lake in the deepest part of the basin; and 3) Fast lake drainage – complete, rapid drainage of a supraglacial lake via a crevasse or moulin opening within the lake basin. The three lake clusters in this study exhibit the characteristics of two of these typologies: Cluster 1 adheres to the characteristics of fast lake drainage (type 3) and the lakes in Clusters 2 and 3 are similar to the characteristics of crevasse pond drainage (type 1).

The lakes in Cluster 1 are of particular interest because of the coincidence of their drainage with changes in surface velocities, runoff, and plume activation at the beginning of the melt season. Lake drainage is linked to longitudinal stretching which occurs in response to a change in glacier dynamics (i.e. ice speed, calving activity), and changes in conditions at the bed which promote enhanced basal sliding (Stevens et al., 2013; Everett et al., 2016). The drainage of the lakes at Kronebreen are likely to be linked to both a change in glacier dynamics and an associated change in bed conditions, in this case an increase in the presence of meltwater. Longitudinal stretching, and consequent crevasse opening, occurs as the glacier accelerates at the beginning of the season, creating more pathways for meltwater to be delivered to the bed. Supraglacial lakes either drain by hydro-fracturing which is promoted by the speed-up, or when they become linked to a common channelised system. Our hydraulic potential modelling supports this as it indicates that Cluster 1 may be located close to a large channel/flow accumulation pathway. Their

drainage indicates that this is an early-season 'flushing event' that occurs in an upglacier progression, as reflected in the timing of their connection to the subglacial environment.

## 7.3 Controls on meltwater plume activity

Three plumes are visible at the north side of the terminus (N1, N2 and N3) during periods of high rainfall, suggesting that more channels become active when there is a rapid input of meltwater to the bed. The location of these plumes matches the location of a major channel outlet in the hydraulic potential model, suggesting that these plumes are the outflow from an efficient drainage system under the north region of the glacier tongue. Observations of increased plume activity during and/or shortly following high-rainfall events suggest that more channels become active on the north side of the terminus to accommodate an abnormally high rate of meltwater delivery to the bed. The rate at which these channels switch on and off (indicated by the short lag between precipitation/runoff and plume activity) indicates that the subglacial environment is highly dynamic and able to adapt rapidly; either dormant channels become active or new channels form to accommodate for high rates of meltwater delivery.

In contrast, one plume is visible at the south region of the terminus (S1). The activity of this plume is intermittent and it is unexpectedly absent during periods of high runoff, suggesting that the outflow of meltwater is not channelised and is instead more distributed at the grounding line. The modelled hydraulic potential indicates that only a small proportion of the total drainage is routed here. It is therefore unlikely that a stable channelised drainage system exists in this region, and a distributed system resides in periods of low discharge. We propose that this plume activity is a signal for subglacial hydraulic pulsing, which represents periodic meltwater flushing. This occurs when sufficient pressure has accumulated to force a channel open, and/or when subglacial water has melted the cavity/conduit walls to allow an increase in discharge.

Few links are observed between plume outflow and tidal level, which suggests that this is an internally-driven process with limited tidal influence. Internally-driven hydraulic pulsing has previously been observed at land-terminating glaciers and associated with abrupt ice motion caused by the gradual failure of 'sticky spots' on the glacier bed (Kavanaugh and Clarke, 2001; Kavanaugh, 2009). This progressive failure transfers basal stress to hydraulically-unconnected regions of the bed and effectively 'squeezes' water through them. This may also be occurring at Kronebreen, although it is difficult to further examine here due to the coarse temporal resolution of the velocity record. If this is the case, hydraulic pulsing could be a major control on subglacial meltwater storage. For example, storage is evident at the beginning of the season when melt production has begun, supraglacial lakes begin to fill, and velocity gradually increases from $\sim 2$ m d$^{-1}$ to $\sim 4$ m d$^{-1}$ (based on velocities from the centreline). The trigger for the release of this water could be related to hydraulic pulsing via mechanical adjustments at the glacier bed.

Plume presence is commonly taken as an expression of the subglacial drainage network near the terminus. For example, Slater et al. (2017) saw no surfacing plume activity in the middle of the summer melt season at Kangiata Nunata Sermia (KNS), Greenland, despite high runoff. They associated this with a distributed drainage system at the bed, producing multiple outlets that did not surface in the fjord. The activity of the plume at KNS is similar to that observed at Plume S1 at Kronebreen, with plume extent disassociated from runoff. Slater et al. (2017) argued that this disassociation may be indicative of a system

that is close to the threshold between a distributed and efficient drainage system. This is likely to also be the case at Kronebreen. It is further suggested here that plume activity can be used as a signal for subglacial hydraulic pulsing, specifically the internal storage and release of meltwater at marine-terminating glaciers.

Satellite imagery with long repeat-pass times is unlikely to adequately represent plume activity, even in long-term studies. Plume extent is controlled by multiple processes acting on different timescales and associating them with glacier hydrology and/or dynamics for a discrete point in time may be misleading. Time-lapse photogrammetry has proved vital here in providing high-frequency records of meltwater plume activity. However, plume activity could not be monitored through storms and cloudy conditions using this technique. Plume activity notably changes during storms and valuable information can be extracted about plume dynamics during these periods. This is a limiting factor in time-lapse photogrammetry and alternatives need to be implemented to overcome this pivotal limitation.

## 7.4 Subglacial drainage of Kronebreen

At the borehole site, there is little or no diurnal signal in the water-pressure record and the subglacial system is consistently close to ice overburden. The water-pressure record at Kronebreen reflects a high hydraulic base-level determined by water depth at the terminus. This ensures that the subglacial environment is persistently pressurised where the bed is significantly below sea level. This permits fast flow, which could preclude the formation of persistent channels.

Few short-term pressure variations are observed in the water-pressure record from May–September 2014, apart from the significant drop in pressure at the end of the melt season. Although the modelled hydraulic potential suggests that the borehole is located within an efficient drainage catchment, it is more likely that the borehole is actually indicative of a region that is inefficiently drained for a large part of the melt season.

Observations of intermittent plume activity at the south side of the terminus suggest that a stable drainage system cannot exist in this region. Meltwater discharge is instead driven by internal hydraulic storage and release. However, the persistent presence of plumes at the north side of the terminus indicates that a channelised system could be active below this part of the glacier for the majority of the melt season. In this area, a stable efficient drainage system is encouraged both by the hydraulic gradient below the glacier, and the relatively low velocity of the ice due to lateral drag at the margin.

The chain of events we recorded at the beginning of the 2014 melt season indicates an upglacier-propagating drainage of the subglacial hydraulic system, notably the activation of the surfacing meltwater plume followed by the drainage of Cluster 1, which occur within 3 days over the lower ∼3 km of the glacier tongue. This is initiated near the terminus as drainage efficiency increases. This is likely to be either initiated via the onset of the speed-up which promotes longitudinal stretching, or by the formation of channels near to the glacier front which propagate upglacier and drawdown subglacial meltwater from the upper catchment area (i.e. a 'flushing event'). A similar event is possibly also seen at the end of the 2014 melt season, with the significant drop in water-pressure and re-activation of near-terminus channels (indicated by plume activity) in mid-September.

The observations from the borehole water-pressure record are strikingly different from borehole records in Alpine settings. These usually exhibit a diurnal signal, which reflects changes in delivery of meltwater to the bed and creates transverse hydraulic gradients that make meltwater pathways highly changeable (e.g., Meier et al., 1994; Hubbard et al., 1995). Consis-

tently high basal water-pressures have been associated with glaciers where the evacuation of meltwater from the subglacial environment is inefficient or where the drainage system is unstable (e.g., Sugiyama et al., 2011). The borehole record from Kronebreen supports this idea and further suggests that consistently high basal water-pressure may be exclusively associated with lake-terminating glaciers, tidewater glaciers, and glaciers undergoing surging. Similar observations were concluded by Doyle et al. (In review), with boreholes drilled to the bed at Store Glacier, Greenland showing consistent high water levels. Rapid drainage events have also been observed at other marine-terminating glaciers (e.g., Danielson and Sharp, 2013). The observed upglacier progression of drainage at Kronebreen, however, does not fit the proposed hypothesis that downglacier progression of drainage may be primarily associated with dynamic tidewater glaciers such as Helheim glacier (Everett et al., 2016).

It has previously been argued that changes in discharge at tidewater glaciers are accommodated through changes in conduit size rather than changes in the hydrological network. This idea largely stems from modelling and indirect measurements from large outlet glaciers in Greenland and Alaska (Pimental et al., 2010; Gimbert et al., 2016). Here, we propose that Kronebreen is able to accommodate fluctuations in discharge through changes in the subglacial hydrological network. This is based on the observation of additional plume activity during periods of rapid meltwater inputs to the bed, which are indicative of active channels. It is likely that the subglacial network can reconfigure because the ice is thinner than large ice sheet outlets. Thus, channels can remain open for longer because the thinner ice promotes slower creep closure rates. Reconfigurations could have a marked effect on the rate of submarine melting at the ice front beneath the waterline (e.g., Slater et al., 2015), and it would be worthwhile to investigate the effect of channel reconfigurations on ice front stability in future work.

### 7.5 Implications for subglacial dynamics

The velocity maps in Figure 5 show that the central/south region of the glacier tongue consistently flows faster than the north region. The largest accelerations are experienced at the terminus during the early-season speed-up event, increasing by 2 m $d^{-1}$ (since 04 June 2014). At this point, surface velocities in the central/south region exceed 3 m $d^{-1}$. These high velocities are likely to be the result of differences in the efficiency of the drainage beneath the north and central/south regions of the glacier tongue. Modelled hydraulic potential suggests that meltwater is channelled to the north region, assuming that flow routing is largely governed by ice-pressure gradients. This effectively isolates the central/south region from an efficient mechanism to evacuate meltwater. It is evident from observations of plume activity that channels cannot form for sufficiently long periods in this area, which enhances basal lubrication and is a contributing factor to the localised high velocities throughout the melt season.

Similar velocity patterns have been reported at other large outlet glaciers (e.g., Howat et al., 2005; Nick et al., 2009). It has also been observed in previous years at Kronebreen (Luckman et al., 2015). Schellenberger et al. (2015) emphasised the importance of basal lubrication based on observed links between velocity and surface water production at Kronebreen from 2007 to 2013. Inverse modelling by Vallot et al. (2017) shows that seasonal velocity variations at Kronebreen are controlled by variations in basal friction (closely following surface water runoff) and calving retreat of the front (which reduces back stress), with the former process dominant. Our results show that variations in the velocity field at Kronebreen are not only influenced

by surface water production but also by the specific configuration of the subglacial drainage system which is governed by ice-pressure gradients at the bed.

We also argue here that 2014 is an abnormal year for the dynamics of Kronebreen, based on the observations of a speed-up event at the end of the melt season (Luckman et al., 2015; Vallot et al., Accepted). It is likely that this speed-up was caused by an unusually-high rainfall event that overwhelmed a subglacial drainage system in a late-season phase with low efficiency. Doyle et al. (2015) observed a similar event near to the end of the 2011 melt season at Russell Glacier. They suggested that such speed-ups are amplified due to their late-season timing, which may also be the case at Kronebreen in this instance. Although the inefficiency of the subglacial system is partly accountable for the late-season speed-up, it is also likely that sustained high velocities were caused by the abnormally high rainfall event and the storage of this water in a distributed drainage system that was present under the central/south region of the glacier front. Changes in velocity are thus controlled by the location of efficient drainage at Kronebreen, and resulting patterns of bed friction.

## 8   Conclusions

Subglacial hydrology has been examined at a tidewater glacier in Svalbard using direct measurements of basal water-pressure in conjunction with measurements of hydrological components (supraglacial lake drainage, meltwater plume presence, and plume surface area), modelled components (melt, runoff, and hydraulic potential), and surface velocities derived from TerraSAR-X imagery. Two key events occur at Kronebreen which provide insights into the hydraulic regime during the 2014 melt season: 1) An upglacier-propagating drainage event over a significant region of the glacier tongue, with simultaneous measurements suggesting this was an episode of early-season subglacial flushing which occurred within a 3-day period (25–28 June) over a distance of 5 km; 2) An unusually high-rainfall event in mid-September which re-activated the subglacial drainage system and is argued to be the cause of persistent high surface velocities through the winter season (Vallot et al., Accepted).

Our observations suggest that the event at the beginning of the melt season is linked to changes in subglacial drainage pathways that are initiated near the terminus and result in the drawdown of subglacial meltwater from the adjacent upper catchment. It is likely that subglacial flow routing is largely governed by ice-pressure gradients, routing a significant proportion of meltwater to the north region of the glacier tongue (as shown by the presence of plume activity at the north side of the terminus and indicated by hydraulic potential modelling).

Observations of intermittent plume activity at the south side of the terminus imply that the drainage system for the central/south region of the glacier tongue is disrupted throughout the melt season. It is likely that a stable system cannot form because a smaller proportion of meltwater is routed to this area (as suggested by hydraulic potential modelling) and there is high basal motion due to persistent fast velocities through the melt season. Plume activity is disassociated from modelled runoff, which indicates that a distributed drainage system is active through the majority of the melt season (Slater et al., 2017). Periodic presence of a surfacing plume is suggested here to be a signal for storage and release of meltwater that is related to internally-driven processes that operate independent from tidal influence (Kavanaugh and Clarke, 2001). In effect, the plume activity is an indicator of modulated subglacial pulsing under the central/south region of the glacier tongue.

This storage of subglacial water is a key control on ice flow over the 2014 melt season. Surface velocities show that the onset of the seasonal speed-up is relatively early compared to modelled runoff (i.e. melt production at the surface which enters the englacial zone). This implies that meltwater could be bypassing storage at the surface earlier in the melt season than anticipated. The absence of plume activity early in the season further suggests that this meltwater is not being quickly evacuated from the subglacial zone. Therefore meltwater is possibly being stored at the bed and enhancing basal lubrication, which facilitates the early onset of the seasonal speed-up.

The surface velocities also reveal that the central/south region of the glacier tongue is faster flowing than the north region. This suggests that meltwater is being retained in the subglacial environment within the central/south region and a local distributed drainage system presides despite the presence of an efficient drainage system in the north region. This spatial pattern in surface velocity has been identified tentatively in previous years (Luckman et al., 2015; Schellenberger et al., 2015). It is evident that variations in the velocity field at Kronebreen are not only influenced by surface runoff but also by the specific configuration of the subglacial drainage system. The high velocities observed in the latter part of the 2014 melt season are abnormal due to an unusually high-rainfall event and storage of this water in a localised region of the glacier tongue which enhanced basal lubrication (Doyle et al., 2015). While it is acknowledged that glacier dynamics play a key role in ice velocity, it is argued here that changes in velocities are also controlled by the location of efficient and inefficient drainage, and the regions where water is stored and evacuated.

*Code and data availability.* It is intended to publicly release the PyTrx photogrammetry toolbox at a later date, along with the photogrammetry datasets used in this research.

*Author contributions.* PH is the primary author of this paper. In addition she developed the photogrammetric tools in PyTrx and processed the borehole and photogrammetric datasets used here. DIB is the project leader and had an active role in developing the ideas presented in this paper. NRJH designed the time-lapse camera systems, developed the photogrammetric tools in PyTrx and carried out the hydraulic potential calculations. BH led the borehole fieldwork. AL provided velocities from feature tracking through TerraSAR-X satellite imagery. HS assisted on all related fieldwork and aided in data processing. WJJVP provided melt and runoff data. KL provided a bed DEM for the Kronebreen catchment. JK provided a surface DEM, facilitated fieldwork, and gave helpful insight into the ideas presented in this paper. WB designed and installed the pressure sensors which were placed on the glacier bed.

*Competing interests.* No competing interests are present.

*Acknowledgements.* This work was funded by the Conoco Phillips-Lundin Northern Area Program through the CRIOS project (Calving Rates and Impact On Sea level, RiS-ID 6155). P. H. is supported by a NERC PhD studentship. TerraSAR-X data were provided by DLR (project OCE1503). This work would not have been possible without the logistical support provided by Airlift AS, the Norwegian Polar Institute Sverdrup Research Station in Ny Ålesund, and the University Centre in Svalbard (UNIS) Tech and Logistics team. We greatly acknowledge Alex Hart and the GeoSciences Mechanical Workshop at the University of Edinburgh for manufacturing the time-lapse camera enclosures that were used in this study. BH acknowledges capital equipment support from HEFCW/Aberystwyth University. We would also like to thank Silje Smith-Johnsen for her assistance in the deployment of the time-lapse cameras, Fiona Clubb for her guidance on data visualisation, Donald Slater for useful comments and feedback on this paper, Andreas Vieli as the nominated editor of this paper, and the three appointed reviewers which consisted of Shin Sugiyama and two anonymous reviewers.

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
