# Peer review of "Rapidly-changing subglacial hydrological pathways at a tidewater glacier revealed through simultaneous observations of water pressure, supraglacial lakes, meltwater plumes and surface velocities"

_The Cryosphere, 2017_

## Referee Comment (RC1) · S. Sugiyama (Referee) · 29 Jun 2017

General comments:

This paper presents field and satellite observations of glacier hydrology and ice dynamics near the front of a calving glacier in Svalbard. The data include subglacial water pressure measured in a bottom reaching borehole, which is one of a few direct observations of subglacial conditions near the front of a calving glacier. The observational data are compared with the results of glacier melt and run off modeling as well as subglacial hydraulic potential analysis. Based on the observations and analysis, the authors provides interpretations on hydrological conditions and their influence on the glacier dynamics during the summer 2014.

Glacier hydrology near the front of tidewater glaciers is drawing attention because it plays key roles in glacier dynamics and glacier discharge into the fjord. Despite its importance, processes related to subglacial hydrology are not well understood because in-situ observations are difficult. This study combined several different observations and numerical analysis to reveal the evolution of glacier hydrology over one summer melt season. This kind of integrated observational data set is available at only a few limited glaciers, thus presented data are valuable to improve our understanding of the hydrology of tidewater glaciers. Text is very well written and nicely organized, which clearly explains relatively complex methodology and observational results. I like the way of interpretation, first in chronological order in Section 6 and then discussion on selected important processes in Section 7. Plots and photographs are carefully prepared. Overall, I find the manuscript is already in a good standard and interesting to many of the journal readers.

The interpretation and discussion on the glacier hydrology are reasonable, but they are based on surficial observations and not much supported by direct evidences. I agree that they are likely scenarios, but other possibilities should be also mentioned. I listed such comments on the authors' interpretations followed by relatively minor comments and suggestions, which can be considered for revision.

Scientific issues:

5.5 Borehole pressure I understand the borehole pressure was recorded from September 2013. Why not showing all the data from the beginning of the observation? Water pressure over one year period provides insights into basal conditions as well as the connectivity of the borehole to the subglacial hydrological system. At least, overview

of the pressure record over the entire period should be described in the text.

7.2 Upward-propagating supraglacial lake drainage I wonder if glacier dynamics can be the cause of the lake drainage. When the glacier accelerates near the front, a longitudinally stretching flow regime is enhanced. This causes crevasse opening and increases chance of lake drainage. Assuming that such acceleration initiates near the glacier front and propagates upglacier, the observed lake drainage can be explained by this process.

7.4 Subglacial drainage of Kronebreen Throughout the paper, the authors assume the borehole pressure represents the subglacial water pressure over the region. Nevertheless, the lack of short-term pressure variations gives me an impression that the borehole is not well connected to active subglacial drainage system. The pressure drops in September, but it is only 15 m out of 280 water depth. I agree that the authors' interpretation is one of likely scenarios, but it is worth mentioning that there is a possibility that the borehole pressure does not represent basal conditions in the region.

Specific comments:

page 1, title: I think "Rapidly changing subglacial hydrology pathways" is not supported by evidence and does not fit the presented results. For example, "a stable efficient drainage system effectively transported this water through the north region... (page 1, line 6–7)" contradicts to "rapidly changing pathways". What about something like "Subglacial hydrology at a tidewater glacier as revealed through ..."?

page 1, line 7: What is "this water"? "subglacial meltwater"?

page 2, line 3: "Borehole data is spatially limited" » "Borehole data provide only spatially limited information"?

page 2, line 11: "upper section of the glacier" » Not necessary because it is not clear where it is.

page 2, line 17: "spatial patterns" » It sounds odd because borehole data do not show

spatial patterns, but useful to understand temporal variations.

page 3, line 16: "Subglacial transient pressure waves" » Not clear what are these waves. Please provide citations if this term is defined and used in previous studies.

page 4, line 3: "Most direct observations" » It sounds strange to say satellite imagery are the most direct observation.

page 4, line 8: Delete "a complete".

page 5, line 12: Why "Calving activity persists throughout the year"? Because of absence of sea ice?

page 5, line 31: "real-world" » Is this a common expression? "velocities, areas and distances in real space"?

page 6, line 2: "Digital Elevation Model" » "digital elevation model"

page 6, line 5: Trimble GeoXR "GPS" rover

page 6, line 6: post-processed "in a kinematic mode"?

page 6, line 15: from Site 3 on Colletthogda "(Fig. 1)".

page 7, line 1: "real-world extents" » surface area?

page 7, line 7: " Glacier" surface velocities were...

page 7, line 29: This was undertaken in order to isolate the hydrology of the glacier tongue (» isolate from what?) and better observe direct hydrological influence (» influence of what?) in the region of interest.

page 8, line 2: Can you explain more about the wireless pressure sensors (specification of the sensor and communication system)? Any citation for the borehole instrument used in this study?

page 8, line 3–4: Can you give uncertainties to the bed elevation and the ice thickness?

page 8, line 9–10: "High temporal resolution of the GPS data did not add any further insights to this study." » Even if you did not find short-term variations, it gives very important information to this study. Please clarify what you measured by the GPS.

page 8, line 11: "Local bed pressure" » "Subglacial water pressure"?

page 8, line 31: "Digital Elevation Model" is not necessary because DEM is already defined.

page 8, line 31–32: Any citation for the radar measurement?

page 9, line 4: The subsection title is odd because lake level was not measured. "Supraglacial lake drainage" or "Supraglacial lake area"?

page 9: line 9–18: Please refer to each of the photographs in Figure 3. Please also refer Fig. 2E to explain the lake evolution.

Figure 2: The order of the subplots is not consistent with that in the text. Why not listing the plots starting from the lake measurements, then melt modeling, velocity, and borehole pressure?

Figure 2A: MPa is more common (MKS unit system) as a unit of pressure.

Figure 2B: Unit of rainfall should be mm/time (mm/d?). To avoid the overlapping of the line (melt) and bar (rainfall), I suggest to plot rainfall upside down, i.e. bar extending downward from the top axis.

Figure 2D: Can you provide uncertainty range in the plot and describe in the text?

Figure 2E: The variations of Lake 2 and 3 are very difficult to read. What about plot them for a more suitable scale taken on the right axis?

Figure 4: Plume 1–4 in the plots should be Plume N1, N2, N3 and S1.

page 13, line 6: "Upon intersecting" » "Upon reaching?"

page 14, line 4: 297 m is a little higher than I expect as the floatation level of the 320

m thick ice. What kind of ice density did you use?

page 16, line 11: "2477" » "2,477"?

Figure 6: The lakes and plumes on the map are difficult to find. Please indicate the texts (Supraglacial lake, Plume N1, etc.) directly on the map. Please indicate numbers on the bed contour lines on the map and provide the contour interval in the caption.

page 20, line 3: What is "This water"?

page 20, line 7: "Water is not being stored in the snowpack and firn layer." » It is not likely that meltwater penetrates through snowpack and drains without storage in snow. Is there a possibility that snow cover was not accurately modelled, and in reality bare ice was already exposed in June?

page 20, line 15–16: "sufficient pressure has accumulated to force a channel, or multiple channels, to open." » I wonder if high pressure can open subglacial channels. Enlargement/closure of a channel is the result of melting of the conduit wall and ice deformation due to isostatic pressure. This expression "pressure accumulates and force a channel open" appears again and again (page 21, line 21 and 33; page 24, line 23). Please make sure if this sentence accurately explains processes in your mind. Is it pressure or meltwater which accumulates at the bed?

page 21, line 10: Omit "episodes where there is"?

page 22, line 11: What kind of glaciers are you referring to by "other tidewater glaciers"? This question is because water depth is usually far below sea level at the fronts of tidewater glaciers. Kronebreen is not a special case, I think.

page 22, line 17: "efficient channels briefly form then collapse" » Why they collapse briefly? When water pressure is high (effective pressure is low), closure of channels should be slow.

page 22, line 24: It is not clear to me what "transient low-pressure wave". Do you

mean that subglacial water pressure drops as drainage efficiency increases, and it propagates upglacier as a channels system develops from the glacier front to upper reaches? Can you provide references for this process?

page 23, line 19: "as melwater continues to accumulate" » Something like "as surface meltwater gets access to the glacier bed"?

page 24, line 15: "A stable efficient drainage system can form here ...." » Ice speed is slower near the margins, but shear deformation is larger than the fast flowing central region. I wonder if this hypothesis is convincing enough to be here in conclusion.

page 24, line 20–21: disassociated from "modelled" runoff.

page 24, line 26: compared to "modelled" runoff

---

## Referee Comment (RC2) · Anonymous Referee #2 · 4 Jul 2017

Summary This manuscript presents a combination of field and satellite observations of borehole water pressure, supraglacial lake area, ice velocity and plume extent at Kronebreen, Svalbard. These observations are combined with surface melt, runoff and subglacial hydropotential modelling to infer spatial and temporal variations in the glacier's subglacial drainage system, and the relationships between these variations and ice flow. The manuscript presents a useful, multi-faceted dataset, but currently the analysis of the data is simplistic and not fully supported by the evidence presented (for

example, suggestions that tides influence the timing of plume pulses). As a result the conclusions are rather vague, and less significant than they could be. Also, it seems as though from the outset that hydrology was identified as the principal control on ice dynamics, and other potential factors have been ignored.

Main points

1. No attempt is made to investigate alternative controls on ice velocity apart from variations in subglacial hydrology. This is especially pertinent for the early season 'flushing event' which causes the up-glacier drainage of the supraglacial lakes. 2. The borehole water pressure gradually decreases while ice velocity is increasing, which does not tie in with your explanations of ice motion being controlled ' by the location of efficient/inefficient drainage and the position of regions where water is stored and evacuated from' (pg. 1) 3. The description of seasonal variations in ice flow (i.e. that the speedup is constrained to the southerly part of the near-terminus region) does not seem to be supported by the example velocity images shown. It would be useful to produce some plots showing relative changes in ice velocity, so that the reader can see the justification for the discussion. 4. Assertions made in the discussion should be backed up with data and results. For example the suggestion that tides influence the timing of plume pulses (but there are many other similar examples as detailed in the specific comments below).

Specific points (by page and line number)

P2, L3: Data are plural. 'data is' should be 'data are'

P2, L16: This strikes me as a bit of a strange statement; what is a 'pressure environment'? Presumably you mean basal water pressure and effective pressure?

P2, L20: How rapid? Maybe better to specify several hours etc.

P2, L23: Presumably the statement about 'inefficient evacuation of meltwater' relates to the fact that the basal hydraulic system at the terminus is not at atmospheric pressure

so the along-glacier hydraulic gradient is less steep? It would be worth clarifying this.

P2, L26: In terms of land-terminating glaciers on ice sheets, this description is only really applicable to the marginal 20 km or so. Further up-glacier, maximum ice flow typically occurs later in the overall melt season (i.e. later than the onset of melting at the margin), once melt has commenced at higher elevations. It may be worth specifying that you are referring to land-terminating valley glaciers here?

P2, L28: What exactly do you mean by 'subglacial drainage re-organisation'?

P2, L30: 'Ice velocity records indicate similarities to land-terminating glaciers'. Do you have a reference for this statement? And do you mean the terminus region of tidewater glaciers of further inland?

P3, L6: 'The drainage of supraglacial lakes provides an additional meltwater input into the subglacial environment'. The use of 'additional' here is a little odd - additional to what? I assume you mean in addition to the drainage of surface meltwaters before they accumulate into lakes, but this is not obvious from the previous paragraphs.

P3, L15: You need to make clear that this phenomenon has been observed in the terminus region (last 20 km) of a single tidewater glacier in Greenland. As written, you risk suggesting that such drainage is as common or prevalent as the up-glacier progression of lake drainage with time, which is not the case.

P3, L18: 'terminus' should be 'near-terminus'

P3, L16: Wat exactly are 'Subglacial transient pressure waves'? How do these control up-glacier progression of lake drainage? This requires more explanation and a reference.

P3, L21: What is 'the bed system'? Better to avoid these vague terms and simply describe what you actually mean, which you do in the next-but-one sentence. Also, I'm not quite sure what the point of this sentence is 'This has largely been studied in inland and near-terminus settings'. Are there other 'settings' from which such observations

are missing?

P3, L23-24: 'long periods of time'. Be more specific - days, weeks, months, years, decades? I assume decades based on the Tedstone reference but this would not be obvious to someone unfamiliar with the literature.

P3, L26: How do these 'long residence times' tie in with observed reductions in ice surface uplift and velocity about 24 h after a lake drainage?

P3, L27: Would not injection of water into a distributed system or a small channel also be capable of causing uplift?

P3, L28: I think there are too many 'systems' mentioned in these introductory sections. It gets a bit confusing after a while.

P4, L5: I think you need to note here that Slater et al (2017) used photographs from a time-lapse camera rather than satellite images. Also, time-lapse is also temporally intermittent - it is the frequency of that intermittency that is important!

P5, L15: This sentence is convoluted: 'Kronebreen retreated ∼1 km between 2011 and 2016' would say the same more succinctly.

P6, L7: The GPS position errors seem quite large. Why not post-process them using for example TRACK software? Also, it is not mentioned anywhere how these positional errors affect your measurements, of, for example, supraglacial lake area.

P6, L10: 'data was' should be 'data were'

P6, L16-17: How did you avoid falsely identifying shadows and sediment-rich ice as lakes (both also have a high contrast with bare ice)?

P8, L32: Seems like a word is missing before 'which'. Could it be 'campaigns'?

P9, L8: 'but more central and nearer to the glacier's central flow line' should be 'but nearer to the glacier's central flow line'

P9, L21-23: Both sentences are pretty much repeated from the first paragraph of this section.

Figure 2: It would be useful to also put the dates along the top x axis.

P11, L6-7: Or more accurately, fill enough that water is visible in the TL imagery...

P11, L22-23: 'Modelled melt production has a diurnal pattern with a maximum in the day and minimum at night' – I think this is well enough known that it is not necessary to report here.

P11, L27: The bracket after the units shouldn't be superscript.

P11, L29-31: Some repetition from previous paragraph re diurnal variations etc.

P12, L6: It would be informative to outline these on the velocity plots in Figure 5

P12, L16: Based on Fig. 2, it seems like the melt season lasts through until midway through September. I therefore do not think it is correct that the velocities were consistent for this period.

P13, L3: They definitely coincide (inasmuch as the temporal frequency of the TSX data allows), so do you mean 'possibly caused by'?

P13, L3-4: This is a bit vague. Ok, at the near-terminus centreline, velocities are higher in September than in the pre-melt season period, but at the other two ROIs they are broadly the same as before the melt season.

P13, L7: But despite this, there is actually very little variation in borehole water pressure (as shown in fig. 2). Perhaps the borehole is actually not that well connected to the regional basal hydrological system?

P14, L26: 'supraglacial lakes' should be 'supraglacial lake area'

Figure 5: It is difficult to distinguish between no data (transparent) and >2.4 m/d (partly opaque white?). Could the colour scale be changed to make this clearer? It is not

currently clear where the down-glacier extent of good data is.

P16, L11 (and throughout): 'upward-propagating' should be 'upglacier-propagating' (otherwise suggests vertical propagation).

P16, L18-19: 'This implies that meltwater is present at the bed and is enhancing basal lubrication '. Why is this necessarily the case? You should refer to another dataset - e.g. basal water pressure etc. to support this statement. What is to say that the acceleration is not due to a reduction in buttressing at the calving front?

P16, L21-22: And all other glacier catchments assuming a normal lapse rate...

P16, L23: So do you think that basal frictional melt is an important factor for accounting for the remaining meltwater at the bed?

P16, L25-26: No, it only indicates that theoretically this is the expected route of subglacial water, not the configuration of the drainage system.

P16, L30-31: Can you not easily quantify this from your hydropotential analysis? This would be a useful addition.

Figure 6: Based on the size of the subglacial catchments (N bigger than S), why is the southern plume expression so much more extensive? What date is the plume extent from?

P17, L3-4: Might this indicate that the drainage is linked to a perturbation at the calving front? For example, an acceleration and consequent longitudinal stretching related to a calving event or break-up of seasonal cover adjacent to the glacier?

P17, L4: 'upglacier-propagating'

P18, l8: What exactly does 'spatially discrete' mean in this context?

P18, l23: But if the plume is periodically visible, does this not suggest that the basal water is also purged periodically? Do you see a difference in velocity between times

when the southern plume is and is not visible?

P20, L14-15: Could it not be that the volume of meltwater is just insufficient for the plume to either reach or be visible at the fjord surface?

P20, L18: 'upglacier-propagating'

P21, L15: 'unexpectedly absent during periods of high runoff'. Suggesting more distributed outflow of meltwater at the grounding line?

P21, L18: 'varies over only a small range' (it does vary)

P21, L22-23: 'The precise timing of each outflow is possibly controlled by marine dynamics such as tidal level.' Do you see any evidence for cyclicity on the frequency of tides? Otherwise, what evidence is this statement based upon?

P21, L25-26: 'The trigger for the release of this water could be related to this hydraulic pulsing'. This seems a little too speculative.

P22, L11: 'The key difference at Kronebreen, and other tidewater glaciers, is the high hydraulic base-level...' should be 'The key difference at Kronebreen, and other tidewater glaciers, compared to land-terminating glaciers, is the high hydraulic base-level...'

P22, L15-16: You should say why 'a stable drainage system cannot exist in this region'; presumably because the high velocities preclude the formation of persistent channels?

P22, l24: What do you think causes the 'glacier-wide transient low-pressure wave that is initiated near the terminus', and what evidence do you have to support this assertion?

P22, L29: What does 'This' refer to?

P23, L8: 'due to the difference in ice thickness'. Be more specific, it is not just the difference, but the fact that the ice is shallower that is key here. Also, you should state that the thinner ice leads to slower creep closure rates, meaning that channels are easier to open and maintain.

P23, L10-11: You should be clear that you are referring here to the mean melt rate for the entire submerged ice front rather than the localised melt rate (which is likely to be greater for more spatially-focused discharge).

P23, L13: I'm not sure I agree with this statement. From Fig. 5 it looks like there is also a speedup at the northern part of the tongue. Could you provide a relative change in velocity map to evidence your assertion?

P23, L20: What exactly is mean by 'consistent' here? Spatially consistent (if so this is different to the results presented in this paper), temporally consistent?

P23, L27-29: This sentence seems contradictory - please clarify...

P24, L9: It might be worth also referencing Doyle et al. (2015), Nature Geoscience who saw a similar effect at a land-terminating glacier in west Greenland.

P24, L27: Or indeed that it is something other than meltwater that triggers this initial speed-up - e.g. a calving event, break up of sea ice etc.

---

## Author Comment (AC1) · 6 Jul 2017

We would like to thank the reviewer (Shin Sugiyama) for their comments, and their positive response to our manuscript. Sugiyama's enthusiasm and curiosity for the subject is evident in his feedback, which is very refreshing to read. We have edited our manuscript accordingly, including edits to Figure 2, the inclusion of glacier dynamics as an explanation for the cause of the lake drainage at the beginning of the 2014 melt season, and the inclusion of a scenario where the borehole pressure does not represent basal conditions in the region.

Details of our response to the reviewer's three key comments (numbered) and minor comments are outlined below. All typos, grammatical corrections and minor sentence changes that were suggested by the reviewer have also been agreed to and changed within the manuscript. These smaller changes are not outlined here in order to keep this response as brief as possible.

*1. I understand the borehole pressure was recorded from September 2013. Why not showing all the data from the beginning of the observation? Water pressure over one year period provides insights into basal conditions as well as the connectivity of the borehole to the subglacial hydrological system. At least, overview of the pressure record over the entire period should be described in the text.*

The borehole pressure record covers a 14-month period from September 2013 to December 2014. We understand that this is a very valuable dataset that should be shared with the scientific community as soon as possible. However, it was decided to only focus on the 2014 melt season because of two main reasons:

- We believe that the inclusion of the whole record is beyond the scope of the paper. The inclusion of the whole record may detract from the key aim in this paper, which is to build a detailed theoretical model of the hydrology at the glacier terminus of a tidewater glacier during a single melt season. We believe that the entire dataset is not needed to fulfil this aim.

- The beginning of the record (September 2013–March 2014) is strikingly different from the rest of the record. For instance, basal water-pressure appears to exhibit strong, consistent diurnal variability (roughly between 10–50 kPa) from September 2013–March 2014, whilst the rest of the record does not indicate any

diurnal variability. This may be because sensor took a while to settle and give consistent readings, or basal pressure drastically changed over the monitoring period, or the sensor may have been located on a different part of the bed and was subject to a different pressure/hydrological environment. This in itself is an interesting observation and we are still attempting to understand this. Once we have gained a better understanding (and potentially integrated it with subglacial hydrology modelling), it is intended to publish the borehole dataset in its entirety at a later date in a CRIOS project publication.

For these reasons, the entire borehole record will not be included here. Also, an overview will not be included in the text because we believe that the significant difference in the record from September 2013 to March 2014 does not reflect the subglacial conditions in the 2014 melt season.

*2. I wonder if glacier dynamics can be the cause of the lake drainage. When the glacier accelerates near the front, a longitudinally stretching flow regime is enhanced. This causes crevasse opening and increases chance of lake drainage. Assuming that such acceleration initiates near the glacier front and propagates upglacier, the observed lake drainage can be explained by this process.*

Section 7.2 (Upward-propagating supraglacial lake drainage) outlines the dynamics of the three lake clusters monitored in this study and compares their dynamics to other observations from the literature. The lakes in Cluster 1 are focused on in particular because of the coincident timing of their drainage in relation to changes in velocity, runoff and plume activity. The nature of their drainage is discussed in relation to hydrology and it is hypothesised that their drainage is related to their connectivity to efficient drainage in the subglacial environment. Glacier dynamics were not discussed here to avoid repetition with Section 7.5 (Implications for subglacial dynamics).

However, the reviewer rightfully points out that glacier dynamics may be the cause of

the lake drainage and the reader may gain the impression that the drainage of the lakes in Cluster 1 is exclusively linked to hydraulic connectivity. Glacier dynamics may also play a key role in their drainage. Longitudinal stretching is likely to be enhanced at the beginning of the season when the glacier begins to accelerate and this could, in turn, promote the likelihood of lake drainage. As suggested by the reviewer, this hypothesis has now been included in section 7.2 to provide a more detailed explanation for the drainage of these lakes. It is suggested that their drainage may be related to glacier dynamics as well as glacier hydrology:

'The lakes in Cluster 1 are of particular interest because of the coincident timing of their drainage in relation to changes in surface velocities, runoff, and activation of the plume at the beginning of the melt season. This suggests that these lakes are linked to a common channelised system when they drain. The upward-propagating nature of their drainage indicates that channels develop in an upglacier progression as reflected in the timing of their connection to thr subglacial environment. The hydraulic potential modelling supports this as it indicates that Cluster 1 may be situated close to a large channel/flow accumulation pathway. Glacier dynamics may also play a key role in the cause of this lake drainage. Longitudinal stretching occurs as the glacier accelerates at the beginning of the season, which facilitates the opening of crevasses and increases the chance of lake drainage. The upward-propagating nature of the drainage may be a result of this early-season acceleration, assuming that it initiates at the glacier front and propgates upglacier.'

*3. Throughout the paper, the authors assume the borehole pressure represents the subglacial water pressure over the region. Nevertheless, the lack of short-term pressure variations gives me an impression that the borehole is not well connected to active subglacial drainage system. The pressure drops in September, but it is only 15 m out of 280 water depth. I agree that the authors' interpretation is one of likely scenarios, but it is worth mentioning that there is a possibility that the borehole pressure does not*

*represent basal conditions in the region.*

Hydraulic potential modelling suggests that the borehole is located close to/within the catchment of an efficient channel system, and thus the record reflects basal water-pressure in a well connected region of the glacier bed. However, the borehole record shows few short-term variations over the entire study period that this manuscript covers (May–September 2014), which suggests that the borehole is isolated from the active subglacial drainage system.

The reviewer is right to point out that there is a possibility that the borehole may not be located in an efficient drainage catchment based on the lack of short-term pressure variations. A paragraph (page 22, line 16 – 21) has been added to Section 7.4 (Subglacial drainage of Kronebreen) to address this point:

'Few short-term pressure variations are observed in the water-pressure record from May–September 2014, apart from the significant drop in pressure at the end of the melt season. It is possible that the borehole is located on an area of the bed that is not well connected to an active, efficient drainage system. However, changes in water-pressure have been observed to coincide with other features in the hydrological system (i.e. plume activity and supraglacial lake drainage), which suggests that the borehole is hydraulically connected to some degree. This is also supported by the modelled hydraulic potential, which indicates that the borehole is located close to, or possibly within, an efficient drainage catchment.'

*Page 1, title: I think 'Rapidly changing subglacial hydrology pathways' is not supported by evidence and does not fit the presented results. For example, "a stable efficient drainage system effectively transported this water through the north region... (page 1, line 6–7)' contradicts to 'rapidly changing pathways'. What about something like 'Subglacial hydrology at a tidewater glacier as revealed through ...'?*

The title 'Rapidly changing subglacial hydrology pathways' is in reference to the unstable, changing drainage system beneath the south region of the glacier terminus, which is suggested to facilitate the upward-propagating nature of the speed-up observed at the beginning of the 2014 melt season. This is what we believe is the key take-home message of the paper, hence why the paper is titled accordingly. The authors wanted a title that made the paper stand apart from other subglacial hydrology studies. The title suggested by the reviewer, 'Subglacial hydrology at a tidewater glacier as revealed through. . .', is over-used in our opinion and does not grab the readers' attention. The title we have chosen reflects the uniqueness of the study and makes it stand apart from others. For these reasons, the title remains unchanged.

*Page 3, line 16: 'Subglacial transient pressure waves'. Not clear what are these waves. Please provide citations if this term is defined and used in previous studies.*

The term has commonly been used to describe events where high-pressures propagate through the subglacial zone of a glacier due to high pressure gradients. They have been associated with surges (Kamb et al., 1985) and have been used to propose an alternative explanation to hydrofracturing for the filling/draining of supraglacial lakes (Everett et al., 2016). However, the term can also lend itself to instances where low-pressures propagate through the subglacial zone of a glacier.

Both reviewer 1 and reviewer 2 have stated that the use of the term 'subglacial transient pressure wave' is convoluted and it appears that this may be misinterpreted by the reader. For this reason, the term has been omitted from this paper. The term was largely used to describe the events at the beginning of the melt season, which has now been replaced with better details concerning the glacier-wide drawdown of meltwater in the near-terminus area.

*Page 5, line 12: Why 'Calving activity persists throughout the year'? Because of absence of sea ice?*

Calving activity is persistent throughout the year because of the warm, saline Atlantic water that can freely enter the fjord throughout the year (Luckman et al., 2015). This sentence has now been changed to clarify that this:

'Calving activity persists throughout the year due to the presence of warm sub-surface ocean water, even in the winter season...'

*Page 5, line 31: 'real-world'. Is this a common expression? 'velocities, areas and distances in real space'?*

The term 'Real-world' measurements is often used in photogrammetry to distinguish absolute measurements (e.g. metres, $md^{-1}$, $m^3$) from relative, pixel measurements that are made from images. It is important to distinguish between these two types of measurements. The text has been left unchanged.

*Page 7, line 29: 'This was undertaken in order to isolate the hydrology of the glacier tongue (isolate from what?) and better observe direct hydrological influence (influence of what?) in the region of interest'.*

It was decided to reduce the melt/runoff model catchment size to isolate the hydrology of the glacier tongue from hydrological influence in the upper catchment (i.e. Holtedahl-fonna), and better observe direct, immediate hydrological effects in the region of interest. This sentence has now been changed accordingly:

'This was undertaken in order to isolate the hydrology of the glacier tongue from hydrological influence in the upper catchment area (i.e. Holtedahlfonna), and better observe direct hydrological effects in the region of interest.'

*Page 8, line 2: Can you explain more about the wireless pressure sensors (specification of the sensor and communication system)? Any citation for the borehole instrument used in this study?*

The wireless pressure sensor is a WiSe (Wireless Sensor system) developed at the Institute for Marine and Atmospheric Research, Utrecht, first used by Smeets et al. (2012) at Russell Glacier (and at the drill site of the North Greenland and Eemian Ice Drilling project – NEEM –.for testing purposes). These probes are custom-made for glacial applications. It measures in-situ pressure, temperature and tilt every two hours, and these readings can be transmitted to a receiver through < 2500 m thick ice. Smeets et al. (2012) fully document the design of the WiSe systems, therefore it was decided to include this as a citation in the manuscript:

'More details about the specifications of these wireless sensors is presented in Smeets et al. (2012).'

*Page 8, line 3–4: Can you give uncertainties to the bed elevation and the ice thickness?*

Spot heights (for the bed and ice surface) from the borehole sites were derived from the bed and surface DEMs outlined in section 4.5 (Methods: hydraulic potential modelling). Ice thickness was calculated from these spot heights. The maximum vertical root mean-squared uncertainty in the interpolated surface and subglacial DEMs is approximately $\pm 15$ m. This information has now been added to both section 4.4 and section 4.5.

*Page 8, line 9–10: "High temporal resolution of the GPS data did not add any further insights to this study." Even if you did not find short-term variations, it gives very important information to this study. Please clarify what you measured by the GPS.*

The GPS data was not included in this study for three main reasons:

- The GPS velocity record is incomplete. The GPS was offline at the beginning of September 2014, whilst the rest of the dataset record carries on till the end of September 2014. The record duration is therefore mismatched.

- The higher temporal resolution of the GPS velocities does not appear to add anything new to the study. There were difficulties in processing the GPS data and short-term variations cannot be distinguished from the daily positions that we extracted. The dataset generally appears noisy. To resolve this and provide an alternative, velocities were derived from the TerraSAR-X imagery and then a spot velocity was extracted from the borehole site. These appear much less noisy and fit well with the rest of the 2014 record.

- The key findings from the velocity data focus on the spatial variability in velocity over the glacier tongue, rather than changes in velocity over time. These are better addressed with the TerraSAR-X velocities rather than the GPS velocities. The inclusion of the TerraSAR-X velocities from the borehole site are also consistent with the velocities derived from the other ROI's (i.e. from the centreline and the supraglacial lakes).

For these reasons, the GPS data will not be included in this paper. The difficulties with integrating the GPS velocities has been clarified at the end of the paragraph (page 8, line 11) stating:

'It was decided to use the surface velocities derived from TerraSAR-X images rather than the GPS because the GPS velocity record was incomplete and the higher temporal resolution of the GPS data did not add any further insights to this study. The GPS data appeared noisy due to difficulties in processing the positions.'

*Page 8, line 31–32: Any citation for the radar measurement?*

The paper that includes this work is still in preparation for submission, albeit the paper will be submitted imminently:

Lindbäck, K., Kohler, J., Pettersson, R., Myhre, P.I., Nuth, C., Langley,K., Brandt,O., Messerli, A. and Vallot, D., In Prep. Subglacial topography, geology and future bathymetry of Kongsfjorden, northwestern Svalbard.

The citation will be added if it is submitted before all corrections are compiled and re-submitted for this paper.

*Page 9, line 9–18: Please refer to each of the photographs in Figure 3. Please also refer Fig. 2E to explain the lake evolution.*

The paragraph has been changed to include references to each of the photographs in Figure 3 and to the surface areas from the composite graph (Figure 2):

'While the lake clusters appear to act independently, the lakes within Cluster 1 fill and drain almost simultaneously, indicating that they are hydrologically linked. A timeline of changes in lake surface area at Cluster 1 is shown in Figure 3. Cluster 1 fills and drains first, beginning to fill from 01/06/2014 07:00 (Fig. 3A–D) and initially draining on 27/06/2014 03:00 over 59 hours (Fig. 3E–F), decreasing from a total surface area of 41,374 $m^2$ to 2477 $m^2$ (see Lake 1 group surface area in Fig. 2A). The lakes gradually drain after this, leaving them empty by 21/07/2014 14:00 (Fig. 3G–J).'

*Figure 2: The order of the subplots is not consistent with that in the text. Why not listing the plots starting from the lake measurements, then melt modeling, velocity, and borehole pressure?*

The authors agree with the reviewer that the plots should be listed in the same order

as introduced in the text. This will make the manuscript easier for the reader to follow. The order of the plots in Figure 2 have been changed to:

A) lake areas; B) Plume presence; C) Plume surface area; D) Melt and precipitation; E) Runoff; F) Velocity; G) Borehole pressure

Figure references in the manuscript have also been changed to correspond with this new ordering.

*Figure 2A: MPa is more common (MKS unit system) as a unit of pressure.*

Pascal is the SI unit of pressure, and it is agreed that these units would be more appropriate than bar units. It was decided to use kilopascal (kPa) references rather than megapascal (MPa) references as changes in water-pressure in this study are relatively small.
Kilopascal now replaces bar as the used unit of pressure. This change has been made to Figure 2 and also any reference to bar pressure measurements have been changed to kilopascals.

*Figure 2B: Unit of rainfall should be mm/time (mm/d?). To avoid the overlapping of the line (melt) and bar (rainfall), I suggest to plot rainfall upside down, i.e. bar extending downward from the top axis.*

This was merely a typo mistake and the units of rainfall have now been changed to mm per day. The plot has now been changed so that rainfall bars are extending downward from the top axis, making it easier to distinguish.

*Figure 2D: Can you provide uncertainty range in the plot and describe in the text?*

The uncertainty range for the velocities derived from TerraSAR-X image pairs is <0.4 m/day (Luckman et al., 2015). The figure has been changed to include this uncertainty range in the velocity plot. The figure caption has also been changed to clarify this.

*Figure 2E: The variations of Lake 2 and 3 are very difficult to read. What about plot them for a more suitable scale taken on the right axis?*

The authors agree that variations in Lakes 2 and 3 are difficult to read. Therefore the plot has been changed and they have been plotted on a seconary axis with a more appropriate scale.

*Figure 4: Plume 1–4 in the plots should be Plume N1, N2, N3 and S1.*

The authors agree with the reviewer and the plumes in this figure are now labelled accordingly.

*Page 14, line 4: 297 m is a little higher than I expect as the floatation level of the 320 m thick ice. What kind of ice density did you use?*

Floatation (m w.e.) is calculated given the ice thickness (320 m at the borehole site) multiplied by ice density. For ice density with no snow or firn layer, a density between 910 and 917 kg m$^{-3}$ is commonly used. Therefore the floatation level here is between 291–293 m, so the reviewer rightly points out that the value given in the paper (297 m) is a little higher than expected. This is simply a mathematical error that was not spotted by the authors. However, there are a lot of unknowns and the local bed topography around the borehole will change and could easily vary by 5–10 m. Therefore, 297 m is still a realistic floatation value.

The floatation value has been changed to 291–293 m, based on an ice density between

910 and 917 kg m$^{-3}$. This has been clarified in the manuscript.

*Figure 6: The lakes and plumes on the map are difficult to find. Please indicate the texts (Supraglacial lake, Plume N1, etc.) directly on the map. Please indicate numbers on the bed contour lines on the map and provide the contour interval in the caption.*

All recommendations that the reviewer outlines here have been agreed upon and changed. The plume names and lakes are now labelled on the map, and bed contours have been annotated with the contour interval (50 m) provided in the caption.

*Page 20, line 3: What is "This water"?*

Sentence omitted. Not needed.

*Page 20, line 7: 'Water is not being stored in the snowpack and firn layer.' It is not likely that meltwater penetrates through snowpack and drains without storage in snow. Is there a possibility that snow cover was not accurately modelled, and in reality bare ice was already exposed in June?*

The authors agree that it is unlikely that meltwater penetrates through the snowpack and drains without storage in the snow. This is what the authors were attempting to convey in this sentence, but obviously better clarification is needed. It is likely that either water is transported to the bed via crevasses or there is bare ice already exposed in the early part of the melt season.

The wording of the first sentence has been changed to better clarify that water is potentially bypassing storage and the paragraph has been changed to better encompass different scenarios for why this is occurring:

'This implies that water is bypassing storage in the snowpack and firn layer. The lower area of the glacier tongue is a heavily crevassed surface, providing abundant meltwater pathways to the glacier bed. It is likely that early-season melt production is directly routed to the bed in the lower region of the glacier tongue via these pathways. Equally, there is a possibility that snow cover is absent in June, and bare ice is already exposed in the early part of the melt season. Van Pelt and Kohler (2015) clarify that the model does not account for small-scale variability in precipitation and snow cover. For this reason, it is possible that water is being delivered to the bed earlier than the model anticipates.'

*Page 20, line 15–16: 'sufficient pressure has accumulated to force a channel, or multiple channels, to open.' I wonder if high pressure can open subglacial channels. Enlargement/closure of a channel is the result of melting of the conduit wall and ice deformation due to isostatic pressure. This expression 'pressure accumulates and force a channel open' appears again and again (page 21, line 21 and 33; page 24, line 23). Please make sure if this sentence accurately explains processes in your mind. Is it pressure or meltwater which accumulates at the bed?*

It is also possible that channel melting is also a key process at Kronebreen. However, it is difficult to distinguish in this study whether it is pressure or meltwater which accumulates at the bed. For this reason, a scenario where meltwater accumulation causes channel melt-back has been added to instances where channel opening is hypothesised.

Page 21, line 21: Sentence changed – 'It is likely that it is released either when sufficient pressure has accumulated to force a channel to open, or when subglacial water has sufficiently melted the cavity/conduit wall.'

Page 21, line 33: Sentence changed in a similar manner to previous.

Page 21, line 33: Sentence removed to avoid repetition.

Page 24, line 23: Sentence changed – 'Meltwater is released when a sufficient amount of pressure has accumulated to force a channel open and/or when subglacial meltwater has sufficiently melted the cavity/conduit wall.'

*Page 22, line 11: What kind of glaciers are you referring to by "other tidewater glaciers"? This question is because water depth is usually far below sea level at the fronts of tidewater glaciers. Kronebreen is not a special case, I think.*

The sentence was meant to highlight the key difference between subglacial water-pressure at tidewater glaciers (including Kronebreen) in comparison to land-terminating glaciers. It is clear that the sentence was not appropriately worded to convey this. This idea is also explained in page 22 line 27 – page 23 line 2. The sentence has been removed to avoid repetition.

---

## Author Comment (AC2) · 11 Jul 2017

We would like to thank the reviewer for their comments and constructive response to our manuscript. Their attention to detail has been very valuable for addressing the key points outlined. The authors have taken time and care to respond to each of these points.

We have edited our manuscript accordingly, including the proposal of alterntive controls

on ice velocity, and revisions to Figure 5 and associated descriptions concerning the observed early-melt season speed-up event. Details of our responses to the individual comments are outlined below.

1. No attempt is made to investigate alternative controls on ice velocity apart from variations in subglacial hydrology. This is especially pertinent for the early season 'flushing event' which causes the up-glacier drainage of the supraglacial lakes.

The main focus of this manuscript is to examine subglacial hydrology at a tidewater glacier, and investigate its influence on glacier dynamics, including ice velocity. The authors felt that investigating alternative controls on ice velocity was beyond the scope of the study, and little data was collected to adequately examine other influences (i.e. calving dynamics and oceanic forcing). However, the reviewer rightfully emphasises throughout their comments that the exploration of other influences is important to presenting a rounded paper that is not weighted towards one set of influences. There is a risk that the reader could misinterpret that subglacial hydrology is the sole control on ice velocity.

For this reason, the manuscript has been extensively altered to better represent alternative controls on subglacial hydrology. A large effort has been made to better outline all alternative influences, especially in the interpretation and discussion sections where we begin to introduce explanations and ideas concerning the changes we see at Kronebreen over the 2014 melt season. We hope that this is reflected in the detailed responses to subsequent comments. To summarise here, the following alternative controls on ice velocity have now been included in the manuscript:

- · Changes in calving activity
- Tidal influences
- · Changes in fjord conditions (e.g. subsurface temperature)
- Basal frictional melting
- Ice thickness and shallowness

In particular, changes in calving activity have been more thoroughly explored as an explanation for the 'flushing event' that is observed at the beginning of the melt season.

These alternative controls have been outlined in the interpretation section (Section 6), explored further in the discussion section (Section 7) if needed, and acknowledged in the conclusion section (Section 8). Additional datasets (such as tidal data, calving activity, and fjord temperature) have not been included to examine these alternative controls within this study. The authors argue that too much focus on these aspects will detract away from the main focus of the paper which is to investigate subglacial hydrology and its influence on glacier dynamics. The authors wish to retain one of the main message of the study – that, in addition to glacier dynamics, subglacial hydrology plays a vital role in ice velocity at tidewater glaciers.

2. The borehole water pressure gradually decreases while ice velocity is increasing, which does not tie in with your explanations of ice motion being controlled ' by the location of efficient/inefficient drainage and the position of regions where water is stored and evacuated from' (pg. 1).

Reviewer 1 previously highlighted that the borehole water-pressure record may not have strong connectivity to the active drainage catchment of the glacier, based on similar observations to those made here. A gradual decrease in water-pressure at the borehole whilst ice velocity increases suggests that the borehole is effectively isolated from the main drainage system.

However, small, coinciding changes have been observed between water-pressure and
other observed signals for subglacial hydrology – for example, the observed pressure drop in the early-season 'flushing event' which coincides with the activation of the main plume and the drainage of the supraglacial lakes. Such changes indicate that the borehole is influenced by changes in pressure within the active drainage catchment. This is supported by the hydraulic potential modelling, which shows that the borehole is likely to be located within, or at least near to, an active channel network.

The manuscript has been changed to better convey these possible scenarios. Also, modifications have been made in Section 6, 7 and 8 (the Interpretation, Discussion and Conclusion sections) where arguments have been supported with evidence from the borehole water-pressure record. These have been made in an attempt to clarify that the borehole record may not be connected to the active drainage system.

3. The description of seasonal variations in ice flow (i.e. that the speedup is constrained to the southerly part of the near-terminus region) does not seem to be supported by the example velocity images shown. It would be useful to produce some plots showing relative changes in ice velocity, so that the reader can see the justification for the discussion.

During the 2014 melt season, sequential velocity maps from TerraSAR-X image pairs show an early-melt season speed-up which initiates at the terminus and propagates upglacier. The fastest velocities are seen in the southern/central region of the glacier tongue (>2.4 md-1), whilst velocities in the north region generally are less (<2.4 md-1). The high velocities subside by August. A second speed-up is observed in September, which propagates upglacier in a similar manner. The authors realise that the wording used to describe these events was misleading. High velocities are constrained to the southern/central region of the glacier tongue, not the speed-up itself. This has now been changed throughout the manuscript to better convey this, and hopefully this will make more sense in relation to the velocity maps presented in Figure 5.
It was challenging to convey all the information regarding the velocities in one figure, so three sequential velocity maps were chosen for Figure 5 that best represented the speed-up in the early-melt season. The reviewer's comment highlights that the images used in Figure 5, along with descriptions of the velocity event, did not effectively convey this. To rectify this, Figure 5 has been amended to include six sequential velocity maps to better show the nature of the early-melt season event. It is understood that plots showing relative changes in velocity would also convey this, however the authors feel that this is good opportunity to showcase more of the velocity maps produced from the CRIOS project.

4. Assertions made in the discussion should be backed up with data and results. For example the suggestion that tides influence the timing of plume pulses (but there are many other similar examples as detailed in the specific comments below).

Within the discussion section, events observed over the 2014 melt season are summarised and potential processes driving these events are proposed. Hydrological processes are largely discussed because subglacial hydrology is the main focus of the paper, and there is sufficient evidence from the data to suggest that these events are hydrologically driven to some extent. Other processes are also outlined, as recommended by the reviewer, such as oceanic influences and glacier dynamics (see the first comment in this response for more details).

The reviewer suggests that data (calving rate, tidal level, fjord temperature etc.) should be used to support assertions and explore these alternative processes in this study. The authors believe that this is beyond the scope of the study. The inclusion of other datasets would detract from the manuscript's primary focus on examining subglacial hydrology at a tidewater glacier. We intend to write a second paper at a later date which looks more closely at the dynamics of Kronebreen, specifically exploring calving dynamics in relation to oceanic forcing and glacial influences. We hope that the ideas
presented here are further explored in this future work.

We recognise that it is valuable to outline these additional influences though, as discussed in the first comment of this response. To address this, alterations have been made to the manuscript to emphasise where assertions have been made and where additional datasets are needed. In addition, each of these instances are stated as good ideas for potential work in the future. Specific changes are detailed in the subsequent comments.

Page 2, Line 3: Data are plural. 'data is' should be 'data are'.

Agreed. Change made.

Page 2, Line 16: This strikes me as a bit of a strange statement; what is a 'pressure environment'? Presumably you mean basal water pressure and effective pressure?

Here, 'pressure environment' is used as an encompassing term for basal water pressure and effective pressure. As this is not clear, the sentence has been changed for better clarification.

Page 2, Line 20: How rapid? Maybe better to specify several hours etc.

The rapid changes outlined here refer to processes such as abrupt channel collapse and extensive hydraulic overhauls in the subglacial zone. These changes commonly occur between 0 and 12 hours. This has now been specified in the manuscript.

Page 2, Line 23: Presumably the statement about 'inefficient evacuation of meltwater' relates to the fact that the basal hydraulic system at the terminus is not at atmospheric
pressure so the along-glacier hydraulic gradient is less steep? It would be worth clarifying this.

The reviewer is correct, 'inefficient evacuation of meltwater' in the near-terminus area of a glacier is related to the area not being at atmospheric pressure. Therefore the along-glacier hydraulic gradient is small. To clarify this, we have included greater detail in the manuscript:

'It has also been attributed to basal hydraulic systems which are not operating at atmospheric pressure, such as lake-terminating and tidewater glaciers, and there is an inefficient evacuation of meltwater because the hydraulic gradient is small (Sugiyama et al., 2011).'

Page 2, Line 26: In terms of land-terminating glaciers on ice sheets, this description is only really applicable to the marginal 20 km or so. Further up-glacier, maximum ice flow typically occurs later in the overall melt season (i.e. later than the onset of melting at the margin), once melt has commenced at higher elevations. It may be worth specifying that you are referring to land-terminating valley glaciers here?

The authors agree with the reviewer that the influence of basal water-pressures on glacier dynamics can be more accurately attributed to land-terminating valley glaciers. Change made.

Page 2, Line 28: What exactly do you mean by 'subglacial drainage re-organisation'?

Subglacial drainage re-organisation refers to the changes in the network of channelised and distributed pathways that are typically observed at the beginning and end of the melt season. This has now been added to the text for clarification:

'Ice velocities stabilise or fall later in the melt season in response to subglacial drainage
re-organisation (i.e. changes in the network of channelised and distributed drainage pathways at the beginning and end of the melt season) and the establishment of efficient channels that reduce water-pressure at the bed (Iken and Truffer, 1997; Hewitt, 2013).'

Page 2, Line 30: 'Ice velocity records indicate similarities to land-terminating glaciers'. Do you have a reference for this statement? And do you mean the terminus region of tidewater glaciers of further inland?

This section has now been significantly changed to address one of the reviewer's subsequent comments (second to last comment in this response) concerning the inclusion of the study by Doyle et al. (2015).

Page 3, Line 6: 'The drainage of supraglacial lakes provides an additional meltwater input into the subglacial environment'. The use of 'additional' here is a little odd - additional to what? I assume you mean in addition to the drainage of surface meltwaters before they accumulate into lakes, but this is not obvious from the previous paragraphs.

The reviewer is right in assuming that by 'additional' we mean that the drainage of supraglacial lakes is an input of meltwater in addition to surface melt production that immediately enters the snowpack/englacial zone. A sentence has been added to the beginning of this paragraph to clarify this:

'Meltwater typically enters the subglacial environment from the glacier surface via surface melt production. The drainage of supraglacial lakes provides an additional meltwater input into the subglacial environment.'

Page 3, Line 15: You need to make clear that this phenomenon has been observed in
the terminus region (last 20 km) of a single tidewater glacier in Greenland. As written, you risk suggesting that such drainage is as common or prevalent as the up-glacier progression of lake drainage with time, which is not the case.

The authors agree that observations of lakes that drain in a down-glacier progression are uncommon in comparison to observations of upglacier-propagating lake drainage.

Changes have been made to the manuscript to clarify this:

'On the contrary, lakes have also been observed to drain in a down-glacier progression, albeit such instances are less common (e.g. Everett et al., 2016).'

Page 3, Line 18: 'terminus' should be 'near-terminus'.

Agreed. Change made.

Page 3, Line 16: What exactly are 'Subglacial transient pressure waves'? How do these control up-glacier progression of lake drainage? This requires more explanation and a reference.

The term has commonly been used to describe events where high-pressures propagate through the subglacial zone of a glacier due to high pressure gradients. They have been associated with surges (Kamb et al., 1985) and have been used to propose an alternative explanation to hydrofracturing for the fill/draining of supraglacial lakes (Everett et al., 2016). However, the term can also lend itself to instances where low-pressures propagate through the subglacial zone of a glacier.

Reviewer 1 and reviewer 2 have both mentioned that the use of the term "subglacial transient pressure wave" is convoluted and it appears that this may be misinterpreted by the reader. For this reason, the term has been omitted from this paper. The term is largely used to describe the events at the beginning of the melt season. It has now

TCD
been replaced with better details concerning the glacier-wide drawdown of meltwater in the near-terminus area.

Page 3, Line 21: What is 'the bed system'? Better to avoid these vague terms and simply describe what you actually mean, which you do in the next-but-one sentence. Also, I'm not quite sure what the point of this sentence is 'This has largely been studied in inland and near-terminus settings'. Are there other 'settings' from which such observations are missing?

The bed system refers to the subglacial basal dynamics in a large area of the glacier, in this case being at glacier margins. Bed system has been changed to 'bed' to be more direct. The sentence that states 'This has largely been studied in inland and near-terminus settings' has been removed from the manuscript.

Page 3, Line 23–24: 'long periods of time'. Be more specific - days, weeks, months, years, decades? I assume decades based on the Tedstone reference but this would not be obvious to someone unfamiliar with the literature.

In this context, 'long periods of time refers to decadal time frames, as correctly pointed out by the referee. The sentence has been changed to clarify this.

Page 3, Line 26: How do these 'long residence times' tie in with observed reductions in ice surface uplift and velocity about 24 h after a lake drainage?

The 'long residence times' are in reference to the water pockets that have been observed and modelled (e.g. Dow et al., 2015), whilst the ice surface uplift is in reference to how rapid lake drainage can cause ice surface uplift over much shorter timescales (seen by Stevens et al., 2013). The sentence has been re-written to better distinguish Interactive comment

these two aspects:

'This has been observed to cause ice to uplift from the bed over short (0–48 hours) time periods (e.g. Stevens et al., 2013), and is likely to form subglacial water pockets or "blisters" with relatively high residence times over longer (days to weeks) periods (e.g. Dow et al., 2015).'

**Page 3, Line 27: Would not injection of water into a distributed system or a small channel also be capable of causing uplift?**

It is unclear what the reviewer is particularly questioning here. Yes, the injection of water into a distributed system or a small channel would indeed be capable of causing uplift. This idea has been conveyed with regard to below thick ice in the interior of an ice sheet (see previous amendment, Page 3, Line 26). It is assumed that the reviewer would also like this to be clarified at glacier outlets of ice sheets and near-terminus settings. Uplift has also been observed in these settings and this has now been added to the beginning of the paragraph where these settings are discussed (Page 3, Line 23):

'For instance, a rapid input of meltwater has been observed to cause localised uplift of the ice surface, and has also been observed to make a channelized system become more efficient...'

Page 3, Line 28: I think there are too many 'systems' mentioned in these introductory sections. It gets a bit confusing after a while.

The authors understand that this may confuse the reader at times. Where possible, the word 'system' has been replaced with an appropriate alternative in the introduction and background sections. There are still one or two 'systems' in relation to the drainage
system of a glacier, which is unavoidable, but the majority have been changed.

Page 4, Line 5: I think you need to note here that Slater et al (2017) used photographs from a time-lapse camera rather than satellite images. Also, time-lapse is also temporally intermittent - it is the frequency of that intermittency that is important!

Slater et al. (2017) used time-lapse imagery to distinguish plume activity, as did Schild et al. (2016) who used this in conjunction with MODIS satellite imagery. To better distinguish this, references to plume detection from time-lapse imagery have now been distinguished from those referring to plume detection from satellite imagery.

Time-lapse imagery provides high-temporal resolution data, but it is agreed that this is dependent on equipment that does not malfunction and, in some cases, re-visiting the field to download data. This means that, like satellite imagery, time-lapse imagery is also intermittent. This has now been acknowledged in the manuscript.

Page 5, Line 5: This sentence is convoluted: 'Kronebreen retreated 1 km between 2011 and 2016' would say the same more succinctly.

The sentence has been changed to convey that Kronebreen has recently undergone rapid retreat:

'Kronebreen is currently in a period of rapid retreat, having retreated 1 km between 2011 and 2016.'

Page 6, Line 7: The GPS position errors seem quite large. Why not post-process them using for example TRACK software? Also, it is not mentioned anywhere how these positional errors affect your measurements, of, for example, supraglacial lake area.
The camera sites were occupied for a minimum of 15 minutes, and post-processed using Trimble Business Centre software. The position errors are larger because of the lack of satellite passes in Svalbard (this is a problem we have each time that we survey our camera sites). Errors would likely be just as large even if they were post-processed using another piece of software. We have rectified this in future work by occupying camera sites for longer, to ensure that we gain adequate satellite coverage.

These positional errors introduce uncertainty into the measurements of supraglacial lake area and plume area. A sensitivity test was carried out on a subset of the supraglacial dataset to examine the uncertainty range. This was undertaken by simulating changes in the camera position, and examining changes in the lake areas based on georectification using these altered parameters. Changes in the vertical position of the camera ( $\pm 1.92$  m) introduce the greatest uncertainty as expected, with an uncertainty range of  $\pm 5\%$ . Changes in the horizontal position of the camera ( $\pm 1.15$  m) introduce an uncertainty range of  $\pm 0.6\%$ .

The authors believe that these errors are relatively small. The main aim of the plume and supraglacial lake areas is to show relative changes in two key signals of subglacial hydrology over the 2014 melt season. As shown in this manuscript, key findings are drawn from the timing of relative changes rather than absolute values. Therefore absolute values are not essential, and the authors argue that the uncertainty ranges calculated here do not need to be added to this manuscript. If the reviewer has good reason as to why they need to be included then the authors are happy to further discuss this.

Page 6, Line 10: 'data was' should be 'data were'. Agreed. Change made.
Page 6, Line 16–17: How did you avoid falsely identifying shadows and sediment-rich ice as lakes (both also have a high contrast with bare ice)?

The imagery went through a number of enhancements and processing steps to reduce the chance of misidentification:

- The images were masked to the specific area where lakes are visible. This reduced the chance of classifying other artefacts as lakes and also quickened processing time.
- The images are enhanced to better distinguish blue colours and ensure that 'lakelike' objects are distinguished. This filters out a large majority of noise. However, this also does mean that re-frozen lakes are often undistinguished (as clarified in the manuscript).
- These 'lake-like' objects are manually verified in PyTrx to filter out falsely detected lakes, such as shadows and sediment-rich ice, and only retain real lakes.

This information has been added to the paragraph to better inform the reader about PyTrx and the process of detecting lake from time-lapse imagery. Also, as outlined in the manuscript, sequential image sets with changes in illumination and shadowing are unavoidable. To counteract this, we detected lake surface areas in images from every half-hour and then took a moving-average of these measurements. This effectively "flattened" any noise introduced by these factors.

The manuscript has now been changed accordingly:

'These lakes were automatically detected from images based on the high contrast in pixel intensity between the ice and water surface. This process involves multiple steps to reduce the change of misidentification: (i) the images were masked to reduce the chance of misidentifying lakes; (ii) the images were enhanced to better distinguish blue
colours and ensure that 'lake-like' objects were distinguished; and (iii) these objects were manually verified in PyTrx to filter out falsely detected lakes such as shadows.'

TCD
The reviewer is correct, there is a word missing from the sentence. The word 'surveys' has been added.

Page 8, Line 32: Seems like a word is missing before 'which'. Could it be 'campaigns'?

Page 9, Line 8: 'but more central and nearer to the glacier's central flow line' should be 'but nearer to the glacier's central flow line'

Agreed. Change made.

Page 9, Line 21–23: Both sentences are pretty much repeated from the first paragraph of this section.

The authors agree that there is repetition in this paragraph. Repeated sections have been merged together into the first paragraph.

Figure 2: It would be useful to also put the dates along the top x axis.

Agreed. Change made.

Page 11, Line 6–7: Or more accurately, fill enough that water is visible in the TL imagery...

The authors agree that this is a more accurate phrasing and has thus been changed in

Page 11, Line 22–23: 'Modelled melt production has a diurnal pattern with a maximum in the day and minimum at night' – I think this is well enough known that it is not necessary to report here.

Agreed. Sentence has been deleted from the manuscript.

Page 11, Line 27: The bracket after the units shouldn't be superscript.

Agreed. Change made.

Page 11, Line 29–31: Some repetition from previous paragraph re diurnal variations etc.

Whilst the authors agree with the reviewer that there is repetition, a broad description is needed to illustrate the dataset to the reader. For this reason, obvious notes about the diurnal pattern have been removed from the manuscript. This has been replaced with a brief description of the diurnal range that is visible:

'...coinciding with the drainage of Lake Cluster 1 and the activation of the meltwater plumes. From this point, melt and runoff regularly reaches 20–26 m3 s-1 in the day and between 0–3 m3 s-1 at night. Towards the end of August...'

Page 12, Line 6: It would be informative to outline these on the velocity plots in Figure 5.

The locations of two of the ROI's (the supraglacial lakes and the borehole site) are
already included on both Figure 1 (the site map) and Figure 6 (the hydraulic potential modelling). The inclusion of these locations on the velocity plots in Figure 5 would merely be repeating information already relayed elsewhere.

In addition, Figure 5 already has a lot of information on it and the inclusion of the ROI locations would detract and/or obscure the information presented.

For these reasons, the authors have decided not to include the additional information in Figure 5.

Page 12, Line 16: Based on Fig. 2, it seems like the melt season lasts through until midway through September. I therefore do not think it is correct that the velocities were consistent for this period.

Agreed. This has been corrected to '...till the end of August...'

Page 13, Line 3: They definitely coincide (inasmuch as the temporal frequency of the TSX data allows), so do you mean 'possibly caused by'?

Agreed. The sentence has been changed to the wording suggested by the reviewer.

Page 13, Line 3–4: This is a bit vague. Ok, at the near-terminus centreline, velocities are higher in September than in the pre-melt season period, but at the other two ROIs they are broadly the same as before the melt season.

The reviewer is correct. Velocities remain high at the near-terminus centreline, but velocities from the other two ROI's remain constant. This has now been clarified in the manuscript in order to provide better detail:

'Whilst velocities remain constant at the lake and borehole ROI's through the rest of
September, high velocities persist at the centreline ROI and they do not return to premelt season conditions.

Page 13, Line 7 : But despite this, there is actually very little variation in borehole water pressure (as shown in fig. 2). Perhaps the borehole is actually not that well connected to the regional basal hydrological system?

This was also remarked by reviewer 1. The authors believe that the introduction of this hypothesis early on in the results section would be inappropriate. The results section should be reserved for presenting data with broad descriptions to convey key observations. However, a paragraph has been added to the discussion section (7.4 Subglacial drainage of Kronebreen) to discuss this idea that the borehole may not be well connected to an active, efficient drainage system.

Page 14, Line 26: 'supraglacial lakes' should be 'supraglacial lake area' Agreed. Change made.

Page 16, Line 11 (and throughout): 'upward-propagating' should be 'upglacierpropagating' (otherwise suggests vertical propagation).

The authors agree that the term 'upward-propagating' is vague and could infer vertical propagation rather than upglacier propagation which is the intended description. All references to 'upward-propagating' processes have now been changed to 'upglacier-propagating', including the title for Section 7.2 (Upglacier-propagating supraglacial lake drainage) and the several instances that the term is used in the interpretation and discussion sections.

TCD
Page 16, Line 18–19: 'This implies that meltwater is present at the bed and is enhancing basal lubrication '. Why is this necessarily the case? You should refer to another dataset - e.g. basal water pressure etc. to support this statement. What is to say that the acceleration is not due to a reduction in buttressing at the calving front?

The surface velocity of the glacier begins to gradually increase from 10 June, based on the velocities from the ROI's (the centreline, the region of the supraglacial lakes, and the borehole site). This is likely to be associated with the presence of meltwater at the bed, enhancing basal lubrication, based on observations of the on-set melt production and the filling of the supraglacial lakes. Assuming these lakes are connected to the bed at this point in time, these lakes are an indication of a build-up of meltwater in the subglacial zone.

Other causes for the speed-up are not explored because the main focus of the paper is to examine subglacial hydrology and its influences on dynamics at a tidewater glacier. Interpretation is limited to the datasets that we obtained in this study. It is unlikely that the acceleration is associated with a reduction in buttressing at the calving front because substantial sea ice rarely forms in Kongsfjorden given the year-round input of warm, saline Atlantic water. Marine influences (such as submarine melting and the seasonal increase in fjord temperature) could play a role in this speed-up, but it is beyond the scope of this study to fully explore this idea with additional datasets.

The authors believe that meltwater presence at the bed is a key influence in the speedup at the beginning of the season based on the evidence we present in this paper. We understand though that different, alternative hypotheses should also be outlined here to convey to the reader that there are many potential influences on glacier speed-up events. Other influences (including marine influences) have now been outlined in this section as possible additional contributing factors. It has also been noted that it is difficult to examine these influences in this manuscript with the data available:

'There are several likely reasons for this. Marine influences could play a key role, such
as submarine melting and the seasonal increase in fjord temperature, which could be causing changes at the terminus that reduce back-stress further upglacier and enable glacier flow. It is difficult to examine these influences here with the data available, but could be better examined in future work. Another influence is the presence of meltwater at the bed, which enhances basal lubrication and enables sliding...'

**Page 16, Line 21–22 : And all other glacier catchments assuming a normal lapse rate...**

The authors agree that meltwater could have originated from other glacier catchments, in addition to higher elevation on Holtedahlfonna. This has been added to the sentence:

'This meltwater may have originated from higher elevations and/or other glacier catchments, but it is unlikely given that early-season melt production is understood to first originate from the lower elevations of this glacier catchment (Van Pelt and Kohler, 2015).'

**Page 16, Line 23: So do you think that basal frictional melt is an important factor for accounting for the remaining meltwater at the bed?**

Basal frictional melting could account for the remaining meltwater at the bed, given the little surface meltwater inputs predicted by the runoff model. Modelling attempts suggest that changes in basal friction is evident in the near-terminus area of Kronebreen, but this is not as strong an influence on surface velocities in comparison to changes in surface runoff. This idea comes from a paper currently in review, which models conditions over the same time period to our study:

Vallot, D., Pettersson, R., Luckman, A., Benn, D.I., Zwinger, T., van Pelt, W.J.J., Kohler, J., Sch'afer, M., Claremar, B. and Hulton, N.R.J.: Surface changes influence on spatiotemporal variations of basal properties for Kronebreen, Svalbard, Geophys. Res. Lett.,
In review.

The reviewer has made a key point that basal frictional melt should be considered as an alternative hypothesis for why there is meltwater present at the bed in the early part of the 2014 melt season. The authors feel that it is appropriate to present this hypothesis in the discussion section (Section 7.1: Early melt season meltwater storage) rather than in the interpretation as this is where hypotheses are first introduced to explain this. A sentence has been added to provide the alternative hypothesis that basal frictional melting could also be a contributing factor to the presence of meltwater (Section 7.2, second paragraph, first line):

'This implies that water is either being generated at the bed, or it is bypassing storage in the snowpack and firn layer. Basal frictional melting could play a role in the generation of meltwater at the bed, but modelling of the Kronebreen's basal properties suggest that surface runoff is more likely to be the key influencing factor (Vallot et al., In review.) The lower area of the glacier tongue is a heavily crevassed surface...'

Page 16, Line 25–26: No, it only indicates that theoretically this is the expected route of subglacial water, not the configuration of the drainage system.

The presence of a surfacing plume has previously been associated with active channels, based on modelling and observations which suggest that plumes fed by distributed drainage pathways are less likely to surface in a fjord (Slater et al. 2017). Here, we also suggest that the presence of a surfacing plume infers that there is at least one active channel is active below the north region of the near-terminus area of Kronebreen.

The authors appreciate that the wording of the sentence may be misleading and the reader may gain the impression that a larger-scale drainage system is being inferred by the presence of surfacing plume activity. The sentence has been changed to better

TCD
describe that plume observations can be linked to the presence of at least one active basal channel:

'The continuous presence of a plume at the north side of the terminus (N1) suggests that a channel is established here from 25 June (Fig. 2C).'

Page 16, Line 30–31: Can you not easily quantify this from your hydropotential analysis? This would be a useful addition.

The reviewer is questioning whether the proportion and/or likelihood of meltwater routed through the north region of the glacier tongue can be quantified. The authors are unsure in how this could be usefully quantified. It is evident in Figure 6 that the channels on the north side of the terminus cover a significant portion of the Kronebreen/Holtedahlfonna catchment, and therefore it is logical to assume that they drain a large proportion of the glacier catchment. This section has been left unchanged for now, but we are happy to explore this further if the reviewer can specify how we could go about quantifying this.

Figure 6: Based on the size of the subglacial catchments (N bigger than S), why is the southern plume expression so much more extensive? What date is the plume extent from?

Plume activity was observed from one of the time-lapse cameras positioned on Colletthøgda (camera site 1 in Figure 1). The plume/plumes on the north side of the terminus (N1, N2 and N3) were clearly visible from this camera position (as seen in Figure 5). However, the plume on the south side of the terminus (S1) was much further away which made it challenging to distinguish its shape accurately. Adequate surface areas could not be derived from the plume on the south side for this reason, as stated in the methodology section (4.1.2 Visible meltwater plume extent). Plume S1 activity could TCD
still be observed though, hence it was included in the plume activity timeline shown in Figure 2.

For the purpose of creating Figure 6, shapes representing the surface expressions of each of the plumes had to be derived from different days because there was never an instance where all plumes were active simultaneously. Listed below are the dates and times for each of the plume shapes shown in Figure 6:

- Shape for Plume N1 taken from 11/07/2014 08:00
- Shape for Plume N2 taken from 11/07/2014 08:00
- Shape for Plume N3 taken from 20/07/2014 10:00
- Shape for Plume S1 taken from 16/09/2014 04:00

An effort was made to acquire shapes for Plume N1, N2 and N3 from approximately the same period of time for use in this figure. Shapes representing the surface expression of Plume S1 were taken from the day that the plume expression was most apparent to ensure that we were exhibiting an accurate surface expression. This happened to be on a day where Plume S1 was significantly larger than the surface expressions for Plumes N1, N2 and N3.

The authors understand that this could mislead the reader to believe that Plume S1 is more extensive than the other three plumes. Information has been added to the caption of Figure 6 detailing the date-time of each plume expression, and to explain that the expression of Plume S1 is not representative of its expression over the entire melt season:

'Potential subglacial water pathways at Kronebreen, as calculated from a scenario where hydraulic potential is governed by ice-pressure gradients (i.e. the cryostatic pressure ratio, K, is above 0.6). The surface expressions for Plume N1 and N2 are
taken from 11 July 08:00, Plume N3 is taken from 20 July 2014 10:00, and Plume S1 is taken from 16 September 04:00. The expression of Plume S1 is, on average, smaller than the expression shown here. This expression was chosen because is is the most accurate shape of the surface expression that could be acquired during the monitoring period. The base map is a Landsat image (taken on 11 June 2014) overlaid with bed elevation and corresponding contours at 50 m intervals.'

Page 17, Line 3–4: Might this indicate that the drainage is linked to a perturbation at the calving front? For example, an acceleration and consequent longitudinal stretching related to a calving event or break-up of seasonal cover adjacent to the glacier?

This comment is related to ideas concerning whether calving is predominantly the cause or result of changes in glacier dynamics (i.e. acceleration/deceleration). Although we understand a considerable amount about calving dynamics, there remain uncertainties in this master-slave theory that need to be addressed.

In specifically investigating subglacial hydrology and glacier dynamics, the authors appreciate that the ideas presented in this section promote a scenario where glacier dynamics are the predominant cause of changes at the glacier front. The reviewer is right to point out that the change in glacier velocity (and the coinciding lake drainage) could also be caused by changes at the terminus (such as an increase in calving). By presenting this alternative scenario, we present a balanced outlook of the master-slave theory concerning calving, and also convey to the reader that there are many different scenarios that could explain the events observed in the early-melt season at Kronebreen (the lake drainage, the activation of the main plume, the drop in basal water-pressure, and the gradual increase in surface velocity).

A paragraph has been added to this section to better outline other scenarios concerning the events that happen at the beginning of the melt season, including a scenario where these events are linked to increased calving activity and consequent longitudinal TCD
**stretching**

*Page 17, Line 4: 'upglacier-propagating'* Agreed. Change made.

Page 18, Line 8: What exactly does 'spatially discrete' mean in this context?

The sequential velocity maps show that this speed-up is confined to a specific area of the near-terminus zone. The phrasing 'spatially discrete' here refers to the confined nature of the speed-up to the south region of the glacier tongue. This phrasing has now been changed in the manuscript to better describe this:

'The sequential velocity maps show that this speed-up propagates  $\sim$ 4 km upglacier between 31 May – 16 August. Surface velocities appear to be highest to the central/southern region of the glacier tongue, with some of the north region only reaching velocities between  $\sim$ 1.6–2 md-1 (Fig 6).'

Page 18, Line 23: But if the plume is periodically visible, does this not suggest that the basal water is also purged periodically? Do you see a difference in velocity between times when the southern plume is and is not visible?

The reviewer is correct in suggesting that periodic visibility of plume activity on the south side of the terminus may be linked to periodic release of basal water. This is discussed in greater detail later on in the discussion section (7.3 Controls on meltwater plume activity). The reviewer is suggesting that we compare the plume activity on the south side of the terminus with the velocities to examine links between the two datasets. Links could suggest that short-term variations in glacier velocity may be linked to this hydraulic pulsing.
Adequate comparison between velocity and plume activity is beyond the scope of this study because of the temporal resolution of the datasets. As outlined in the manuscript, the plume on the south side of the terminus surfaces for short phases every 5 days on average. The velocities were acquired from TerraSAR-X imagery, with daily velocities averaged from feature-tracking between image pairs that were 11 days apart. Therefore we cannot examine short-term changes in velocity and relate these changes to plume activity.

Although we cannot investigate this idea within this study, a remark has been added to the discussion section (7.3 Controls on meltwater plume activity) to illustrate the potential for further work (Section 7.3, paragraph 3, sentence 5 and 6):

'This pulsing could be the cause of short-term changes in glacier dynamics in the nearterminus area, such as basal sliding and velocity. Although this idea cannot be further explored here, the examination of glacier dynamics in relation to plume presence could be a promising area for future studies.'

Page 20, Line 14–15: Could it not be that the volume of meltwater is just insufficient for the plume to either reach or be visible at the fjord surface?

The reviewer is correct in suggests an alternative explanation for the surfacing of the plume on the north side of the glacier at the beginning of the season. This has now been added as a possible explanation to the section:

'The activation of the main plume on the north side of the terminus (N1) suggests that either a sufficient volume of meltwater is being discharged to surface in the fjord, or an efficient system is established to evacuate meltwater on 25 June. If the second of these instances is true then meltwater is stored at the bed for  $\sim$ 15 days before it is evacuated, based on the timing of the onset of the speed-up and the activation of Plume N1.'
Page 20, Line 18: 'upglacier-propagating'

Agreed. Change made.

**Page 21, Line 15: 'unexpectedly absent during periods of high runoff'. Suggesting more distributed outflow of meltwater at the grounding line?**

The authors agree with the suggestion that activity of the plume on the south side of the terminus is intermittent and unexpectedly absent during period of high runoff possibly because a distributed drainage network presides. This information has now been added to the manuscript:

'The activity of this plume is intermittent and it is unexpectedly absent during periods of high runoff, suggesting that the outflow of meltwater is not channelised and is instead more distributed at the grounding line. '

Page 21, Line 18: 'varies over only a small range' (it does vary).

Agreed. Change made.

Page 21, Line 22–23: 'The precise timing of each outflow is possibly controlled by marine dynamics such as tidal level.' Do you see any evidence for cyclicity on the frequency of tides? Otherwise, what evidence is this statement based upon?

Further exploration of the influence of tides on outflow is not presented here because it is beyond the scope of the study (see major comment 4 for more details on this). It has been emphasised that this passage is proposing mechanisms for hydraulic pulsing at tidewater glaciers and this could be an interesting focus for future work. A sentence has been added to paragraph to convey this:
'The precise timing of each outflow is possibly controlled by marine dynamics such as tidal level. Although it cannot be further explored here, this could be an interesting focus for future work.'

Page 21, Line 25–26: 'The trigger for the release of this water could be related to this hydraulic pulsing'. This seems a little too speculative.

As stated in the previous comment and in the major comments section, further exploration of the causes of hydraulic pulsing is beyond the scope of the study. Ideas are merely proposed here and it has previously been emphasised that these ideas provide motivation for future work. Two sentences have been added to the end of the paragraph to emphasise this:

'The trigger for the release of this water could be related to this hydraulic pulsing. This pulsing could be the cause of short-term changes in glacier dynamics in the near-terminus area, such as basal sliding and localised speed-up events. Although this idea cannot be further explored here, the examination of glacier dynamics in relation to plume presence could be a promising area for future studies.'

Page 22, Line 11: 'The key difference at Kronebreen, and other tidewater glaciers, is the high hydraulic base-level' should be 'The key difference at Kronebreen, and other tidewater glaciers, compared to land-terminating glaciers, is the high hydraulic base-level'

This sentence was drastically modified based on the recommendation of referee 1. No further changes have been made.

Page 22, Line 15–16: You should say why 'a stable drainage system cannot exist in
Page 22, Line 29: What does 'This' refer to?

**this region' presumably because the high velocities preclude the formation of persistent**

Agreed. Sentence changed to:

channels?

'This permits fast flow, which precludes the formation of persistent channels in this region.'

Page 22, Line 24: What do you think causes the 'glacier-wide transient low-pressure wave that is initiated near the terminus', and what evidence do you have to support this assertion?

The chain of events recorded at the beginning of the 2014 melt season (the plume activation, upglacier-propagating drainage lakes, drop in basal water-pressure, and gradual increase in surface velocities) have been described in this section to set-up accessible comparison to other studies which examine subglacial hydrology at tidewater glaciers. Potential causes for these individual events have been proposed in earlier sections (i.e. Section 7.1: Early melt season meltwater storage, and Section 7.2 Upglacier-propagating supraglacial lake drainage), including alternative scenarios as recommended by the reviewer.

The authors feel that exploring causes of the chain of events here would merely be repeating what has previously been outlined. It would also detract from the main focus of this section of the discussion which is to compare the hydrological observations at Kronebreen with other studies, with particular interest in comparisons to other borehole records and comparisons to model simulations of subglacial hydrology. For these reasons, no changes have been made in regard to this comment.
TCD

The structure of this paragraph is a little muddled. The opening two sentences of the paragraph have been changed to give better context for the reader.

'The observations from the borehole water-pressure record are strikingly different from borehole records in Alpine settings. These usually exhibit a diurnal signal, which reflects changes in delivery of meltwater to the bed and creates transverse hydraulic gradients that make meltwater pathways highly changeable (Meier et al., 1994; Hubbard et al., 1995).'

Page 23, Line 8: 'due to the difference in ice thickness'. Be more specific, it is not just the difference, but the fact that the ice is shallower that is key here. Also, you should state that the thinner ice leads to slower creep closure rates, meaning that channels are easier to open and maintain.

Agreed. Detail has been added to better specify why the subglacial drainage network can reconfigure on the north side of the glacier tongue:

'It is likely that the subglacial network can reconfigure because the ice is shallower and thinner, compared to large ice sheet outlets. This also means that channels to be open for longer because the thinner ice promotes slower creep closure rates.'

Page 23, Line 10–11: You should be clear that you are referring here to the mean melt rate for the entire submerged ice front rather than the localised melt rate (which is likely to be greater for more spatially-focused discharge).

Agreed. Change made.

Page 23, Line 13: I'm not sure I agree with this statement. From Fig. 5 it looks like there is also a speedup at the northern part of the tongue. Could you provide a relative
change in velocity map to evidence your assertion?

Changes have been made to Figure 5 and related descriptions to better convey the nature of this speed-up event. For more details, see major comment 3.

For this instance, the sentence has bee modified to better describe the speed-up event:

'The seasonal speed-up observed at the beginning of the 2014 melt season shows that the highest velocities exhibited are within the central/southern region of the tongue of Kronebreen, as presented in the velocity maps in Figure 5.'

Page 23, Line 20: What exactly is mean by 'consistent' here? Spatially consistent (if so this is different to the results presented in this paper), temporally consistent?

The word 'consistent' refers to the temporal consistency in speed-up events year-onyear at Kronebreen. The word has been deleted from the sentence to avoid confusion.

**Page 23, Line 27–29: This sentence seems contradictory - please clarify...**

The authors agree that the first two sentences of this paragraph are convoluted. They have now been changed to better convey the idea that the 2014 melt season was abnormal for the dynamics of Kronebreen:

'We also argue here that 2014 is actually an abnormal year for the dynamics of Kronebreen, based on the observations of a speed-up at the end of the melt season (Luckman et al., 2015; Vallot et al., In review). It is likely that the speed-up at the end of the 2014 melt season was caused by an unprecedented high rainfall event that overwhelmed a subglacial drainage system in a late-season phase with low efficiency.'

Page 24, Line 9: It might be worth also referencing Doyle et al. (2015), Nature Geo-

TCD
**science who saw a similar effect at a land-terminating glacier in west Greenland.**

The authors agree that the study by Doyle et al. (2015) is a key comparable study, especially the similarities in late-season speed-ups which are likely to be linked to the timing of high-rainfall events. We believe that it would be appropriate to introduce this study first in Section 2 (Background) of the manuscript, and then discuss it in greater detail in Section 7.5 (Discussion: Implications for subglacial dynamics).

A sentence has been added to Section 2 (Background): 'Precipitation can disrupt drainage due to the high influx of water over a short period of time, in some cases causing speed-ups due to the timing of high-rainfall events in relation to a melt season (e.g. Doyle et al., 2015).' (Section 2, paragraph 2, sentence 4)

Two sentences have been added to Section 7.5 (Discussion: Implications for subglacial dynamics): '...here we propose that this speed-up was largely caused by an unprecedented high rainfall event. Doyle et al. (2015) observed a similar event near to the end of the 2011 melt season at Russell Glacier. They suggest that such speed-ups are amplified due to their late-season timing.' (Section 7.5, paragraph 3, sentence 3 and 4)

The citation has also been added to Section 8 (Conclusion) in reference to observed high-rainfall events and speed-ups in the late part of a melt season.

Page 24, Line 27: Or indeed that it is something other than meltwater that triggers this initial speed-up - e.g. a calving event, break up of sea ice etc.

Other alternative scenarios have now been discussed in the interpretation and discussion sections, as recommended by the reviewer. The authors believe that there is no need to re-iterate them in the conclusions, as this would be merely repeating ideas outlined earlier in the manuscript. No change made.

TCD

---

## Referee Comment (RC3) · Anonymous Referee #3 · 12 Jul 2017

This paper outlines a detailed and complementary suite of observations and modeling of the subglacial hydrologic system and associated glacier dynamics of a tidewater glacier in Svalbard. Time lapse photogrammetry is used to observe, in high temporal detail, the location and patterns of supraglacial lakes and meltwater plumes adjacent to the calving front. A coupled energy balance and snow model was used to determine the production and timing of melt and runoff delivered to the subglacial hydrologic system. Glacier velocities were monitored in high spatial resolution using feature tracking

of repeat-pass TerraSAR-X images to determine larger scale patterns in glacier flow. Calculations of hydraulic potential are made to illustrate potential pathways of meltwater routing at the base of the glacier. Finally, water level and pressure measurements made at a single borehole site were collected and related to the evolution of the other hydrologic components of the glacier.

The study is well written, well organized, and thorough in its analysis. The complementary observations are used to illustrate the dynamics and evolution of the hydrologic system of Kronebreen, which evolves in a number of interesting ways, both spatially and temporally. The evidence is strong for the conclusions that the southern portion of the glacier tongue is more dominated by a distributed, pressurized hydrologic system, whereas the central/northern tongue receives more meltwater and has a more stable efficient drainage system. Both the delivery of meltwater to the base of the glacier and the organization of the subglacial drainage system need to be taken into account to explain the observed evolution of plumes and glacier speeds. This study will be a welcome contribution to the growing field of coupled glacier dynamics and hydrology.

I have only minor recommendations, questions, and comments on the manuscript:

p2, line 8: if such observations are rare, this suggests that there are some. Can you provide reference(s) here?

Figure 1: it seems that you could change the aspect ratio of the figure to zoom in more on the study area. Areas to the north and south of Kronebreen aren't really necessary to include, other than to make room for your inset panels.

P6, lines 9-10: it's not clear exactly what you mean here, or why this is necessary. Why is it necessary to smooth an initially high-resolution DEM, and what do you mean by "homogeneous surface"?

Melt modeling: what is the spatial resolution of the model? Is the model driven solely by the weather station data from Ny Ålesund? Are there any metrics of model validation/calibration that you can discuss?

P8, line 31: It sounds here as if you're using a different surface DEM from the Norwegian Geodetic Survey DEM that you discussed above with respect to the photogrammetry. Is this the case? If so, how different are the two DEM's you used? Why not use the same DEM throughout the study? For example, how different would the hydraulic potential calculations be if you used the Norwegian Geodetic Survey DEM instead?

Borehole GPS: it seems surprising (even if true) that the borehole GPS didn't add anything insightful. Why not show this and demonstrate that this is the case? It seems that you could overlay the GPS-derived velocity on Figure 2d to show this.

Figure 4 caption: panels are not numbered clockwise as indicated.

Section 5.6: it's not clear how you arrived at a value of k=0.6 as a sort of threshold for routing of meltwater between the northern and southern sections of the glacier. You state that "results suggest" this, but don't specifically describe why. "Several scenario were considered" (line 15), but what do you mean by this? How do you arrive at the conclusion that flow routing changes between a value of k=0.5 and 0.6 (line 19)? This seems different than what you describe in line 18 about threshold routing above and below a value of k=0.6? I guess I'm just a bit confused about this section, perhaps it's just a matter of describing more specifically what you've done here.

Borehole pressure: the pressure variations you record indeed seem to suggest that you are not actually located to a connecting channel. I would expect more pressure variations if you were. You seem to suggest that you might not be located at a channel, but argue that you are "near" a connecting channel if not connected to a channel that is consistently full of meltwater (in which case I would still expect to see more pressure variations).

Section 6.2: you describe here a cyclic pattern of the plume surface area, and suggest that it may be related to internal cycles of storage and release within the glacier. You

describe the pulsing as having a "duration" of 4-5 days, what do you mean by "duration"? Is there a particular period of the cycling? It seems that there could be sources of cyclicity in fjord circulation or tides which could also play a role in the patterns you see.

P19, line 16: here (and in other places) you describe "storage" of water at the base of the glacier. I think you should define what you mean by storage. Does this mean that you think the water is held stationary beneath the glacier, or rather that it is just inefficiently drained? You actually provide a bit of a definition on p20 lines 3-4, but it might be worth defining this sooner.

P21, lines 18-19: you say here that runoff has a diurnal signal, but then state that plume pulsing is independent of meltwater inputs. Related to comments above, I think you should describe the cyclicity of the plume pulses in a bit more detail to support the claim that this pulsing is not related to diurnal meltwater inputs (unless I've missed something here).

P21, line 22 (and elsewhere): you describe in numerous places that pressure forces a channel to open. However, the prevailing theories for meltwater channels is that they represent a balance between creep closure due to ice overburden pressure and melting caused by water flowing against the channel walls. So are you really claiming that you have something different than this at Kronebreen, some kind of elastic deformation at the base of the glacier from water pressure forcing channels to open?

P21, lines 21-26: you imply here that marine dynamics such as tides may play a role in the periodic flushing of meltwater. However, this claim is not supported by subsequent sentences that describe supraglacial lakes and velocity signals. So what is the evidence for your claim that marine dynamics plays a role?

P22, line 24: can you describe what you mean physically by a "transient low-pressure wave"? What would be the source of such a wave, and how would it originate at the terminus?

P 24, line 3: here you list melt and runoff under "measurements of hydrologic components" but these are actually model outputs.

---

## Author Comment (AC3) · 18 Jul 2017

We would like to thank the reviewer for their comments and positive response to our manuscript. We have edited our manuscript accordingly, including better definition of frequently-used terms and more detailed descriptions in places. We have also addressed the comment on the inclusion of the GPS velocity data with the figure included in the supplementary material associated with this response. We hope that the re-

viewer finds that all their comments have been carefully considered and thoroughly addressed.

Details of our response to the reviewer's comments are outlined below.

*p2, line 8: if such observations are rare, this suggests that there are some. Can you provide reference(s) here?*

There are a couple of these studies dotted as citations throughout the manuscript. We have now included these citations in this sentence:

' However, simultaneous measurements of all these manifestations of the subglacial system are rare (e.g., Kamb et al., 1994; Sugiyama et al., 2011).'

*Figure 1: it seems that you could change the aspect ratio of the figure to zoom in more on the study area. Areas to the north and south of Kronebreen aren't really necessary to include, other than to make room for your inset panels.*

The authors attempted to: 1) change the aspect ratio; 2) move the inset panels; and 3) zoom into Kronebreen. We found that numerous problems occurred when trying to accomplish this. Primarily, the Landsat image becomes pixelated and coarse when we tried to zoom into our field site, which isn't as visually pleasing. It was also difficult to move the inset panels without covering the plume extents or camera positions, even when we had changed the aspect ratio. Equally we felt that it was valuable to include some of the fjord and neighbouring glaciers (Kongsvegen and Kongsbreen) as context for the reader.

For these reasons, the authors have decided not to change Figure 1.
*P6, lines 9-10: it's not clear exactly what you mean here, or why this is necessary. Why is it necessary to smooth an initially high-resolution DEM, and what do you mean by 'homogeneous surface'?*

The DEM used in this study was obtained from airborne photogrammetric surveying in 2008 by the Norwegian Geodetic Survey. This DEM is older than the time-lapse imagery, which was acquired in 2014. Even at an initially high-resolution, this DEM has topographic features that represent the surface at the time of acquisition. These features are not present in the actual 2014 glacier surface. It was therefore smoothed to create a homogenous surface (i.e. flattened, without abrupt changes/artefacts) to better represent the glacier surface in 2014.

This information has now been added to these lines for better clarification: 'This DEM was smoothed using a linear interpolation approach to reduce discrepancies between the glacier surface in 2008 and in 2014. Data could thus be projected onto a homogenous surface (i.e. flattened and without abrupt changes/artefacts).'

*Melt modeling: what is the spatial resolution of the model? Is the model driven solely by the weather station data from Ny Ålesund? Are there any metrics of model validation/calibration that you can discuss?*

The melt model used in this study is the same one used in the Van Pelt and Kohler (2015) study. The model calculated melt/runoff in $100 \times 100$ m grids across the Kronebreen/Kongsvegen/Holtedahlfonna catchment. The model is primarily driven by the weather station data from Ny Ålesund (i.e. air temperature, wind speed), but elements of the model were initially calibrated with in-situ measurements. For instance, the subsurface model was calibrated with snow density pits, and the energy balance model was calibrated with flux measurements of incoming and outgoing shortwave and longwave radiation from radiometer measurements in the field.

Details about the model validation and calibration are fully outlined in Van Pelt and Kohler (2015). It has been made clearer in the manuscript that full details about the model can be found in this citation:

'Further details about the model, including model validation and calibration, are outlined in detail in Van Pelt and Kohler (2015).' (Section 4.3 Melt modelling, paragraph 1, last sentence)

*P8, line 31: It sounds here as if you're using a different surface DEM from the Norwegian Geodetic Survey DEM that you discussed above with respect to the photogrammetry. Is this the case? If so, how different are the two DEM's you used? Why not use the same DEM throughout the study? For example, how different would the hydraulic potential calculations be if you used the Norwegian Geodetic Survey DEM instead?*

Two different surface DEMs were used for the photogrammetric measurements and the hydropotential modelling. The Norwegian Geodetic Survey DEM (from 2008) was used for the photogrammetric measurements whilst the radar surveying DEM was used for the hydropotential modelling. The DEMs for the hydropotential modelling were acquired simultaneously in 2009–2010 and 2014–2016 and will form part of a study soon to be submitted:

Lindbäck, K., Kohler, J., Pettersson, R., Myhre, P.I., Nuth, C., Langley, K., Brandt, O., Messerli, A. and Vallot, D., In Prep. Subglacial topography, geology and future bathymetry of Kongsfjorden, northwestern Svalbard.

The surface and bed DEMs are exclusive to this study. They were only used for hydropotential modelling because they represent the surface and bed topography at the same period in time. Differences between the Norwegian Geodetic Survey DEM and the radar surveying surface DEM are relatively small, and thus there would be little difference in the hydraulic potential calculations if they were interchanged.

*Borehole GPS: it seems surprising (even if true) that the borehole GPS didn't add anything insightful. Why not show this and demonstrate that this is the case? It seems that you could overlay the GPS-derived velocity on Figure 2D to show this.*

This point was also noted by Reviewer 1. As stated in their response, the GPS data was not included in this study for three main reasons:

- The GPS velocity record is incomplete. The GPS was offline at the beginning of September 2014, whilst the rest of the dataset record carries on till the end of September 2014. The record duration is therefore mismatched.

- The higher temporal resolution of the GPS velocities does not appear to add anything new to the study. There were difficulties in processing the GPS data and short-term variations cannot be distinguished from the daily positions that we extracted. The dataset generally appears noisy. To resolve this and provide an alternative, velocities were derived from the TerraSAR-X imagery and then a spot velocity was extracted from the borehole site. These appear much less noisy and fit well with the rest of the 2014 record.

- The key findings from the velocity data focus on the spatial variability in velocity over the glacier tongue, rather than changes in velocity over time. These are better addressed with the TerraSAR-X velocities rather than the GPS velocities. The inclusion of the TerraSAR-X velocities from the borehole site are also consistent with the velocities derived from the other ROI's (i.e. from the centreline and the supraglacial lakes).

To better show the noise in the dataset, we have included Figure 2 as supplementary material which includes the GPS velocities in panel F (shown as the green plotted line) along with the TerraSAR-X derived velocities. It is clear from this that the GPS velocities do not add any additional information to the study, and any changes seen in velocities are difficult to associate with the other datasets.

[Figure]

For these reasons, the GPS data will not be included in this paper. The difficulties with integrating the GPS velocities has been clarified in the methods section of the manuscript (Section 4.4, page 8, line 11).

*Figure 4 caption: panels are not numbered clockwise as indicated.*

Agreed. The caption has now been changed accordingly:

'Meltwater plume scenarios from time-lapse imagery at Kronebreen. Top-left to bottom-right: 1) Surfacing meltwater plume from the main source on the north side of the glacier terminus, N1; 2) Sources from Plume N1 and Plume N2; 3) Sources from Plume N1 and Plume N3; 4) Plume N1 and Plume S1, the main source on the south side of the glacier terminus.'

*Section 5.6: it's not clear how you arrived at a value of k=0.6 as a sort of threshold for routing of meltwater between the northern and southern sections of the glacier. You state that "results suggest" this, but don't specifically describe why. "Several scenario were considered" (line 15), but what do you mean by this? How do you arrive at the conclusion that flow routing changes between a value of k=0.5 and 0.6 (line 19)? This seems different than what you describe in line 18 about threshold routing above and below a value of k=0.6? I guess I'm just a bit confused about this section, perhaps it's just a matter of describing more specifically what you've done here.*

Subglacial hydraulic potential was calculated primarily based on ice thickness and bed elevation. The crostatic pressure factor ($k$) is the ratio of water-pressure to ice over-burden pressure. Variations in the value of $k$ reflect the degree to which subglacial drainage is pressurised with $k$=0 reflecting open channel flow at atmospheric pressure, and $k$=1 reflecting pressurised flow.

We calculated subglacial hydraulic potential over several iterations, changing the value of $k$ each time. In total, we ran 11 simulations with the value of $k$ between 0.0–1.0 (i.e. hydraulic potential was calculated each time with a $k$ value of 0.0, 0.1, 0.2, 0.3, 0.4, 0.5, 0.6, 0.7, 0.8, 0.9, and 1.0). These are what are referred to in Section 5.6 as the 'several scenarios' that we considered.

We found that there is little change in the configuration of the channel network when $k$ is 0.0–0.5. In all these instances, the calculations suggest that a major channel connects Holtedahlfonna to the south region of the glacier tongue. There are significant changes in the channel configuration when $k$ is 0.6 and above (i.e. $k$=0.6–1.0). The major channel diverts to the north region of the glacier tongue in these scenarios. Therefore there is a significant difference when we consider hydraulic potential with a $k$ value of 0.5 and below, and 0.6 and above. This is what we refer to in the manuscript as the 'threshold' as it is apparent that this difference occurs between a $k$ value of 0.5 and 0.6.

The authors appreciate that some of the terms used in this section are not accurate and more appropriate, detailed wording could be used instead. We have changed the section accordingly to make this clearer:

'Several scenarios were considered in calculating the hydraulic potential at the bed of Kronebreen based on the $k$ value, which represents cryostatic pressure ratio (i.e. the extent to which meltwater routing is dictated by ice-pressure gradients). Subglacial hydraulic potential was calculated over several iterations, changing the value of $k$ each time. In total, we ran 11 simulations with the value of $k$ between 0.0–1.0 (i.e. hydraulic potential was calculated each time with a $k$ value of 0.0, 0.1, 0.2, 0.3, 0.4, 0.5, 0.6, 0.7, 0.8, 0.9, and 1.0).' (Section 5.6, paragraph 1)

*Borehole pressure: the pressure variations you record indeed seem to suggest that you are not actually located to a connecting channel. I would expect more pressure*

*variations if you were. You seem to suggest that you might not be located at a channel, but argue that you are 'near' a connecting channel if not connected to a channel that is consistently full of meltwater (in which case I would still expect to see more pressure variations).*

The reviewer is correct in stating that we would expect to see more changes in the water-pressure record if the borehole sensor was located in a connecting channel. This is now clearly stated in Section 7.4 (Subglacial drainage of Kronebreen) following similar comments from Reviewer 1.

We suggest that the borehole is possibly located near to an active drainage system based on instances where changes in pressure have coincided with other changes related to subglacial hydrology (e.g. the early-melt season 'flushing' event, and the significant pressure drop in September). This is also supported by the hydraulic potential modelling which indicates that the location of the borehole intersects with one of the main channels in the catchment. We propose that the borehole is located within the catchment of an active drainage system based on these arguments. Absolute changes in the water-pressure record suggest differently as noted by the reviewer.

Therefore we have two lines of evidence, with one suggesting that the water-pressure is indicative of an active drainage catchment, and the other suggesting that the record reflects an isolated, consistently pressurised region of the bed.

A paragraph has been added to better outline these ideas in Section 7.4:

'Few short-term pressure variations are observed in the water-pressure record from May–September 2014, apart from the significant drop in pressure at the end of the melt season. It is possible that the borehole is located on an area of the bed that is not well connected to an active, efficient drainage system. However, changes in water-pressure have been observed to coincide with other features in the hydrological system (i.e. plume activity and supraglacial lake drainage), which suggests that the borehole is hydraulically connected to some degree. This is also supported by the modelled

hydraulic potential, which indicates that the borehole is located close to, or possibly within, an efficient drainage catchment.'

*Section 6.2: you describe here a cyclic pattern of the plume surface area, and suggest that it may be related to internal cycles of storage and release within the glacier. You describe the pulsing as having a 'duration' of 4-5 days, what do you mean by 'duration'? Is there a particular period of the cycling? It seems that there could be sources of cyclicity in fjord circulation or tides which could also play a role in the patterns you see.*

From measuring the surface area of the plume expressions, we found that the plume surface expression fluctuates in size on a regular basis. This behaviour is repeated throughout August (08–28 August), and each fluctuation phase (i.e. a period of expansion followed by a reduction in surface area) has a duration of 4–5 days. This is what we refer to in the manuscript as a cyclic pattern, and we associate this with hydraulic pulsing.

Changes in surface melt and runoff appear to have little influence on this pulsing. The reviewer rightfully points out that the source of this cyclicity could be associated with fjord circulation and tides. From looking at the time-lapse images, the authors believe that wind direction may also play a significant role as the plume size can be affected by sea ice and icebergs that have been pushed towards the terminus.

These further details have now been added to Section 6.2:

'Plume activity at the north side of the terminus is persistent throughout August (Fig. 2B). The main plume (N1) is visible throughout, the secondary plume (N3) is present for most of the month (01–20 August), and the third (N2) is briefly active on 29 August. The total surface area/expression of these plumes fluctuates in size on a regular basis. This behaviour is repeated throughout August (08–28 August), and each fluctuation phase (i.e. a period of expansion followed by a reduction in surface area) has a duration of

4–5 days (Fig. 2C). Changes in surface melt and runoff appear to have little influence on this pulsing. This implies that there are additional controls on subglacial outflow. The source of this cyclicity could be associated with marine influences such as fjord circulation, tidal cycles, and wind direction. However, it is difficult to examine these influence here due to the limited datasets. Cycles of internal storage and release in the subglacial environment could also be a influence on subglacial outflow, which is possibly confined to the terminus zone because the signal is not evident higher up the glacier tongue in the water-pressure record from the borehole.' (Section 6.2, paragraph 2)

*P19, line 16: here (and in other places) you describe 'storage' of water at the base of the glacier. I think you should define what you mean by storage. Does this mean that you think the water is held stationary beneath the glacier, or rather that it is just inefficiently drained? You actually provide a bit of a definition on p20 lines 3-4, but it might be worth defining this sooner.*

'Storage' is used when describing inefficient drainage at the south side of the glacier tongue. In these cases, the storage of water is used as an encompassing term to outline that water is being inefficiently drained and, as a result, is likely to also be held stationary beneath the glacier.

Changes have no been made throughout the manuscript in instances where the 'storage' of water is described in relation to inefficient drainage at the south side of the glacier tongue. For each case, we have now added that meltwater at the bed is slow-moving and/or being stored. For example:

'It is likely that this meltwater is slow-moving and/or being stored, which would enhance basal lubrication and is a likely reason for high surface velocities in this region at this late stage in the melt season' (Section 6.3, paragraph 2, last sentence)

[Figure]

*P21, lines 18-19: you say here that runoff has a diurnal signal, but then state that plume pulsing is independent of meltwater inputs. Related to comments above, I think you should describe the cyclicity of the plume pulses in a bit more detail to support the claim that this pulsing is not related to diurnal meltwater inputs (unless I've missed something here).*

More detail has been added to Section 6.2 has stated in the previous related comment about cyclicity in the plume surface expression. This adequately conveys that the hydraulic pulsing observed in this record is not related to diurnal meltwater inputs. The sentence referred to in this comment is convoluted due to the mention of diurnal patterns in runoff. It has now been made clearer that this pulsing is independent of dirunal changes in runoff:

'It is proposed here that this plume activity is a signal for subglacial hydraulic pulsing. As the water level at the borehole site varies over only a small range (298–300 m), it is suggested that this pulsing is independent of meltwater inputs and is the result of processes confined to the near-terminus region (i.e. not glacier-wide).' (Section 7.3, paragraph 2, last sentence)

*P21, line 22 (and elsewhere): you describe in numerous places that pressure forces a channel to open. However, the prevailing theories for meltwater channels is that they represent a balance between creep closure due to ice overburden pressure and melting caused by water flowing against the channel walls. So are you really claiming that you have something different than this at Kronebreen, some kind of elastic deformation at the base of the glacier from water pressure forcing channels to open?*

Meltwater channels open and close due to two main processes: 1) when subglacial water has sufficiently melted the channel wall; and 2) when the overlying ice causes the channel to close. Channels remain open when these two forces are balanced, and open/close when one prevails over the other. These two processes have largely been

studied on land-terminating and alpine glaciers.

There is limited understanding about channel formation at tidewater glaciers due to the difficulty in obtaining direct observations. There are additional controls at tidewater glaciers, such as marine influences, which affect the pressure environment at the bed. This could promote elastic deformation at the base of the glacier which could 'force' a channel to open. Although these processes cannot be thoroughly examined in this study, the authors felt that it was necessary to differentiate the processes for channel opening and closure from those at land-terminating and alpine glaciers.

However, it is understood that channel meltback could also be an active process at Kronebreen and therefore should be outlined as a possible mechanism for channel opening. We have added this as a possible mechanism in each instance where we describe that pressure forces a channel to open:

'Hydraulic pulsing represents a periodic flushing of meltwater in the local vicinity, which occurs when sufficient pressure has accumulated to force a channel open and/or when subglacial water has sufficiently melted the cavity/conduit wall.' (Section 7.3, paragraph 3, first sentence)

This was previously recommended by Reviewer 2, in which all instances were changed. No further changes have been made.

*P21, lines 21-26: you imply here that marine dynamics such as tides may play a role in the periodic flushing of meltwater. However, this claim is not supported by subsequent sentences that describe supraglacial lakes and velocity signals. So what is the evidence for your claim that marine dynamics plays a role?*

The main aim of this manuscript is to better understand subglacial hydrology at a tidewater glacier from direct and indirect observations. The examination of marine influences, such as tidal level, in relation to the periodic flushing of meltwater is beyond

the scope of the study. Therefore it is difficult to support this idea with evidence, but the authors felt that it needed to be mentioned in order to represent alternative influences on subglacial hydrology and glacier dynamics. As recommended by Reviewer 2, the manuscript was changed to better outline that the examination of these alternative influences will not be further explored in this study, but could be a promising focus for future work. No further changes have been made following on from the changes recommended by Reviewer 2:

'Hydraulic pulsing represents a periodic flushing of meltwater in the local vicinity, which occurs when sufficient pressure has accumulated to force a channel open and/or when subglacial water has sufficiently melted the cavity/conduit wall. The precise timing of each outflow is possibly controlled by marine dynamics such as tidal level. Although it cannot be further explored here, this could be an interesting focus for future work.' (Section 7.3, paragraph 3, first line)

*P22, line 24: can you describe what you mean physically by a "transient low-pressure wave"? What would be the source of such a wave, and how would it originate at the terminus?*

The term has commonly been used to describe events where high-pressures propagate through the subglacial zone of a glacier due to high pressure gradients. They have been associated with surges (Kamb et al., 1985) and have been used to propose an alternative explanation to hydrofracturing for the filling/draining of supraglacial lakes (Everett et al., 2016). However, the term can also lend itself to instances where low-pressures propagate through the subglacial zone of a glacier.

All the reviewers have mentioned that the use of the term 'subglacial transient pressure wave' is convoluted and it appears that this may be misinterpreted by the reader. For this reason, the term has been omitted from this paper. The term is largely used to describe the events at the beginning of the melt season. It has now been replaced with

better details concerning the glacier-wide drawdown of meltwater in the near-terminus area.

*P 24, line 3: here you list melt and runoff under "measurements of hydrologic components" but these are actually model outputs.*

Agreed. The sentence has now been changed to:

'Subglacial hydrology has been examined at a tidewater glacier in Svalbard using direct measurements of the basal pressure environment in conjunction with measurements of hydrological components (supraglacial lake drainage, meltwater plume presence, and plume surface area), modelled components (melt, runoff, and hydraulic potential), and surface velocities derived from TerraSAR-X imagery.'

[Figure]

**Supplement:**

[Figure]

**Figure 2.** *Composite graph showing hydrological results from Kronebreen, including GPS velocities (in panel F) from the borehole site. A) Surface area of the three visible lake clusters (moving averages included); B) Timeline of the appearance of the four plumes, three visible at the north side of the terminus (N1, N2, N3) and one visible from the south side (S1); C) Total surface area of Plume N1, N2 and N3 (moving averages included), plus episodes when the plume extent is out of the image frame (noted as "max. plume extent"); D) Modelled melt (0–500 m elevation) and precipitation; E) Modelled runoff (0–500 m elevation); F) Glacier surface velocities, with spatial averages from the glacier centreline (<2 km from the terminus), the region of the supraglacial lakes, and the location of the borehole site, and GPS velocities from the borehole site. The faint area around each velocity line is the uncertainty range (<0.4 m/day); G) Water-pressure and corresponding water level from the borehole site.*

---

## Author Response (AR1)

**THE UNIVERSITY of EDINBURGH**
**School of Geosciences**

Penelope How
*School of Geosciences*
*University of Edinburgh*
*Drummond Street*
*Edinburgh, EH8 9XP*
*Phone: +44 (0) 131 650 9172*
*Email: p.how@ed.ac.uk*

Andreas Vieli
Editor, The Cryosphere

July 19, 2017

Dear Prof. Vieli,

Thank you for considering our manuscript 'Rapidly-changing subglacial hydrology pathways at a tidewater glacier revealed through simultaneous observations of water pressure, supraglacial lakes, meltwater plumes and surface velocities'. We are grateful to the reviewers for providing constructive feedback, which has enabled us to improve the manuscript.

We have made significant changes to our manuscript following the comments we received. Throughout the manuscript we have added suitable terminology and better descriptions where needed. Alternative controls on ice velocity (i.e. marine influences and glacier dynamics) have been proposed in the Interpretation and Discussion sections. A scenario has been included to suggest that the borehole water-pressure record may not represent an active channel system. Figure 5 has been edited to include three additional velocity maps to better show the upglacier-propagation of the early-melt season speed-up event. We also address the concerns regarding the inclusion of the entire borehole pressure record and the associated GPS velocities from the borehole site.

Please find below detailed responses to the main points raised by each of the reviewers, along with a version of our manuscript highlighting the changes we have made to answer the reviewer comments. We felt it was unnecessary to outline all the minor comments raised by the reviewers here. These comments and resulting corrections can be found in the public responses to each of the reviewers. We refer to line numbers in our manuscript throughout our responses outlined here. These are the correct line numbers in the manuscript with the changes incorporated. We have endeavoured to carefully consider and address all concerns and return the manuscript in a publication-ready state.

Sincerely,

Penelope How

**Reviewer 1: Shin Sugiyama**

*This study combined several different observations and numerical analysis to reveal the evolution of glacier hydrology over one summer melt season. This kind of integrated observational data set is available at only a few limited glaciers, thus presented data are valuable to improve our understanding of the hydrology of tidewater glaciers. Text is very well written and nicely organized, which clearly explains relatively complex methodology and observational results. I like the way of interpretation, first in chronological order in Section 6 and then discussion on selected important processes in Section 7. Plots and photographs are carefully prepared. Overall, I find the manuscript is already in a good standard and interesting to many of the journal readers.*
*The interpretation and discussion on the glacier hydrology are reasonable, but they are based on superficial observations and not much supported by direct evidences. I agree that they are likely scenarios, but other possibilities should be also mentioned. I listed such comments on the authors interpretations followed by relatively minor comments and suggestions, which can be considered for revision.*

We would like to thank the Shin Sugiyama for his comments and positive response to our manuscript. Sugiyama's enthusiasm and curiosity for the subject is evident in his feedback, which is very refreshing to read. We have edited our manuscript accordingly, including edits to Figure 2, the inclusion of glacier dynamics as an explanation for the cause of the lake drainage at the beginning of the 2014 melt season, and the inclusion of a scenario where the borehole pressure does not represent basal conditions in the region. Details of our response to the reviewer's three key comments are outlined subsequently.

**Major comments**

*1. I understand the borehole pressure was recorded from September 2013. Why not show all the data from the beginning of the observation period? Water pressure over one year period provides insights into basal conditions as well as the connectivity of the borehole to the subglacial hydrological system. At least, overview of the pressure record over the entire period should be described in the text.*

The borehole pressure record covers a 14-month period from September 2013 to December 2014. We understand that this is a very valuable dataset that should be shared with the scientific community as soon as possible. However, it was decided to only focus on the 2014 melt season because of two main reasons:

- We believe that the inclusion of the whole record is beyond the scope of the paper. The inclusion of the whole record may detract from the key aim in this paper, which is to build a detailed theoretical model of the hydrology at the glacier terminus of a tidewater glacier during a single melt season. We believe that the entire dataset is not needed to fulfil this aim.

- The beginning of the record (September 2013–March 2014) is strikingly different from the rest of the record. For instance, basal water-pressure appears to exhibit strong, consistent diurnal variability (roughly between 10–50 kPa) from September 2013–March 2014, whilst the rest of the record does not indicate any diurnal variability. This may be because the sensor took a while to settle and give consistent readings, or basal pressure drastically changed over the monitoring period, or the sensor may have been located on a different part of the bed and was subject to a different pressure/hydrological environment. This in itself is an interesting observation and we are still attempting to understand this. Once we have gained a better understanding (and potentially integrated it with subglacial hydrology modelling), it is intended to publish the borehole dataset in its entirety at a later date in a CRIOS project publication.

For these reasons, the entire borehole record will not be included here. Also, an overview will not be included in the text because we believe that the significant difference in the record from September 2013 to March 2014 does not reflect the subglacial conditions in the 2014 melt season.

*2. I wonder if glacier dynamics can be the cause of the lake drainage. When the glacier accelerates near the front, a longitudinally stretching flow regime is enhanced. This causes crevasse opening and increases chance of lake drainage. Assuming that such acceleration initiates near the glacier front and propagates upglacier, the observed lake drainage can be explained by this process.*

Section 7.2 (Upward-propagating supraglacial lake drainage) outlines the dynamics of the three lake clusters monitored in this study and compares their dynamics to other observations from the literature. The lakes in Cluster 1 are focused on in particular because of the coincident timing of their drainage in relation to changes in velocity, runoff and plume activity. The nature of their drainage is discussed in relation to hydrology and it is hypothesised that their drainage is related to their connectivity to efficient drainage in the subglacial environment. Glacier dynamics were not discussed here to avoid repetition with Section 7.5 (Implications for subglacial dynamics).

However, the reviewer rightfully points out that glacier dynamics may be the cause of the lake drainage and the reader may gain the impression that the drainage of the lakes in Cluster 1 is exclusively linked to hydraulic connectivity from the original manuscript. Glacier dynamics may also play a key role in their drainage. Longitudinal stretching is likely to be enhanced at the beginning of the season when the glacier begins to accelerate and this could, in turn, promote the likelihood of lake drainage. As suggested by the reviewer, this hypothesis has now been included in section 7.2 to provide a more detailed explanation for the drainage of these lakes. It is suggested that their drainage may be related to glacier dynamics as well as glacier hydrology:

'The lakes in Cluster 1 are of particular interest because of the coincident timing of their drainage in relation to changes in surface velocities, runoff, and activation of the plume at the beginning of the melt season. This suggests that these lakes are linked to a common channelised system when they drain. The upward-propagating nature of their drainage indicates that channels develop in an upglacier progression as reflected in the timing of their connection to thr subglacial environment. The hydraulic potential modelling supports this as it indicates that Cluster 1 may be situated close to a large channel/flow accumulation pathway. Glacier dynamics may also play a key role in the cause of this lake drainage. Longitudinal stretching occurs as the glacier accelerates at the beginning of the season, which facilitates the opening of crevasses and increases the chance of lake drainage. The upward-propagating nature of the drainage may be a result of this early-season acceleration, assuming that it initiates at the glacier front and propgates upglacier.' (Section 7.2, second paragraph)

*3. Throughout the paper, the authors assume the borehole pressure represents the subglacial water pressure over the region. Nevertheless, the lack of short-term pressure variations gives me an impression that the borehole is not well connected to active subglacial drainage system. The pressure drops in September, but it is only 15 m out of 280 water depth. I agree that the authors interpretation is one of likely scenarios, but it is worth mentioning that there is a possibility that the borehole pressure does not represent basal conditions in the region.*

Hydraulic potential modelling suggests that the borehole is located close to/within the catchment of an efficient channel system, and thus the record reflects basal water-pressure in a well connected region of the glacier bed. However, the borehole record shows few short-term variations over the entire study period that this manuscript covers (May–September 2014), which suggests that the borehole is isolated from the active subglacial drainage system.

The reviewer is right to point out that there is a possibility that the borehole may not be located in an efficient drainage catchment based on the lack of short-term pressure variations. A paragraph has been added to Section 7.4 (Subglacial drainage of Kronebreen) to address this point:

'Few short-term pressure variations are observed in the water-pressure record from May–September 2014, apart from the significant drop in pressure at the end of the melt season. It is possible that the borehole is located on an area of the bed that is not well connected to an active, efficient drainage system. However, changes in water-pressure have been observed to coincide with other features in the hydrological system (i.e. plume activity and supraglacial lake drainage), which suggests that the borehole is hydraulically connected to some degree. This is also supported by the modelled hydraulic potential, which indicates that the borehole is located close to, or possibly within, an efficient drainage catchment.' (Section 7.4, second paragraph)

**Reviewer 2**

*The manuscript presents a useful, multi-faceted dataset, but currently the analysis of the data is simplistic and not fully supported by the evidence presented (for example, suggestions that tides influence the timing of plume pulses). As a result the conclusions are rather vague, and less significant than they could be. Also, it seems as though from the outset that hydrology was identified as the principal control on ice dynamics, and other potential factors have been ignored.*

We would like to thank the reviewer for their comments and constructive response to our manuscript. Their attention to detail has been very valuable for addressing the key points outlined. The authors have taken time and care to respond to each of these points.

We have edited our manuscript accordingly, including the proposal of alterntive controls on ice velocity, and revisions to Figure 5 and associated descriptions concerning the observed early-melt season speed-up event. Details of our responses to the major comments are outlined below.

**Major comments**

*1. No attempt is made to investigate alternative controls on ice velocity apart from variations in subglacial hydrology. This is especially pertinent for the early season flushing event which causes the up-glacier drainage of the supraglacial lakes.*

The main focus of this manuscript is to examine subglacial hydrology at a tidewater glacier, and investigate its influence on glacier dynamics, including ice velocity. The authors felt that investigating alternative controls on ice velocity was beyond the scope of the study, and little data was collected to adequately examine other influences (i.e. calving dynamics and oceanic forcing). However, the reviewer rightfully emphasises throughout their comments that the exploration of other influences is important to presenting a rounded paper that is not weighted towards one set of influences. There is a risk that, in the manuscript's current state, the reader could misinterpret subglacial hydrology as the sole control on ice velocity.

For this reason, the manuscript has been extensively altered to better represent alternative controls on subglacial hydrology. A large effort has been made to better outline all alternative influences, especially in the interpretation and discussion sections where we begin to introduce explanations and ideas concerning the changes we see at Kronebreen over the 2014 melt season. We hope that this is reflected in the detailed responses to subsequent comments. To summarise here, the following alternative controls on ice velocity have now been included in the manuscript:

- Changes in calving activity
- Tidal influences
- Changes in fjord conditions (e.g. subsurface temperature)

- Basal frictional melting

- Ice thickness and shallowness

In particular, changes in calving activity have been more thoroughly explored as an explanation for the 'flushing event' that is observed at the beginning of the melt season.

These alternative controls have been outlined in the interpretation section (Section 6), explored further in the discussion section (Section 7) if needed, and acknowledged in the conclusion section (Section 8). Additional datasets (such as tidal data, calving activity, and fjord temperature) have not been included to examine these alternative controls within this study. The authors argue that too much focus on these aspects will detract away from the main focus of the paper which is to investigate subglacial hydrology and its influence on glacier dynamics. The authors wish to retain one of the main message of the study – that, in addition to glacier dynamics and marine influences, subglacial hydrology plays a vital role in ice velocity at tidewater glaciers.

*2. The borehole water pressure gradually decreases while ice velocity is increasing, which does not tie in with your explanations of ice motion being controlled by the location of efficient/inefficient drainage and the position of regions where water is stored and evacuated from (pg. 1).*

Reviewer 1 previously highlighted that the borehole water-pressure record may not have strong connectivity to the active drainage catchment of the glacier, based on similar observations to those made here. A gradual decrease in water-pressure at the borehole whilst ice velocity increases suggests that the borehole is effectively isolated from the main drainage system.

However, small, coinciding changes have been observed between water-pressure and other observed signals for subglacial hydrology – for example, the observed pressure drop in the early-season 'flushing event' which coincides with the activation of the main plume and the drainage of the supraglacial lakes. Such changes indicate that the borehole is influenced by changes in pressure within the active drainage catchment. This is supported by the hydraulic potential modelling, which shows that the borehole is likely to be located within, or at least near to, an active channel network.

The manuscript has been changed to better convey these possible scenarios. Also, modifications have been made in Section 6, 7 and 8 (the Interpretation, Discussion and Conclusion sections) where arguments have been supported with evidence from the borehole water-pressure record. These have been made in an attempt to clarify that the borehole record may not be connected to the active drainage system.

*3. The description of seasonal variations in ice flow (i.e. that the speedup is constrained to the southerly part of the near-terminus region) does not seem to be supported by the example velocity images shown. It would be useful to produce some plots showing relative changes in ice velocity, so that the reader can see the justification for the discussion.*

During the 2014 melt season, sequential velocity maps from TerraSAR-X image pairs show an early-melt season speed-up which initiates at the terminus and propagates upglacier. The fastest velocities are seen in the southern/central region of the glacier tongue ($>2.4$ md$^{-1}$), whilst velocities in the north region generally are less ($<2.4$ md$^{-1}$). The high velocities subside by August. A second speed-up is observed in September, which propagates upglacier in a similar manner. The authors realise that the wording used to describe these events was misleading. High velocities are constrained to the southern/central region of the glacier tongue, not the speed-up itself. This has now been changed throughout the manuscript to better convey this, and hopefully this will make more sense in relation to the velocity maps presented in Figure 5.

It was challenging to convey all the information regarding the velocities in one figure, so three sequential velocity maps were chosen for Figure 5 that best represented the speed-up in the early-melt season. The reviewer's comment highlights that the images used in Figure 5, along with descriptions of

the velocity event, did not effectively convey this. To rectify this, Figure 5 has been amended to include six sequential velocity maps to better show the nature of the early-melt season event. It is understood that plots showing relative changes in velocity would also convey this, however the authors feel that this is good opportunity to showcase more of the velocity maps produced from the CRIOS project.

*4. Assertions made in the discussion should be backed up with data and results. For example the suggestion that tides influence the timing of plume pulses (but there are many other similar examples as detailed in the specific comments below).*

Within the discussion section, events observed over the 2014 melt season are summarised and potential processes driving these events are proposed. Hydrological processes are largely discussed because subglacial hydrology is the main focus of the paper, and there is sufficient evidence from the data to suggest that these events are hydrologically driven to some extent. Other processes are also outlined, as recommended by the reviewer, such as oceanic influences and glacier dynamics (see the first comment in this response for more details).

The reviewer suggests that data (calving rate, tidal level, fjord temperature etc.) should be used to support assertions and explore these alternative processes in this study. The authors believe that this is beyond the scope of the study. The inclusion of other datasets would detract from the manuscript's primary focus on examining subglacial hydrology at a tidewater glacier. We intend to write a second paper at a later date which looks more closely at the dynamics of Kronebreen, specifically exploring calving dynamics in relation to oceanic forcing and glacial influences. We hope that the ideas presented here are further explored in this future work.

We recognise that it is valuable to outline these additional influences though, as discussed in the first comment of this response. To address this, alterations have been made to the manuscript to emphasise where assertions have been made and where additional datasets are needed. In addition, each of these instances are stated as good ideas for potential work in the future. Specific changes are detailed in the minor comments, which are included in the public response to Reviewer 2.

**Reviewer 3**

*The study is well written, well organized, and thorough in its analysis. The complementary observations are used to illustrate the dynamics and evolution of the hydrologic system of Kronebreen, which evolves in a number of interesting ways, both spatially and temporally. The evidence is strong for the conclusions that the southern portion of the glacier tongue is more dominated by a distributed, pressurized hydrologic system, whereas the central northern tongue receives more meltwater and has a more stable efficient drainage system. Both the delivery of meltwater to the base of the glacier and the organization of the subglacial drainage system need to be taken into account to explain the observed evolution of plumes and glacier speeds. This study will be a welcome contribution to the growing field of coupled glacier dynamics and hydrology. I have only minor recommendations, questions, and comments on the manuscript.*

We would like to thank the reviewer for their comments and positive response to our manuscript. We have edited our manuscript accordingly, including better definition of frequently-used terms and more detailed descriptions in places. We have also addressed the comment on the inclusion of the GPS velocity data. We hope that the reviewer (and the editor) finds that all their comments have been carefully considered and thoroughly addressed. As the reviewer felt it unnecessary to include major corrections, we have included our response to a selection of the reviewer's key comments. These are outlined below.

**Major comments**

*Figure 1: it seems that you could change the aspect ratio of the figure to zoom in more on the study area. Areas to the north and south of Kronebreen arent really necessary to include, other than to make room for your inset panels.*

The authors attempted to: 1) change the aspect ratio; 2) move the inset panels; and 3) zoom into Kronebreen. We found that numerous problems occurred when trying to accomplish this. Primarily, the Landsat image becomes pixelated and coarse when we tried to zoom into our field site, which isn't as visually pleasing. It was also difficult to move the inset panels without covering the plume extents or camera positions, even when we had changed the aspect ratio. Equally we felt that it was valuable to include some of the fjord and neighbouring glaciers (Kongsvegen and Kongsbreen) as context for the reader.

For these reasons, the authors have decided not to change Figure 1.

*Borehole GPS: it seems surprising (even if true) that the borehole GPS didnt add anything insightful. Why not show this and demonstrate that this is the case? It seems that you could overlay the GPS-derived velocity on Figure 2d to show this.*

This point was also noted by Reviewer 1. As stated in their response, the GPS data was not included in this study for three main reasons:

- The GPS velocity record is incomplete. The GPS was offline at the beginning of September 2014, whilst the rest of the dataset record carries on till the end of September 2014. The record duration is therefore mismatched.

- The higher temporal resolution of the GPS velocities does not appear to add anything new to the study. There were difficulties in processing the GPS data and short-term variations cannot be distinguished from the daily positions that we extracted. The dataset generally appears noisy. To resolve this and provide an alternative, velocities were derived from the TerraSAR-X imagery and then a spot velocity was extracted from the borehole site. These appear much less noisy and fit well with the rest of the 2014 record.

- The key findings from the velocity data focus on the spatial variability in velocity over the glacier tongue, rather than changes in velocity over time. These are better addressed with the TerraSAR-X velocities rather than the GPS velocities. The inclusion of the TerraSAR-X velocities from the borehole site are also consistent with the velocities derived from the other ROI's (i.e. from the centreline and the supraglacial lakes).

To better show the noise in the dataset, we have included Figure 2 as supplementary material in the public response with the GPS velocities included in panel F. It is clear from this that the GPS velocities do not add any additional information to the study, and any changes seen in velocities are difficult to associate with the other datasets.

For these reasons, the GPS data will not be included in this paper. The difficulties with integrating the GPS velocities has been clarified in the methods section of the manuscript (Section 4.4, second paragraph):

'A Topcon Net-G3A GPS unit was installed at the position of the transmitter to track the approximate movement of the sensors. It was decided to use the surface velocities derived from TerraSAR-X images rather than the GPS because the GPS velocity record was incomplete and the higher temporal resolution of the GPS data did not add any further insights to this study. The GPS data appeared noisy due to difficulties in processing the positions.'

*Section 5.6: its not clear how you arrived at a value of k=0.6 as a sort of threshold for routing of meltwater between the northern and southern sections of the glacier. You state that results suggest this, but dont specifically describe why. Several scenario were considered (line 15), but what do you mean by this? How do you arrive at the conclusion that flow routing changes between a value of k=0.5 and 0.6 (line 19)? This seems different than what you describe in line 18 about threshold routing above and below a value of k=0.6? I guess Im just a bit confused about this section, perhaps its just a matter of describing more specifically what youve done here.*

Subglacial hydraulic potential was calculated primarily based on ice thickness and bed elevation. The crostatic pressure factor ($k$) is the ratio of water-pressure to ice overburden pressure. Variations in the value of $k$ reflect the degree to which subglacial drainage is pressurised with $k=0$ reflecting open channel flow at atmospheric pressure, and $k=1$ reflecting pressurised flow.

We calculated subglacial hydraulic potential over several iterations, changing the value of $k$ each time. In total, we ran 11 simulations with the value of $k$ between 0.0–1.0 (i.e. hydraulic potential was calculated each time with a $k$ value of 0.0, 0.1, 0.2, 0.3, 0.4, 0.5, 0.6, 0.7, 0.8, 0.9, and 1.0). These are what are referred to in Section 5.6 as the 'several scenarios' that we considered.

We found that there is little change in the configuration of the channel network when $k$ is 0.0–0.5. In all these instances, the calculations suggest that a major channel connects Holtedahlfonna to the south region of the glacier tongue. There are significant changes in the channel configuration when $k$ is 0.6 and above (i.e. $k=0.6$–1.0). The major channel diverts to the north region of the glacier tongue in these scenarios. Therefore there is a significant difference when we consider hydraulic potential with a $k$ value of 0.5 and below, and 0.6 and above. This is what we refer to in the manuscript as the 'threshold' as it is apparent that this difference occurs between a $k$ value of 0.5 and 0.6.

The authors appreciate that some of the terms used in this section are not accurate and more appropriate, detailed wording could be used instead. We have changed the section accordingly to make this clearer:

'Several scenarios were considered in calculating the hydraulic potential at the bed of Kronebreen based on the $k$ value, which represents cryostatic pressure ratio (i.e. the extent to which meltwater routing is dictated by ice-pressure gradients). Subglacial hydraulic potential was calculated over several iterations, changing the value of $k$ each time. In total, we ran 11 simulations with the value of $k$ between 0.0–1.0 (i.e. hydraulic potential was calculated each time with a $k$ value of 0.0, 0.1, 0.2, 0.3, 0.4, 0.5, 0.6, 0.7, 0.8, 0.9, and 1.0).' (Section 5.6, paragraph 1)

*Borehole pressure: the pressure variations you record indeed seem to suggest that you are not actually located to a connecting channel. I would expect more pressure variations if you were. You seem to suggest that you might not be located at a channel, but argue that you are 'near' a connecting channel if not connected to a channel that is consistently full of meltwater (in which case I would still expect to see more pressure variations).*

The reviewer is correct in stating that we would expect to see more changes in the water-pressure record if the borehole sensor was located in a connecting channel. This is now clearly stated in Section 7.4 (Subglacial drainage of Kronebreen) following similar comments from Reviewer 1.

We suggest that the borehole is possibly located near to an active drainage system based on instances where changes in pressure have coincided with other changes related to subglacial hydrology (e.g. the early-melt season 'flushing' event, and the significant pressure drop in September). This is also supported by the hydraulic potential modelling which indicates that the location of the borehole intersects with one of the main channels in the catchment. We propose that the borehole is located within the catchment of an active drainage system based on these arguments. Absolute changes in the water-pressure record suggest differently as noted by the reviewer.

Therefore we have two lines of evidence, with one suggesting that the water-pressure is indicative of an active drainage catchment, and the other suggesting that the record reflects an isolated, consistently

pressurised region of the bed.

A paragraph has been added to better outline these ideas in Section 7.4:

[revised manuscript text omitted]

---

## Author Response (AR2)

**Authors' response to re-review for 'Rapidly-changing subglacial hydrological pathways at a tidewater glacier revealed through simultaneous observations of water pressure, supraglacial lakes, meltwater plumes and surface velocities'**
* * *
**Editor comments (Andreas Vieli)**

*As the editor of this manuscript, I had a detailed look at the 3 reviews, your response and the corresponding revisions of the revised version of the manuscript. In general the reviews were rather positive and highlighted the value and novelty of the presented datasets and related interpretation with regard to hydrology of tidewater glaciers. However, most reviewers also thought that a wider discussion of alternative controls on terminus flow/dynamics (frontal/geometry changes, calving, tides,) should be included, and was somewhat lacking in the first version.*
*The authors undertook substantial revisions at the manuscript and the revised manuscript has been sent to re-review to one of the referees of the first round (Ref2) with the detailed review listed below.*
*According to the re-review and my own detailed analysis of the revised manuscript and author response, the authors addressed most of the major and most of the minor issues raised by the reviewers well.*
*However, the few more substantial concerns of reviewer 2 and reviewer 1 that refer to the point of 'alternative controls on ice velocity (calving/front dynamics, ocean forcing, tides)' were in the view of the referee and myself rather superficially or evasively addressed and moreover quite a number of additional minor editing or formulation issues were introduced with the revisions (see minor comments of re-review and editor below).*
*In the revised manuscript the effect of 'marine influences, glacier dynamics, calving dynamics' are now mentioned in the interpretation as other potential factors explaining for example the observed early season acceleration and potentially related processes but in general these statements remain extremely vague and lacks reference to existing literature/understanding, the potential mechanisms are often rather difficult to follow and the reader is repeatedly left with the sentence along the line of: 'it cannot be further explored here, but could be an interesting focus in future work'.*
*I understand that that given the data available perhaps no detailed analysis of these additional controls can be expected or may be (as stated by authors) beyond the scope of this study.*
*However, leaving these alternative controls that vague weakens the interpretation/conclusion of this study and perhaps more importantly at least some of the existing understanding and literature on the general influence of front retreat/calving or ocean on flow speed etc should be included, in particular as there is even some related literature on this on Kronebreen itself (Luckmann et al 2015, Schellenberger et al 2015 TC). Similar the analysis of the effect of tides on periodic drainage remains vague and could perhaps be improved or at least checked by considering relative frequencies of plume visibility and tides (see re-review below).*

The authors would like to thank Andreas Vieli for his thorough response and the opportunity to amend the manuscript in light of the issues raised. We have endeavoured to address all the points that have been highlighted, including:

1. An amended discussion of subglacial dynamics at Kronebreen, for which changes focus on:

    (a) Exploring alternative controls on the early-season speed-up using related literature to strengthen arguments and steer away from vague statements.

    (b) A conscious effort at distinguishing the discussion concerning the early-season speed-up from the discussion of the spatial patterns in surface velocities over the entire melt season. This

was primarily tackled by including velocity maps showing absolute differences in ice velocity over time to better show the nature of the speed-up event.

2. The inclusion of tidal data to better explore marine influences on episodic plume outflow, and strengthing arguments more effectively with support from the surrounding literature.

All changes are outlined in the subsequent sections, first looking at the editor's major comments and then the minor comments. The authors have taken time and care to ensure that each point has been thoroughly looked at.

**Major comments**

*1) Try to make the discussions of alternative controls on flow (front retreat/calving/ocean) a bit more specific, explain a bit better how these alternative controls may function and importantly include where possible existing understanding/knowledge and refer to the general (from other tidewater glaciers) and specific Kronebreen literature (Luckmann et al 2015, Schellenberger et al 2015 TC) there. It seems to me that we know quite a bit about these controls on flow. I do not think it requires n extensive discussion here but the most relevant mechanisms and literature should be mentioned and clarified. The above refers roughly to p. 16 lines 18–22, p. 18 lines 3–5, p. 20 lines 12–14, p. 21 lines 18–22)*
Our intention with the Interpretation section of the manuscript is to bring together coinciding observations from the datasets to begin to build a picture for what is going on at Kronebreen for the 2014 melt season. The Discussion section is really where we intended to tackle why we see these changes and potential controls. Therefore the Discussion section is where alternative controls are explored using exisiting knowledge from the surrounding literature. However, the authors realise that introducing these ideas in the Interpretation section with little support from the surrounding literature may appear speculative. Relevant studies are needed to outline potential, viable controls, which build a strong foundation for further exploration in the Discussion section. In each of the cases outlined by the editor, we have made an effort to include relevant studies to better outline the mechanisms behind our observations:

1. Page 16, lines 18–22: We have included comparison to observations (Howat et al., 2005) and modelling work (Nick et al., 2009) on Helheim glacier that also see ice acceleration confined to the near-terminus area. We also outline the idea that this is caused by a change in boundary conditions which reduces resistive stresses, and support this with observations from Luckman et al. (2015) which showed that a marked increase in calving retreat preceded the early-season acceleration.

2. Page 18, lines 3–5: We have included comparison to lake drainage observations from Stevens et al. (2013) and Everett et al. (2016) to illustrate how lake drainage can be caused by changes in stresses across the glacier and/or changes in meltwater presence at the bed.

3. Page 20, lines 12–14: Similar to the first point, we have included Howat et al. (2005) and Nick et al. (2009) to illustrate that the speed-up at Kronebreen could be related to changes at the terminus, such as a change in calving activity and/or marine conditions.

4. Page 21, lines 18–22: Similar to the second point, we have included Stevens et al. (2013) and Everett et al. (2016) to better explain controls on the early-season 'flushing event' at Kronebreen.

In addition, we have also included references to other studies which observe subglacial hydraulic pulsing that is disassociated from runoff. Kavanaugh and Clarke (2001) relate this to episodic ice

motion which is associated with the gradual failure of a 'sticky spot' following hydraulic connection. Our TerraSAR-X velocity record is not at a high enough resolution to look into this (and our GPS velocity record is too noisy to distinguish links), but it is an interesting study that provides a viable explanation for internally-driven hydraulic pulsing.

*2) Better consider the the point on tidal forcing raised by the re-review (e.g. by considering relative frequencies of plume visibility and tides, see p. 22 lines 8-16).*

Tidal level data has now been incorporated from a tidal gauge located in Ny Ålesund, which is only 12 km away from the front of Kronebreen. This shows that there are no apparent links between tidal cycles and hydraulic pulsing from the south side of the glacier terminus (i.e. Plume S1). This, in turn, strengthens the argument that this water outflow is controlled by internally-driven hydrualic processes. This has now been added to the Interpretation and Discussion sections.

*3) Fig. 5: I agree with re-review that one can well see the acceleration (everywhere/almost uniformly) over the summer, but not really well see the 'upstream propagation' (meaning wave of acceleration that propagates upstream). If this 'upstream propagation' is really crucial for the argument/interpretation maybe better way of illustrating the spatial flow evolution should be done (perhaps as re-review suggests relative change to first flow field and move current fig. 5 into appendix or so).*

As requested, we have amended Figure 5 with the inclusion of velocity maps that show absolute differences in velocity which are relative to a TerraSAR-X image from the beginning of the melt season (04 June 2014). Reviewer 2 suggested including maps of percentage difference but this highlighted uncertainty in low-speed areas, so maps of absolute difference were deemed a better solution. These maps show that ice acceleration is fastest at the terminus, with lower acceleration experienced upglacier and in the northern region of the terminus during the early-season speed-up event.

It was also suggested to possibly move the original Figure 5 to an appendix and only present the maps showing relative change. The authors believe that the original velocity maps are a key part of the study though, as one of the main messages is that the south region of the glacier tongue consistently flows faster than the north region. We have chosen three velocity maps and three absolute change maps that sufficiently show this difference in velocities and also shows the early-season speed-up event. We believe that this is sufficient for presenting the findings from the study. However, we are open to including other velocity maps as supplementary material if the editor and/or reviewer think this is appropriate.

To summarise, Figure 5 now shows velocity maps (derived between image pairs) and absolute velocity differences (since 04 June 2014). These cover the period between 15 June and 18 July 2014 to effectively convey the spatial differences in velocity and the early-season ice acceleration. We have also re-written the section on subglacial dynamics in order to better discuss the spatial patterns in velocity and the early-season speed-up event separately. We have made a conscious effort to effectively distinguish these aspects, especially when talking about the velocity maps in Figure 5.

*4) Try to avoid sentences that divert unexplored processes/effects to future studies (see also re-review below), as they do not add anything and give the impression that it is completely unknown (which is not always the case).*

The majority of instances where we commented on unexplored processes for future studies have been removed. To compensate, we have attempted to strengthen arguments and, in cases where it is difficult to further examine, removed speculation. Examples of this include:

1. Tidal data was added to strengthen the argument that hydraulic pulsing was not related to tidal cycles.

2. Observations from the time-lapse imagery were also used as supporting evidence. Specifically

snow cover was observed to be absence early in the melt season, which strengthens the argument that meltwater was bypassing storage at the glacier surface. Equally plume observations were used to show that the plume surfaced from single source at the beginning of the plume activity, suggesting that it is channel-fed (see comments from Reviewer 2 for more details).

3. Observations from Luckman et al. (2015) were used to further explore the cause of the 2014 early-season speed-up – links between the 2014 early-season speed-up with an increase in calving retreat that precedes the speed-up, suggesting that the speed-up may have been caused by a change in boundary conditions at the glacier terminus. Similar speed-up events have been observed by Howat et al. (2005) and modelled by Nick et al (2009).

*5) Address carefully all the minor rather technical issues/comments by the re-review and editor listed below.*

    All minor suggestions have been considered and action has been taken to amend these. Full details of how these have been addressed are outlined below.

**Minor comments**

*Page 2, line 28: This sentence refers to alpine glaciers but the Kamb study is on tidewater glaciers (Columbia). Kamb 1994 should certainly be cited but at the relevant place where subglacial hydrology of tidewater glaciers are discussed (e.g. end of same paragraph or on page 3).*

    The reference has now been moved to the end of the paragraph as recommended by the editor.

*Page 5, Fig 1 caption: Rather say: 'the star marks the location of '*

    The caption has been changed as recommended by the editor and now reads '...the star marks the location of...'.

*Page 10, Fig. 2F: I would add symbols (dots or crosses) at each datapoint as currently one could get the impression that the data is continuous (in particular as the uncertainty band is also shown as continuous).*

    It is agreed that the inclusion of symbols at each datapoint would better illustrate that the velocities presented are not a continuous dataset. This has now been changed in Figure 2. The legend for plot 2F was initially obscured by this change. Therefore the position of the legend was also changed to ensure that it was still clear for the reader.

*Page 14, line 10: This 6m increase refers probably to 09 JULY and not 09 September!*

    Yes, the editor is right in pointing out that this refers to 09 July rather than September. Change made.

*Page 18, line 17: A bit odd to refer to this period of 31 may – 16 august when time periods of the imagery in Fig. 5 are different. Maybe replace it with between mid May and end of August.*

    Agreed. Sentence changed to '...between mid-May and the end of August.'

*Page 21, line 18: I would delete the key as this is speculation anyway.*

    Agreed. This sentence has been further edited based on comments for reviewer 2. The sentence now just reads:

'The drainage of the lakes at Kronebreen are likely to be linked to both a change in glacier dynamics and a change in conditions at the bed, namely due to an increase in meltwater at the bed.'

*Page 25, line 11: Evidence suggests what evidence, I guess data/observations from this study is meant here so say something along the line of : Our observations/analysis suggest*

Agreed. The original wording is a bit confusing as an opening to a paragraph in the conclusions. It has been changed, as suggested by the editor, to 'Our observations suggest that...'

*Page 25, line 19: high rate of deformation this implies that basal motion is due to sediment deformation which you do not really know, I would rather say high basal motion.*

Agreed. Wording changed to 'high basal motion'.
* * *
**Re-review by Reviewer 2**

*This is my second review of this manuscript, and although it has certainly improved since I first reviewed it, some of the changes and additions are superficial and do not deal sufficiently thoroughly with my initial suggestions (and those of the other reviewers). Also, there remains a tendency for unjustified speculation. On several occasions (e.g. regarding the potential effect of tides on periodic drainage of subglacially stored water) the speculation from the first submission remains, but an extra sentence along the lines of Although we cant tell with our data, it would be an interesting topic for future work has been added. I dont think this is good enough. Surely for the specific case of tidal influence, a quick check of the relative frequencies of plume visibility and tides would be informative and is fairly simple to achieve?*

We would like to thank the reviewer for their second round of comments and recommendations. Their clear summary and detailed minor comments have been really helpful in improving the manuscript. We hope that the time and care taken to address the reviewer's feedback is reflected in our response.

The main aim of this paper is to examine glacier hydrology at a tidewater glacier in Svalbard, and look at how hydrology influences glacier dynamics. The authors understand that some of the ideas presented in this manuscript were not thoroughly examined, and this is because we wish to maintain this focus on hydrology. We also think that these would be interesting ideas to explore in future work. However, we also appreciate that, in refining the focus of the manuscript, exploring alternative influences without data and thorough investigation may appear speculative. Therefore we have attempted to address this in the re-revised version of the manuscript with the inclusion of tidal data and better examination of alternative controls on glacier dynamics using related work in the surrounding literature. In addition, we have included velocity maps showing absolute velocity changes from the beginning of the melt season. Further details about these changes, along with other minor changes, are outlined below.

**Minor comments**

*Title: 'hydrology pathways' should be 'hydrological pathways'*

Agreed. Change made.

*Page 1, line 3–4: 'water pressure' should be 'borehole water pressure'*
    Agreed. Change made.

*Page 2, line 3: 'provides' should be 'provide' (data are plural)*
    Agreed. Change made.

*Page 2, line 5: 'across' should be 'beneath'*
    Agreed. Change made.

*Page 2, line 11: 'marine' plume seems an odd term to use here as these could be caused by wind-driven upwelling for example. How about 'glacial' or 'meltwater' plume instead?*
    Agreed. 'Marine plume' is not typically used in other studies and has been therefore changed to 'meltwater plume' which is a more commonly used term.

*Page 2, line 22: Is it worth including some references for the observation of high pressures for most of the melt season?*
    There are plenty of studies out there which observe high basal pressures through a melt season. We have now included two of the key, classic studies at the end of this sentence – Meier and Post (1987) and Jansson (1995):
    'Consistently high basal water-pressures have also been observed over long periods of the melt season (e.g. Meier and Post, 1987; Jansson, 1995).'
    In addition, we have included a reference in the preceding sentence to a study which presented hydraulic pulsing in a borehole record – Kavanaugh and Clarke (2001).

*Page 2, line 24: 'and' should be 'where'*
    Agreed. Change made.

*Page 2, line 26–27: This has also been observed at land-terminating margins of the Greenland Ice Sheet*
    True. And details and references related to work at Greenland outlets (namely Doyle et al., 2015) have been outlined in the same paragraph. Therefore the line has now been changed to:
    'Changes in basal water-pressures have been linked to enhanced basal sliding and surface velocities at land-terminating Greenland outlets and valley glaciers.'

*Page 2, line 27: 'are' should be 'is'*
    Agreed. Change made.

*Page 3, line 4–5: Im not quite sure what is meant here by the 'two component structure'. I think this requires clarification.*
    The term 'two-component structure' refers to the two parts of the model for simulating bed dynamics – the first part being the model to calculate ice velocity, and the second being to calculate the basal water-pressure. This has now been better clarified in the sentence and the term 'two-component structure' has been removed to avoid misinterpretation:
    'Ice velocity and basal water-pressure are typically calculated separately before linking them together to create a unifying model.'

*Page 3, line 6–7: The wording here is a little confusing. What if the implementation of the approach were perfect but the model didn't include key features of the real system? If model outputs do not match real-world velocities, is the representation of the subglacial environment really 'adequate'?*

It is agreed that the use of the word 'adequate' in describing the model outputs is contradictory to the second part of the sentence. A better way to describe this is that the models show promise in delivering adequate representations of the subglacial hydro-dynamic environment i.e. future development is needed, but they show great potential. The sentence has been altered to reflect this message:

'This work shows promise in represent the evolution of the subglacial hydro-dynamic environment. However, implementations of this approach are still imperfect as outputs do not always match real-world ice velocities...'

*Page 3, line 10–11: But lakes are also formed from surface melt, so this is not really 'additional'. This needs to be clarified.*

The reviewer is right to point out that the drainage of supraglacial lakes is not an 'additional' input. Instead, supraglacial lakes are the accumulation of meltwater at the glacier surface in topographically low areas with little/no drainage. The beginning of the paragraph has now been changed to better describe this, and the full altered paragraph is included in the subsequent, related comment.

*Page 3, line 11: Once drained (and therefore once a surface-bed connection has been made), the perched lakes become subglacially connected? I think this characterization is a bit confusing. How about surface melt-fed vs. subglacially-fed or something similar?*

The terms 'perched' and 'subglacially-connected' are convoluted as the reviewer rightfully illustrates. Equally, the terms 'surface melt-fed' and 'subglacially-fed' may also be convoluted terms as a 'subglacially-fed' lake can also have meltwater inputs from the surface. Therefore characterizing supraglacial lakes might be unwise in describing different sources of meltwater and changes in connectivity with the bed. It has therefore been decided to take out terms which characterize supraglacial lakes and merely distinguish differences in the source of meltwater in a supraglacial lake and differences in their connectivity to the bed. The paragraph has been changed as follows:

'As previously outlined, meltwater typically enters the subglacial environment from the glacier surface via surface melt production. Melt can collect on the glacier surface in topographically low areas where there are few or no drainage pathways. This creates supraglacial lakes, which are effectively isolated from the influence of subglacial hydrology. These lakes drain when they become connected to the bed by mechanical processes such as hydrofracturing (Van Der Veen, 2007). This can provide an abrupt injection of meltwater into the subglacial environment. Water in supraglacial lakes can also be sourced from the subglacial zone when water-pressure at the bed exceeds ice overburden, effectively squeezing subglacial water up to the glacier surface. This water often contains entrained subglacial sediment, giving the lake a sediment-laden appearance. The water level in supraglacial lakes that are connected to the bed can be used as a measure of basal water-pressure (Danielson and Sharp, 2013).'

*Page 3, line 14: Be clear that you mean supraglacial lakes here.*

An effort has been made to better clarify that supraglacial lakes are being discussed throughout the paragraph. See the amended paragraph in the previous comment.

*Page 3, line 15: 'contains entrained' instead of 'entrains'*

Agreed. Change made. See amended paragraph in comment about page 3, line 11 (two comments

back).

*Page 3, line 22: 'Detailed' is vague. Do you mean higher temporal and/or spatial resolution?*

Here, a higher temporal resolution would be desirable in order to better pinpoint the timing of supraglacial drainage. This is illustrated in the previous sentence, stating that many observations of lake drainage events are based on temporally intermittent records (e.g. low repeat-pass satellite imagery). This has now been amended, with better links to the previous sentence:

'Improved observations (i.e. at a higher temporal resolution) of supraglacial lake drainage events are needed..'

*Page 3, line 25: 'meltwater in the subglacial zone' could be replaced with subglacial meltwater*

Agreed. Change made.

*Page 3, line 26: The sentence beginning 'This has' does not add anything useful  what other settings are there anyway?*

Agreed. The sentence has been omitted and instead, observations have been divided into those from near-terminus settings, and those from the interior of an ice sheet. The sentence following on from the one in line 26 has been changed to:

'In near-terminus settings, a rapid input of meltwater has been observed to cause...'

*Page 3, line 27–28: 'to make a channelized system become more efficient' could be 'to increase the efficiency of a channelized system'*

Agreed. Change made so that the sentence now reads as follows:

'...has also been observed to increase the efficiency of a channelized system by enlarging channels to accommodate the extra meltwater...'

*Page 3, line 29: 'significant periods of time (i.e. decadal)' should be 'decadal timescales'*

Agreed. Change made.

*Page 3, line 33: How do these long residence times relate to surface uplift which typically recedes within 48 hrs? Is there any evidence (apart from modelling) which supports this idea of long residence times of subglacial water 'blisters'? Also, the excessive use of parentheses to characterise timescales is unnecessary (e.g. see my suggested simplification above).*

The juxtaposition of the ideas presented in this passage appear to be contradictory. In the study by Stevens et al. (2013), observed ice uplift occurs over 48 hours. The ideas presented by Dow et al. (2015) suggest that modelled subglacial water 'blisters' could have much longer residency times. Therefore there is a mismatch in timing. As far as the authors are aware, there are no direct observations which confirm that subglacial water blisters are present beneath an ice sheet. Observations of ice uplift are more abundant. Therefore the passage has been altered to better convey that water is likely to be stored at the bed for longer periods of time under the interior of an ice sheet. This storage has been linked to localised uplift that has been observed to last for 48 hours:

'Below thick ice in the interior of an ice sheet, channels cannot grow as rapidly or sensitively to point inputs, and water evacuation is less efficient. Although it is challenging to directly observe, studies have suggested that water is stored at the bed for longer periods of time in these settings, causing localised areas of ice to uplift from the bed for up to 48 hours (e.g., Stevens et al., 2013).'

*Page 4, line 10: Both satellite and time-lapse imagery are always temporally intermittent (unless you use video!). It is the relative rate of change in plume behaviour vs the temporal frequency of image acquisition which is important.*

Yes, this is an important point that is not adequately illustrated in the manuscript. The sentence concerning temporal intermittency has been deleted and replaced with a sentence that better describes this:

'It is also challenging to acquire a temporal frequency of image acqusition that effectively captures the relative rate of change in plume behaviour.'

*Page 4, line 12–13: I dont think this adds anything worthwhile.*

Agreed. This sentence has now been removed.

*Page 4, line 27: I think the calving rate should be positive without the context of its net impact on glacier mass balance (which I suppose would be negative).*

Agreed. The calving rate hs now been changed to a positive value.

*Page 4, line 29: Consider replacing the second use of retreat with 'receded' to avoid repetition.*

It is agreed the the use of the word 'recede' would help avoid repetition in the passage. The second occurrence of the word 'retreat' has now been changed to 'recede'.

*Figure 1 caption: '8 km LONG tongue' otherwise not obvious which dimension you are referring to.*

Agreed. The caption has now been changed to:
'The glacier consists of an 8 km-long tongue...'

*Page 7, line 2: underwent refreezing should be refroze*

Agreed. Wording changed to 'refroze'.

*Page 7, line 29: Both models should be cited here.*

The distributed energy balance model was developed along the lines presented by Klok and Oerlemans (2002), and the snow model is based on SOMARS which was developed by Greuell and Konzelmann (1994). These references have now been added to the manuscript accordingly:

'A distributed energy balance model (based on Klok and Oerlemans, 2002) coupled with a snow model (SOMARS, developed by Greuell and Konzelmann, 1994) was used to compute melt production and runoff for the 2014 melt season.

*Page 8, line 14: 'up-glacier' rather than 'upper'*

Agreed. Changed to 'upglacier', in-keeping with all other uses of the term upglacier (rather than up-glacier).

*Page 8, line 20: 'is' should be 'are'*

Agreed. Change made.

*Page 8, line 23–24: The fact that the GPS did not provide any extra useful information seems surprising - were the data simply too noisy?*

Inserted below is the amended version of Figure 2, which was included as supplementary material to the response to Reviewer 3 in the first round of reviews (updated to include the requested modifications

in this response). This version of the figure includes the velocity data from the GPS that was situated by the borehole drill site in plot 2F.

The velocities from the GPS follow the general trend of the corresponding velocities from the TerraSAR-X imagery (the red line labelled 'borehole' in plot 2F). There is greater variability in the GPS velocities because of the high temporal resolution (i.e. daily) compared to the TerraSAR-X velocities (i.e. every 11 days). However, it is the opinion of the authors that the GPS offers no further insights to the study. No additional links can be established between the higher resolution dataset and the rest of the data presented in Figure 2 (i.e. lake drainage, plume activity, modelled melt/runoff, basal water-pressure).

This comment has appeared multiple times throughout this review process. We have attempted to better incorporate the GPS velocities by smoothing the data, but this results in the data closely resembling the TerraSAR-X velocities, and thus adds no extra detail to the figure. We also do not wish to circulate a dataset that we are not entirely confident in, and we remain unsure as to whether the small variations in the GPS velocities are real or merely noise.

For these reasons, we have not included the GPS velocity data. This has been adequately explained on page 8, line 23–24 of the manuscript, as requested by the reviewers in the first round of corrections.

[Figure]

Amended figure 2: Composite graph showing hydrological results from Kronebreen, including GPS velocities from the borehole drill site in plot 2F.

*Page 9, line 18: 'on' should be 'in'*

Agreed. Change made.

*Page 11, line 22: How was the runoff 'detected'? Surface lakes become visible or plume first reaches the fjord surface?*

Poor use of language. This statement is comparing the modelled runoff to the modelled melt production. Therefore the runoff is not 'detected', it is predicted based on the model. This has now been made clearer in this sentence:

'Surface melt production begins on the 26 May, approximately one month before the onset of runoff is predicted by the model.'

*Figure 3 caption: You cannot know that the lakes are entirely drained from the camera angle just that you cannot see any water in them.*

The reviewer is right to point out that the record does not signify when the lakes completely drain of water, instead they denote when water is no longer visible from the given camera angle. This has now been better worded in the figure caption:

'...F) Downglacier lakes drain and no remaining water is visible from the given camera angle; G) Upglacier lakes partially drain and some remaining water is visible; H and I) Upglacier lakes continue to drain gradually; J) No remaining water is visible in any of the lakes by this point.'

*Page 14, line 29: 'The relative timing of these components' could be 'The relative timing of variations in these components'*

Agreed. Change made.

*Figure 5: It is difficult to tell the difference between >2.4 m/d and no data. Could you use a more effective colour scheme?*

The velocity maps have been altered in two ways to resolve the difficulty in distinguishing cells with high velocities from cells with no data:

1. The velocity maps have been masked to the area of interest. This is to depict the velocities at Kronebreen more effectively and eliminate the noise in the surrounding scene.

2. Cells with no data are represented with no colour, rather than white. This reduces the similarities between pixel cells with high velocities and no data.

We considered changing the colour scheme, but we believe that the changes detailed above provided a more effective solution to the outlined issue.

*Figure 5 caption: use 'up-glacier' instead of 'upwards'*

Agreed. Change made.

*Page 16, line 20: 'enable ENHANCED glacier flow' (after all, the glacier is already flowing)*

Agreed. This paragraph has now been altered to better explore causes of this enhanced glacier flow using similar studies on Kronebreen and Helheim glacier, so this phrasing is no longer used:

'The surface velocity of the glacier begins to gradually increase from 10 June, based on the velocities from the ROI's – the centreline, the region of the supraglacial lakes, and the borehole site (Fig.

2G. The nature of this speed-up is similar to those observed by Howat et al. (2005) and modelled by Nick et al. (2009) at Helheim glacier, with the acceleration beginning at the terminus and propagating upglacier. They attribute this to changing boundary conditions at the glacier terminus. Luckman et al. (2015) observed a marked increase in calving retreat at the front of Kronebreen at the beginning of the 2014 melt season, which precedes this early-season ice acceleration. It is likely that the observed change in conditions at the glacier terminus are linked to the changes in surface velocity. Specifically, the increase in calving rate could have reduced back-stress further upglacier and enable enhanced glacier flow (Nick et al., 2009).'

*Page 16, line 21: Some missing words? 'but the influence of submarine ice melt on glacier dynamics could be'?*

Agreed. As noted in the previous comment, this paragraph has now been altered to better explore causes of this enhanced glacier flow using similar studies on Kronebreen and Helheim glacier, so this phrasing is no longer used. See previous comment for revised paragraph.

*Page 16, line 23: 'supported because of the coinciding observations' could be 'supported by coincident observations'*

Agreed. This passage has now been altered to incorporate support from the surrounding literature on the dynamics of Kronebreen. Therefore this phrasing is no longer needed. The new passage is:

'Another influence is the presence of meltwater at the bed, which enhances basal lubrication and enables sliding. This has previousl been highlighted as a key process at Kronebreen (Schellenberger et al., 2015 and could also be the case fot the 2014 melt season...'

*Page 16, line 26: Other surface catchments perhaps? But not other catchments as determined by the combined bed and surface topography surely?*

It is unlikely that meltwater would have originated from other adjacent catchments based on evidence from the hydraulic potential work. Water may come from land runoff (e.g. runoff from mountains and high topographic features) or from regions beyond the area of interest (e.g. the upper reaches of Holtedahlfonna). These regions beyond the area of interest are what we are referring to by 'other glacier catchments'. This wording may be slightly confusing. The term 'higher elevations' could encompass topographic highs in the land and higher areas in the glacier catchment. Therefore the sentence has been changed, with 'other glacier catchments' deleted and merely 'higher elevations' used to describe potential alternative sources of meltwater:

'Surface meltwater may have originated from higher elevations, but it is unlikely given that early-season melt production is understood to first originate from the lower elevations of this glacier catchment (Van Pelt and Kohler, 2015).'

*Page 18, line 6: 'this' is a bit ambiguous. How about 'the sequence of events described above'*

Agreed. This passage has also been changed to better support ideas with observations in the surrounding literature:

'Longitudinal stretching may have initiated the activation of the plume and the drainage of the lakes at Kronebreen, and this may be controlled by changes at the terminus of Kronebreen (e.g. an increase in calving activity) and/or the observed early-season speed-up. Hydro-fracturing has also been linked to changes in meltwater presence at the bed, which promote drainage via basal slip (e.g., Stevens et al., 2013). A similar scenario at Kronebreen could be an indication of widespread drainage that occurs in an upglacier-propagating pattern...'

*Page 18, line 11: The hydraulic connection (rather than the distance per se) is key here: is the borehole connected to the channelised system and the adjacent regions which experience temporal variations in water pressure? From the data it looks like it is not (although you say that the water level in the borehole dropped substantially when the bed was reached).*

There is evidence for and against the borehole being connected to a channelised system. The water level in the borehole dropped when it first made contact with the bed, indicating an efficient means of drainage was present. In addition, there are changes in water pressure that coincide with changes in other signals of the subglacial hydrology (i.e. supraglacial lake drainage, plume activity), albeit they are small. However, the water-pressure in the borehole remains close to ice overburden pressure throughout the melt season, and we would expect much larger changes in water pressure if it were connected to a channel. It is more likely that the borehole is located in an inefficiently drained region of the bed that is isolated from the major channels.

The authors understand that, in the previous version of the manuscript, changes in water-pressure were subsequently used as supporting evidence in the Discussion section. The authors realise that this should not be used as the borehole connectivity is uncertain. This has now been rectified, with all uses of the borehole record as evidence for connectivity removed. Also, we have better clarified the likely scenario that the borehole is located in a region of the bed that has inefficient drainage:

'These events also coincide with a 3 m drop in the water level at the borehole site over a 12-hour period from 28 June (Fig. 2H). It is uncertain whether the borehole is also hydraulically linked to these components. The water level in the borehole dropped when it first made contact with the bed, indicating an efficient means of drainage was present. However, the water-pressure in the borehole remains close to ice overburden pressure throughout the melt season. This suggests that either a connecting channel is consistently full of meltwater, or the borehole is located in a region of the bed that has inefficient drainage.'

*Page 18, line 16–17: No. The speedup propagates all the way up-glacier to the extent of the data shown. It would be useful (as I mentioned in my previous review) to have plots of the relative change in ice flow (i.e. percentage change for each pixel) using data from the earliest image pair as the baseline. This would show the speed-up much more clearly.*

We have now included maps showing absolute velocity differences along with the original velocity maps (derived between image pairs). These additional maps show absolute change in ice flow since 04 June 2014 (i.e. before the speed-up). Maps showing percentage difference highlighted uncertainty in low-speed areas, hence the absolute velocity difference were deemed a better solution.

These maps show that ice acceleration is uniform across the lower 3 km area of the glacier tongue during the early-season speed-up event. However, the south region of the glacier tongue still consistently flows faster than the north region. As a result, we have re-directed our discussion, looking at spatial pattern in velocity as a separate topic from the early-season speed-up event. For this reasons, it was decided to keep the original maps in Figure 5 along with the new maps showing absolute velocity differences.

As stated previously in the response to the editor, this does not drastically modify our conclusions, but we no longer link the upward-propagating drainage event with the speed-up event. Instead, we have focused on the idea that spatial differences in surface velocity throughout the season are likely to reflect differences in drainage efficiency between the north and south region of the glacier tongue.

*Page 18, line 17: 'appear to be' should be 'are'*
Agreed. Change made.

*Page 18, line 27–28: But you have already admitted that the borehole is unlikely to be connected to*

*the main hydrological system. Therefore you cannot make inferences about the extent of variations in meltwater storage at the bed from the borehole data.*

Arguments based on the assumption that the borehole is connected to a channel have now been removed from the manuscript, including this one. The paragraph now ends with:

'Cycles of internal storage and release in the subglacial environment could be an additional control on subglacial outflow.'

And this is addressed in more detail in the Disucssion section.

*Page 18, line 25: 'influenceS'*

Change made.

*Page 18, line 26: 'aN'*

Change made.

*Page 20, line 13: The speed-up IS a change in glacier dynamics...!*

Agreed. This is now more confidently written.

*Page 20, line 21: 'tongue is a heavily crevassed surface' could be 'tongue is heavily crevassed'*

Agreed. Change made.

*Page 20, line 23: Why a possibility? Why don't you check? Timelapse? Satellite data?*

The reviewer is right to suggest that this can be followed up. The time-lapse images from Kronebreen were used to investigate whether snow cover was present in the early part of the melt season. We found that snow cover gradually diminishes over the beginning of June, with bare ice exposed from 17 June after a small rainfall event (3.2 mm in a 24-hour period, visible in Fig 2D). This has now been used as evidence in the Interpretation section to support the idea that less water could be stored at the glacier surface in the early part of the melt season. This is then referred to in the Discussion section:

'...Observations from the time-lapse images show that bare ice is visible from mid-June, after a small rainfall event on the 17 June. Also, the lower area of the glacier tongue is heavily crevassed, providing abundant meltwater pathways to the glacier bed. It is therefore likely that surface meltwater is bypassing storage in the snowpack earlier than the model predicts, and the model under-represents pathways from the surface to the bed.' (Section 6.1 Interpretation: Beginning of the melt season (May–June). Paragraph 2, final 3 sentences)

'Surface velocities gradually rise at the beginning of the melt season, from mid-June onwards. As previously noted, it is likely that this early-season speed-up is linked to an increase in calving retreat at the terminus (Luckman et al., 2015), which reduced the upstream resistive stresses (Howat et al., 2005; Nick et al., 2009)....' (Section 7.1 Discussion: Early melt season meltwater storage. Paragraph 1, Sentences 1–2)

*Page 20, line 27–28: Even with a distributed subglacial drainage system? Wouldn't you expect the plume to be more spread out along the calving front if this is the case? Do you see this?*

This section attempts to outline potential reasons for the activation of the plume on the north side of the terminus from 25 June. The passage outlines two likely scenarios – 1) A sufficient volume of meltwater is discharged, causing a plume to surface in the fjord; 2) A channel is established to evcuate meltwater from a single source, which surfaces in the fjord. The time-lapse imagery has a clear view of the plume activity on the north side of the terminus, and it is clear that the plume is concentrated in

one spot at the beginning of the plume activity. Rarely does the plume spread out across the front. Most often, water surfaces in the fjord from between one and three sources (i.e. Plume N1, N2 and N3). Therefore the second of these outlined scenarios is more likely.

This information has now been added to the manuscript to strengthen the argument that initial plume activity is indicative of the formation of a channel at the front of Kronebreen:

'This meltwater is being delivered to the bed and stored for a significant period of time before it is efficiently evacuated from the subglacial system. The activation of the main plume on the north side of the terminus (N1) suggests that either a sufficient volume of meltwater is being discharged to surface in the fjord, or an efficient system is established to evacuate meltwater on 25 June. The second of these instances is more likely as the plume was observed to be surfacing from a single source (based on observations from the time-lapse imagery), signifying that it was channel-fed. This being the case, meltwater is stored at the bed for ~15 days before it is evacuated, based on the timing of the onset of the speed-up and the activation of Plume N1.'

*Page 21, line 13: 'coincident timing' could be 'coincidence'*
Agreed. Suggested change has been made.

*Page 21, line 16: 'thr' should be 'the'*
Sentence deleted. Paragraph has now been altered to better explore causes of the lake drainage using surrounding literature (namely Stevens et al., 2013 and Everett et al., 2016):

'Lake drainage is linked to longitudinal stretching which occurs in response to a change in glacier dynamics (i.e. ice speed, calving activity), and changes in conditions at the bed which promote enhanced basal sliding (Stevens et al., 2013; Everett et al., 2016). The drainage of the lakes at Kronebreen are likely to be linked to both a change in glacier dynamics and a change in conditions at the bed, namely due to an increase in meltwater at the bed. Longitudinal stretching occurs as the glacier accelerates at the beginning of the season, which facilitates the opening of crevasses and creates more pathways for meltwater to be delivered to the bed. The lakes either drain due to hydro-fracturing which is promoted by the speed-up, or they drain when they become linked to a common channelised system. The hydraulic potential modelling supports this as it indicates that Cluster 1 may be situated close to a large channel/flow accumulation pathway. The upglacier-propagating nature of their drainage indicates that this is an early-season 'flushing event' that occurs in an upglacier progression, as reflected in the timing of their connection to the subglacial environment.'

*Page 22, line 9: Sufficiently melted the cavity wall for what?*
To allow for a sufficient increase in discharge. This detail has been added to this section and the sentence has now been split into two so that the idea remains clear:

'Hydraulic pulsing represents a periodic flushing of meltwater in the local vicinity. This occurs when sufficient pressure has accumulated to force a channel open, and/or when subglacial water has melted the cavity/conduit wall to allow a sufficient increase in discharge.'

In addition, studies on hydraulic pulsing are also referred to here to better outline causes of this phenomena.

*Page 22, line 10–11: This is really not worth including without some further analysis: for a start, does the pulsing show any variation on tidal frequencies?*
Tidal level has now been included as a data set (presented in Figure 2) to examine whether there are links between hydraulic pulsing and tide. This has now been included as a line of evidence in this section to strengthen the argument that the pusling is not linked to tidal frequencies:

'Few links are observed between plume outflow and tidal level, which suggests that this is an internally-driven process and the influence of marine dynamics is limited.'

*Page 22, line 32–33: But you have already admitted that the borehole is unlikely to be connected to the main hydrological system. Therefore you cannot make inferences about the diurnal subglacial water pressure variations from the borehole data.*

The first two paragraphs of the discussion section now merely summarise the borehole record and state that it is likely the borehole record represents a region of the bed that is inefficiently drainaged for a large part of the melt season. The subsequent paragraphs examine the chain of events at the beginning of the melt season, and all references to the borehole have been removed. Therefore the manuscript has been changed so that no inferences using the borehole connectivity are made.

*Page 23, line 8: But not every inch of the bed will be connected to the channels even in an 'efficient catchment' (although Im not sure what that is)*

Yes, it is not feasible to suggest that the borehole is only hydraulically connected to some degree because not every inch of the bed will be connected to channels. We have now stated more confidently that the most likely scenario is that the borehole is located in region of the bed that is inefficiently drained, and isolated from the main channel network. This paragraph has now been modified to better convey this:

'Few short-term pressure variations are observed in the water-pressure record from May–September 2014, apart from the significant drop in pressure at the end of the melt season. Although the modelled hydraulic potential suggests that the borehole is located within an efficient drainage catchment, it is more likely that the borehole is actually indicative of a region that is inefficiently drained for a large part of the melt season.'

*Page 24, line 24: Really unprecedented? How do you know? Within the last few years only?*

Unprecedented is perhaps not a suitable word-choice here. It is a high-rainfall event that is atypical in terms of the amount of rain that fell at that time of the year. This wording has now been changed to better reflect that this event is merely rare, rather than unprecedented:
'It is likely that this speed-up was caused by an unusually-high rainfall event...'

*Page 25, line 1–2: 'the basal pressure environment' should be 'basal water pressure'*

Agreed. Change made.

[revised manuscript text omitted]

---

## Author Response (AR3)

**Authors' response to editor's comments for 'Rapidly-changing subglacial hydrological pathways at a tidewater glacier revealed through simultaneous observations of water pressure, supraglacial lakes, meltwater plumes and surface velocities'**
* * *
**Response to the Editor**

*Dear P. How,*

*I thank the authors for their careful and substantial revisions in response to a critical but overall positive re-review of the earlier revised version and my own editor-comments. The few more substantial points and minor editing issues have now been satisfactorily addressed and I am pleased to announce that the paper is accepted for publication in TC after correction of the very few minor technical points listed below.*

*At this stage I would also like to thank the authors for their excellent handling of revisions after the re-review.*

*Best regards*
*Andreas Vieli, the editor, 15th Oct 2017*

Dear Prof. Vieli,

We are grateful for your quick response to the second revision of our manuscript. It is great news that the paper has been accepted for publication and we have included our response to your minor corrections below, along with a revised version of the paper.

We would also like to thank you and the reviewers for your feedback throughout this review. Your input has been very helpful in improving our study. As my (Penelope How) first first-author publication, this has been a pleasant and constructive process. It is experiences like this that make all the difference.

Best wishes,
Penelope How, and the authors of this manuscript (16 October 2017)
* * *
**Minor comments**

*p. 7 line 19: ...from... (instead of fom)*
    Agreed. Change made.

*p. 8 line 8: maybe specify: ...2m SPATIAL resolution...*
    Agreed. Change made.

*Fig. 2 (p. 10), caption: in C) it is not clear what the black crosses are (not in legend), are these refered*

*to as the max. plume extent as mentioned in caption??? Clarify.*

The legends in Figure 2 were amended during the previous round of revisions and we accidentally did not include a symbol for the black crosses which denote 'max. plume extent'. This has now been corrected and the legend for Figure 2C now includes this.

*p. 12 line 7: is changeable the right word here, would variable not fit better!*

Agreed. Change made.

*p. 12 line 9: i am just a bit curious here, that strong precipitation affects tides. Is it also related to a wind/storm surge?*

Yes, it is likely that this is related to a storm surge. We could distinguish this from our time-lapse images over Kronebreen during this time period.

*p. 15 line 4: i think ...to the floatation level (... is more appropriate.*

Agreed. Change made.

*p. 22 line 17: atypical tidal fluctuations related to LOW PRESSURE? I miss a bit more explanation here (what is link?). is this related over a storm surge?*

Yes, this is likely to be related to a storm surge (as clarified in the previous related comment). We have now included this in the sentence:
'The atypical fluctuations in tidal level further suggest that this rainfall event is associated with a low-pressure weather front and an associated storm surge.'

*p. 26 line 27: i think the at between velocities and throught should be deleted.*

Agreed. Change made.

*p. 27 line 19: to be consistent throughout, replace unprecdented with unusually*

Agreed. Change made.

*p. 28 line 10: i think the to between tentatively and in previous years should be deleted.*

Agreed. Change made.

[revised manuscript text omitted]